# PRIVACY AMPLIFICATION BY ITERATION WITH PROJECTED ALTERNATING DIRECTION METHOD

## ABSTRACT

Alternating direction method of multipliers (ADMM) is a common approach for privacy amplification and utility guarantees in various machine learning tasks, especially those require cooperation between private and public users (or servers). However, this approach cannot achieve exact feasibility constraint throughout the learning process, and even has a large feasibility gap at the early iterative stage, which cannot handle the small-sampled situations. To solve these problems, we propose a projected alternating direction method that achieves exact feasibility and enables each user to supervise the objective value throughout the learning process. Moreover, it allows both Gaussian and Laplace noise for variable masking and privacy amplification. Third, it does not require the Markov operator condition or double-iterations to achieve one-step privacy and utility guarantees. Fourth, it achieves the same order of privacy-utility tradeoff as that of the existing ADMM methods. In summary, the proposed methodology requires fewer conditions but solves more general privacy amplification problems and enjoys more favorable properties than the existing ADMM methods.

## 1 INTRODUCTION

Privacy preservation and amplification via machine learning methodologies have gained extensive attention recently. Major efforts have been paid to the field of convex optimization to incorporate differential privacy. For instance, Feldman et al. (Feldman et al., 2018); Altschuler and Talwar (Altschuler & Talwar, 2022) demonstrate that the Rényi differential privacy bound for the projected noisy stochastic gradient descent (PNSGD) or federated PNSGD algorithm can be amplified or controlled through iteration. Such works mainly focus on the privacy amplification problem of the following convex optimization model:

$$\min_{\boldsymbol{x} \in \mathbb{R}^p} F(\boldsymbol{x}) \quad s.t. \ \boldsymbol{x} \in \mathcal{C}, \tag{1}$$

where $\boldsymbol{x}$ and $\mathcal{C} \subseteq \mathbb{R}^p$ denote the parameters to be optimized and a convex feasible set for $\boldsymbol{x}$, respectively. $F : \mathbb{R}^p \to \mathbb{R}$ is a convex and differentiable objective function. **To better highlight our motivation, we omit the additive noise mechanism for privacy preservation at this stage.** Then the projected gradient descent (PGD) algorithm for solving (1) is:

$$\boldsymbol{x}_{(k)} = \Pi_{\mathcal{C}}(\boldsymbol{x}_{(k-1)} - \eta \nabla F(\boldsymbol{x}_{(k-1)})), \quad k = 1, 2, \cdots \tag{2}$$

where $\eta > 0$ is the learning rate, and $\Pi_{\mathcal{C}}$ is a projection operator onto $\mathcal{C}$. The projection is implemented in each iteration to ensure that each resulted $\boldsymbol{x}_{(k)}$ is feasible before being brought into the next iteration.

(1) is further extended to a more general joint optimization by both private and public users, which is applicable to centralized, federated, and fully decentralized learning scenarios (Cyffers et al., 2023; Chan et al., 2024).

$$\min_{\boldsymbol{x} \in \mathbb{R}^p, \boldsymbol{y} \in \mathbb{R}^l} \{F(\boldsymbol{x}) + g(\boldsymbol{y})\} \quad s.t. \ \boldsymbol{A}\boldsymbol{x} + \boldsymbol{B}\boldsymbol{y} = \boldsymbol{c}, \tag{3}$$

where $F$ is convex and differentiable, and $g : \mathbb{R}^l \to \mathbb{R}$ is convex but not necessarily differentiable. In this model, $F$ and $\boldsymbol{x}$ contain **private** information, while $g$ and $\boldsymbol{y}$ are **publicly known**. Note that $\boldsymbol{x}$ and $\boldsymbol{y}$ can have different dimensionalities $p$ and $l$, and they are linearly transformed to the same range $\mathbb{R}^m$ by $\boldsymbol{A} \in \mathbb{R}^{m \times p}$ and $\boldsymbol{B} \in \mathbb{R}^{m \times l}$, respectively. **This constraint is much more complicated than $x \in \mathcal{C}$ in (1), especially when adding the noise for privacy preservation.**

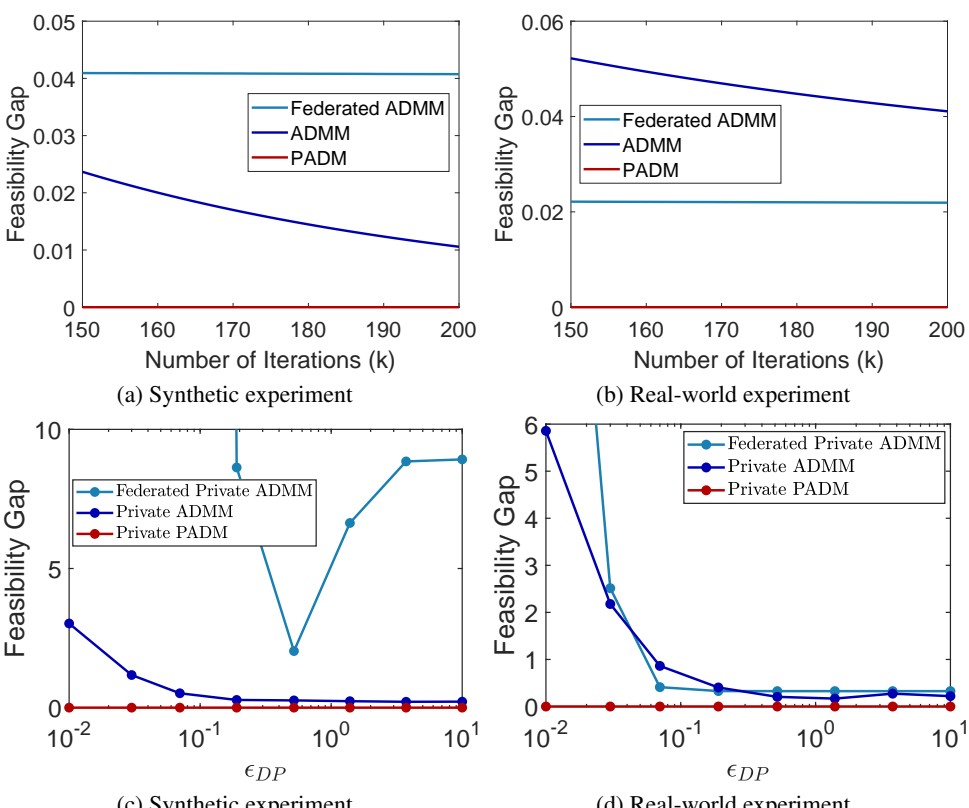

Figure 1: Comparisons of feasibility gaps between federated ADMM (Cyffers et al., 2023), ADMM (Chan et al., 2024) and PADM (ours) in the synthetic and real-world experiments. (a) and (b) present the experimental results from the 150-th to the 200-th iteration for the non-private versions. (c) and (d) present the experimental results at the 1000-th iteration under eight privacy budgets $\epsilon_{DP}$ for the private versions.

A common algorithm for solving (3) is the Alternating Direction Method of Multipliers (ADMM). In this setting, the feasibility constraint is absorbed in the Lagrangian function:

$$\mathscr{L}(\boldsymbol{x}, \boldsymbol{y}, \boldsymbol{\zeta}) := F(\boldsymbol{x}) + g(\boldsymbol{y}) + \boldsymbol{\zeta}^\top (\boldsymbol{A}\boldsymbol{x} + \boldsymbol{B}\boldsymbol{y} - \boldsymbol{c}) + \frac{\beta}{2}\|\boldsymbol{A}\boldsymbol{x} + \boldsymbol{B}\boldsymbol{y} - \boldsymbol{c}\|_2^2, \qquad (4)$$

where $\boldsymbol{\zeta} \in \mathbb{R}^m$ is the dual variable (Lagrangian multipliers). **The feasibility constraint $\boldsymbol{A}\boldsymbol{x} + \boldsymbol{B}\boldsymbol{y} = \boldsymbol{c}$ cannot be exactly satisfied (with zero residual), because the ADMM algorithm has to stop in finite iterations after reaching a preset convergence gap $\varepsilon > 0$ or running out of samples; unless the number of iterations $k = \infty$, which is impossible in real-world computation. Hence ADMM cannot achieve exact feasibility throughout the whole learning process (see Figures 1a and 1b). Worse still, if private user samples are insufficient (such as an online cooperative learning task with limited and unknown participants), then ADMM has to be terminated with insufficient iterations and a large feasibility gap (primal residual) $\|\boldsymbol{A}\boldsymbol{x}_{(k)} + \boldsymbol{B}\boldsymbol{y}_{(k)} - \boldsymbol{c}\|_2$. In private settings, the noise vectors introduced for privacy preservation even further enlarge the feasibility gaps (see Figures 1c and 1d).**

To address the above problems, **we propose a novel Projected Alternating Direction Method (PADM) for privacy amplification by iteration, which requires fewer conditions but solves more general privacy amplification problems with more favorable properties than the existing ADMM methods.** Our main contributions can be summarized as follows:

**1.** We simultaneously add a noise mechanism to the range $\boldsymbol{A}\boldsymbol{x}$ instead of the domain $\boldsymbol{x}$ and conduct a projection, which not only amplifies privacy but also achieves exact feasibility in each iteration without use of dual variable $\boldsymbol{\zeta}$.

**2.** With Item 1, PADM enables each private user to supervise the objective value under exact feasibility.

**3.** PADM is a one-pass algorithm that visits each sample for only once, which results in a computational complexity of $\mathcal{O}(n)$. It achieves the same order of privacy-utility tradeoff as that of the existing ADMM methods while fixing their feasibility problem.

**4.** PADM requires neither the Markov operator condition and double-iterations like (Chan et al., 2024), nor the smoothness and classic strong convexity conditions like (Cyffers et al., 2023). Hence PADM is able to tackle more general privacy amplification scenarios with less requirements.

## 2 PRELIMINARIES AND RELATED WORKS

Before proposing the novel methodology, we review the concepts of differential privacy and privacy amplification by iteration.

### 2.1 DIFFERENTIAL PRIVACY

We consider an Euclidean Space $\mathbb{R}^p$ equipped with $\ell_2$ norm and the Lebesgue measure, and use the notation $\mathcal{D} := (\boldsymbol{d}_{(1)}, \boldsymbol{d}_{(2)}, \ldots, \boldsymbol{d}_{(n)})$ for a private data set of size $n$. **Two data sets $\mathcal{D}$ and $\mathcal{D}'$ are defined as neighboring and denoted by $\mathcal{D} \sim \mathcal{D}'$, if they differ in at most one element** $\boldsymbol{d}_{(k)} \neq \boldsymbol{d}'_{(k)}$.

**Definition 1** (($\epsilon, \vartheta$)-Differential Privacy, Dwork et al. 2006)**.** A randomized mechanism $\mathcal{A}$ satisfies $(\epsilon, \vartheta)$-differential privacy (($\epsilon, \vartheta$)-DP, $\epsilon \geqslant 0$, $\vartheta \geqslant 0$) if for any neighboring data sets $\mathcal{D} \sim \mathcal{D}'$ and for all events $\mathcal{S}$ in the output space of $\mathcal{A}$, the following inequality holds:
$$Pr[\mathcal{A}(\mathcal{D}) \in \mathcal{S}] \leqslant e^\epsilon Pr[\mathcal{A}(\mathcal{D}') \in \mathcal{S}] + \vartheta.$$
When $\vartheta = 0$, the mechanism $\mathcal{A}$ satisfies pure $\epsilon$-differential privacy ($\epsilon$-DP).

A more generalized version of differential privacy can be derived using the concept of Rényi divergence.

**Definition 2** (Rényi Divergence, Rényi 1961)**.** The Rényi divergence of order $\alpha > 1$ between two random variables $\mathcal{A}(\mathcal{D})$ and $\mathcal{A}(\mathcal{D}')$ is defined as follows:
$$D_\alpha\big(\mathcal{A}(\mathcal{D})\|\mathcal{A}(\mathcal{D}')\big) := \frac{1}{\alpha - 1} \ln \int \left(\frac{P_\mathcal{A}(\boldsymbol{z})}{P_{\mathcal{A}'}(\boldsymbol{z})}\right)^\alpha P_{\mathcal{A}'}(\boldsymbol{z}) d\boldsymbol{z},$$
where $P_\mathcal{A}$ and $P_{\mathcal{A}'}$ denote the densities of $\mathcal{A}(\mathcal{D})$ and $\mathcal{A}(\mathcal{D}')$, respectively. Let $\alpha \to \infty$, then $D_\infty$ is defined as follows:
$$D_\infty\big(\mathcal{A}(\mathcal{D})\|\mathcal{A}(\mathcal{D}')\big) := \sup_{\boldsymbol{z} \in \text{supp}(P_{\mathcal{A}'})} \ln \frac{P_\mathcal{A}(\boldsymbol{z})}{P_{\mathcal{A}'}(\boldsymbol{z})}.$$

Then the $(\alpha, \epsilon)$-Rényi differential privacy can be defined as follows:

**Definition 3** (($\alpha, \epsilon$)-Rényi Differential Privacy)**.** A randomized mechanism $\mathcal{A}$ satisfies $(\alpha, \epsilon)$-Rényi differential privacy (($\alpha, \epsilon$)-RDP, $\alpha > 1$ and $\epsilon \geqslant 0$) if for any neighboring data sets $\mathcal{D}$ and $\mathcal{D}'$, the following inequality holds:
$$D_\alpha\big(\mathcal{A}(\mathcal{D})\|\mathcal{A}(\mathcal{D}')\big) \leqslant \epsilon.$$
From the definitions of $\epsilon$-DP and $(\infty, \epsilon)$-RDP, it can be concluded that a randomized mechanism $\mathcal{A}$ is $\epsilon$-DP if and only if it is $(\infty, \epsilon)$-RDP. Moreover, due to the monotonicity of the Rényi divergence, $(\infty, \epsilon)$-RDP implies $(\alpha, \epsilon)$-RDP for all finite $\alpha$ (Mironov, 2017). Hence $\epsilon$-DP is a stronger privacy-preserving property.

For concrete examples of differential privacy, a Laplace mechanism and a Gaussian mechanism satisfy $\epsilon$-DP and $(\alpha, \epsilon)$-RDP, respectively (Mironov, 2017). Specifically, suppose $\mathcal{T} : (\mathbb{R}^p)^n \to \mathbb{R}$ is a mechanism that depends on $\mathcal{D}$ and $z \in \mathbb{R}$ is sampled from $Laplace(0, \lambda)$, then the Laplace mechanism $\mathcal{A}(\mathcal{D}) := \mathcal{T}(\mathcal{D}) + z$ satisfies $\frac{\Delta_1}{\lambda}$-DP, where $\Delta_1 := \max_{\mathcal{D} \sim \mathcal{D}'} |\mathcal{T}(\mathcal{D}) - \mathcal{T}(\mathcal{D}')|$. Similarly, suppose $\mathcal{T} : (\mathbb{R}^p)^n \to \mathbb{R}^m$ and $\boldsymbol{z} \in \mathbb{R}^m$ is sampled from $\mathcal{N}(\boldsymbol{0}_m, \sigma^2 \boldsymbol{I}_m)$, then the Gaussian mechanism $\mathcal{A}(\mathcal{D}) := \mathcal{T}(\mathcal{D}) + \boldsymbol{z}$ satisfies $(\alpha, \frac{\alpha \Delta_2^2}{2\sigma^2})$-RDP, where $\boldsymbol{0}_m$ denotes the $m$-dimensional zero vector, $\boldsymbol{I}_m$ represents the $m$-dimensional identity matrix, and $\Delta_2 := \max_{\mathcal{D} \sim \mathcal{D}'} \|\mathcal{T}(\mathcal{D}) - \mathcal{T}(\mathcal{D}')\|_2$.

## 2.2 Privacy Amplification by Iteration

With appropriately-designed solving algorithms, privacy can be added to optimization models like (1)(3) and amplified by iteration. For example, a privacy mechanism can be added to (2) to form a PNSGD algorithm (Feldman et al., 2018; Altschuler & Talwar, 2022):

$$\boldsymbol{x}_{(k)} = \Pi_{\mathcal{C}}(\boldsymbol{x}_{(k-1)} - \eta\nabla f(\boldsymbol{x}_{(k-1)};\mathcal{D}) + \boldsymbol{z}_{(k)}), \tag{5}$$

where $f(\boldsymbol{x};\mathcal{D})$ denotes an instantiated version of the private function $F(\boldsymbol{x})$ on a given data set $\mathcal{D}$, $\boldsymbol{z}_{(k)} \sim \mathcal{N}(\boldsymbol{0}_p, \sigma^2\boldsymbol{I}_p)$ is a Gaussian noise for privacy preservation. With certain smoothness conditions of $\nabla f(\cdot;\mathcal{D})$ and a suitable learning rate $\eta$, $\psi := \Pi_{\mathcal{C}} \circ (\boldsymbol{I}_p - \eta\nabla f(\cdot;\mathcal{D}))$ can be a non-expansive operator. Given two possibly different starting points $\boldsymbol{x}_{(0)}$ and $\boldsymbol{x}'_{(0)}$, denote $\boldsymbol{x}_{(n)}$ and $\boldsymbol{x}'_{(n)}$ as the corresponding $n$-th iteration outputs of the same algorithm (5), respectively. Then according to Theorem 1 in (Feldman et al., 2018), the following privacy amplification by iteration holds:

$$D_\alpha(\boldsymbol{x}_{(n)}\|\boldsymbol{x}'_{(n)}) \leqslant \frac{\alpha\|\boldsymbol{x}_{(0)} - \boldsymbol{x}'_{(0)}\|_2^2}{2n\sigma^2}. \tag{6}$$

The essence of amplification is that as the number of iterations $n$ increases, the Rényi divergence between $\boldsymbol{x}_{(n)}$ and $\boldsymbol{x}'_{(n)}$ decreases, for example, in an order of $\mathcal{O}(\frac{1}{n})$. Hence the initial difference $(\boldsymbol{x}_{(0)} - \boldsymbol{x}'_{(0)})$ can be hidden by sufficient iterations. Moreover, if each of the $n$ users provides one private sample in order, the first user will gain the largest advantage in hiding the private information.

Similarly, a feasible ADMM algorithm for model (3) with privacy amplification by iteration can be (Cyffers et al., 2023):

$$\begin{cases} \boldsymbol{y}_{(k)} = \arg\min_{\boldsymbol{y}} \left\{ g(\boldsymbol{y}) + \frac{\beta}{2}\|\boldsymbol{B}\boldsymbol{y} + \boldsymbol{\zeta}_{(k-1)}\|_2^2 \right\} \\ \boldsymbol{x}_{(k)} = \arg\min_{\boldsymbol{x}} \left\{ f(\boldsymbol{x};\mathcal{D}) + \frac{\beta}{2}\|\boldsymbol{A}\boldsymbol{x} + 2\boldsymbol{B}\boldsymbol{y}_{(k)} + \boldsymbol{\zeta}_{(k-1)} - \boldsymbol{c}\|_2^2 \right\} \\ \boldsymbol{\zeta}_{(k)} = \boldsymbol{\zeta}_{(k-1)} + 2\eta(\boldsymbol{A}\boldsymbol{x}_{(k)} + \boldsymbol{B}\boldsymbol{y}_{(k)} - \boldsymbol{c} + \frac{1}{2}\boldsymbol{z}_{(k)}) \end{cases} . \tag{7}$$

Chan et al. (2024) also develop a similar ADMM scheme except that only the gradient value $\nabla f(\boldsymbol{x}_{(k-1)};\mathcal{D})$ needs to be revealed to the $k$-th user. To achieve one-step privacy and utility guarantees, it requires **double iterations to exploit the nonexpansiveness property twice**: one in primal variable $\boldsymbol{x}$ and one in dual variable $\boldsymbol{\zeta}$. Besides, **the Markov operator condition ensures that the nonexpansiveness property can be extended from the primal space $\boldsymbol{x}$ to the joint space of $(\boldsymbol{x}, \boldsymbol{\zeta})$**. In such ADMM schemes, the feasibility constraint $\boldsymbol{A}\boldsymbol{x} + \boldsymbol{B}\boldsymbol{y} = \boldsymbol{c}$ cannot be exactly satisfied (with zero residual), because the ADMM algorithm has to stop in finite iterations after reaching a preset convergence gap $\varepsilon > 0$ or running out of samples; unless the number of iterations $k = \infty$, which is impossible in real-world computation. Worse still, if the number of samples is limited or the number of iterations $k$ is small, the feasibility gap can be very large (see Figure 1). **In this case, each private user (especially that at the early iterative stage) cannot supervise the objective value under exact feasibility.**

Most related works mainly focus on the Gaussian noise mechanism (Feldman et al., 2018; Altschuler & Talwar, 2022; Chan et al., 2024), but seldom address the Laplace noise mechanism. In fact, the Laplace mechanism enjoys the good property of $\epsilon$-DP, which is a good complement to the $(\alpha, \epsilon)$-RDP of the Gaussian mechanism. Hence we establish both Gaussian and Laplace noise mechanisms for the proposed methodology.

## 3 Methodology

**Our methodology is applicable to privacy amplification by either iteration or subsampling.** Since both approaches share most technical details, we mainly illustrate the privacy amplification by iteration for the joint optimization model (3). In our settings, both $F$ and $g$ are convex but not necessarily differentiable, which is more general than (Chan et al., 2024).

### 3.1 Projected Alternating Direction Method

We first develop the PADM for solving (3) without adding any noise mechanism of privacy preservation. As explained above, an ADMM algorithm involves a dual variable that cannot achieve exact

feasibility. To tackle this problem, **we project the current iterate onto the feasible set in every iteration**. This results in a PADM that allows each user to supervise the current objective value under exact feasibility. Therefore, this PADM is particularly suitable for situations with limited or unknown sample sizes, making it preferable to ADMM algorithms.

We begin with defining the following **feasibility set in the extended image space**:

$$\mathcal{W} := \{\boldsymbol{w} \in \mathbb{R}^{2m} : [\boldsymbol{I}_m \; \boldsymbol{I}_m] \cdot \boldsymbol{w} = \boldsymbol{c}\}. \tag{8}$$

Then $\boldsymbol{u} := \boldsymbol{A}\boldsymbol{x} \in \mathbb{R}^m$ and $\boldsymbol{v} := \boldsymbol{B}\boldsymbol{y} \in \mathbb{R}^m$ are actually joint in the image space by the feasibility constraint $\begin{bmatrix} \boldsymbol{u} \\ \boldsymbol{v} \end{bmatrix} \in \mathcal{W}$. Conversely, for $\boldsymbol{u} \in \mathbb{R}^m$, $\boldsymbol{v} \in \mathbb{R}^m$ and $\boldsymbol{w} := \begin{bmatrix} \boldsymbol{u} \\ \boldsymbol{v} \end{bmatrix} \in \mathbb{R}^{2m}$ but not necessarily $\boldsymbol{w} \in \mathcal{W}$, the Euclidean projection of $\boldsymbol{w}$ onto the feasibility set $\mathcal{W}$ can be computed by

$$\Pi_{\mathcal{W}}(\boldsymbol{w}) := \begin{bmatrix} \frac{1}{2}(\boldsymbol{u} - \boldsymbol{v} + \boldsymbol{c}) \\ \frac{1}{2}(\boldsymbol{v} - \boldsymbol{u} + \boldsymbol{c}) \end{bmatrix}. \tag{9}$$

The proposed PADM consists of three main steps:

$$\begin{cases} \hat{\boldsymbol{x}}_{(k)} = \underset{\boldsymbol{x} \in \mathbb{R}^p}{\operatorname{argmin}} \{ F(\boldsymbol{x}) + \frac{\beta}{2} \|\boldsymbol{A}\boldsymbol{x} + \boldsymbol{v}_{(k-1)} - \boldsymbol{c}\|_2^2 \}, \\ \hat{\boldsymbol{y}}_{(k)} = \underset{\boldsymbol{y} \in \mathbb{R}^l}{\operatorname{argmin}} \{ g(\boldsymbol{y}) + \frac{\beta}{2} \|\boldsymbol{u}_{(k-1)} + \boldsymbol{B}\boldsymbol{y} - \boldsymbol{c}\|_2^2 \}, \\ \begin{bmatrix} \boldsymbol{u}_{(k)} \\ \boldsymbol{v}_{(k)} \end{bmatrix} = \Pi_{\mathcal{W}} \begin{bmatrix} \boldsymbol{A}\hat{\boldsymbol{x}}_{(k)} \\ \boldsymbol{B}\hat{\boldsymbol{y}}_{(k)} \end{bmatrix}. \end{cases} \tag{10}$$

Note that $\boldsymbol{u}_{(k)}$ and $\boldsymbol{v}_{(k)}$ are always feasible in the image space, whereas $\hat{\boldsymbol{x}}_{(k)}$ and $\hat{\boldsymbol{y}}_{(k)}$ are not necessarily feasible in the preimage space at this stage (thus we add hats to them). However, assuming that both $\boldsymbol{A}$ and $\boldsymbol{B}$ are of full row rank, we can use pseudo matrices to obtain feasible $\boldsymbol{x}_{(k)}$ and $\boldsymbol{y}_{(k)}$ in the preimage space:

$$\begin{cases} \boldsymbol{x}_{(k)} := \boldsymbol{V}\boldsymbol{u}_{(k)}, \; \boldsymbol{V} := \boldsymbol{A}^\top (\boldsymbol{A}\boldsymbol{A}^\top)^{-1}, \\ \boldsymbol{y}_{(k)} := \boldsymbol{U}\boldsymbol{v}_{(k)}, \; \boldsymbol{U} := \boldsymbol{B}^\top (\boldsymbol{B}\boldsymbol{B}^\top)^{-1}. \end{cases} \tag{11}$$

If $\boldsymbol{A}$ (or $\boldsymbol{B}$) lacks full row rank, we can employ standard matrix transformations to enforce the equivalent full rank condition. If $\boldsymbol{A}$ (or $\boldsymbol{B}$) is invertible, then $\boldsymbol{V}$ (or $\boldsymbol{U}$) reduces to $\boldsymbol{A}^{-1}$ (or $\boldsymbol{B}^{-1}$). In fact, only $\boldsymbol{u}_{(k)}$ is sufficient to guarantee a complete algorithm and exact feasibility in the image space, which will be explained later. Moreover, the noise mechanism for privacy preservation is also added to the image space instead of the preimage space. Hence the retrieval of feasible $\boldsymbol{x}_{(k)}$ and $\boldsymbol{y}_{(k)}$ can be omitted if some users do not need them and want to save some computation.

### 3.2 FIXED-POINT AND NON-EXPANSIVE OPERATOR

Next, we need to formulate PADM as a fixed-point iteration and verify that it is a non-expansive operator, which is crucial for privacy amplification. To do this, we define an extended proximal operator with a convex and proper function $F : \mathbb{R}^p \to \mathbb{R}$ and a matrix $\boldsymbol{A} \in \mathbb{R}^{m \times p}$ as follows:

$$\mathcal{P}_{\beta, F, \boldsymbol{A}}(\boldsymbol{u}) := \underset{\boldsymbol{x} \in \mathbb{R}^p}{\operatorname{argmin}} \{ F(\boldsymbol{x}) + \frac{\beta}{2} \|\boldsymbol{A}\boldsymbol{x} - \boldsymbol{u}\|_2^2 \}, \tag{12}$$

where $\boldsymbol{u} \in \mathbb{R}^m$, and $\beta > 0$ is a regularization parameter. $\mathcal{P}_{\beta, g, \boldsymbol{B}}$ can be defined in a similar way.

Since $\boldsymbol{u}_{(k)} = \boldsymbol{c} - \boldsymbol{v}_{(k)}$, then PADM (10) can be reformulated as follows:

$$\hat{\boldsymbol{x}}_{(k)} = \mathcal{P}_{\beta, F, \boldsymbol{A}}(\boldsymbol{u}_{(k-1)}), \quad \hat{\boldsymbol{y}}_{(k)} = \mathcal{P}_{\beta, g, \boldsymbol{B}}(\boldsymbol{c} - \boldsymbol{u}_{(k-1)}), \quad \boldsymbol{u}_{(k)} = \boldsymbol{H} \cdot \left( \Pi_{\mathcal{W}} \left( \begin{bmatrix} \boldsymbol{A}\hat{\boldsymbol{x}}_{(k)} \\ \boldsymbol{B}\hat{\boldsymbol{y}}_{(k)} \end{bmatrix} \right) \right), \tag{13}$$

where $\boldsymbol{H} := [\boldsymbol{I}_m \; \boldsymbol{0}_{m \times m}]$ and $\boldsymbol{0}_{m \times m}$ represents the $m \times m$-dimensional zero matrix. Moreover, (13) can be further integrated into a unified operator $\mathcal{T}_F : \mathbb{R}^m \to \mathbb{R}^m$:

$$\boldsymbol{u}_{(k)} = \mathcal{T}_F(\boldsymbol{u}_{(k-1)}) := \boldsymbol{H} \cdot \left( \Pi_{\mathcal{W}} \left( \begin{bmatrix} \boldsymbol{A} \circ \mathcal{P}_{\beta, F, \boldsymbol{A}}(\boldsymbol{u}_{(k-1)}) \\ \boldsymbol{B} \circ \mathcal{P}_{\beta, g, \boldsymbol{B}}(\boldsymbol{c} - \boldsymbol{u}_{(k-1)}) \end{bmatrix} \right) \right). \tag{14}$$

**It formulates PADM as a fixed-point iteration $\mathcal{T}_F$ on $\boldsymbol{u}$.** The following theorem characterizes the fixed point(s) of $\mathcal{T}_F$.

**Theorem 3.1.** *A point $\boldsymbol{u}_* \in \mathbb{R}^m$ is a fixed point of $\mathcal{T}_F$ if and only if*

$$\boldsymbol{u}_* = \frac{\boldsymbol{A}\hat{\boldsymbol{x}}_* - \boldsymbol{B}\hat{\boldsymbol{y}}_* + \boldsymbol{c}}{2}, \quad (\hat{\boldsymbol{x}}_*, \hat{\boldsymbol{y}}_*) \in \operatorname*{argmin}_{(\boldsymbol{x}, \boldsymbol{y}) \in \mathbb{R}^{p \times l}} \{F(\boldsymbol{x}) + g(\boldsymbol{y}) + \frac{\beta}{4}\|\boldsymbol{A}\boldsymbol{x} + \boldsymbol{B}\boldsymbol{y} - \boldsymbol{c}\|_2^2\}. \tag{15}$$

The proof is provided in Appendix A.1. $(\hat{\boldsymbol{x}}_*, \hat{\boldsymbol{y}}_*)$ is a minimum point of the regularized joint objective function, which is assumed to exist in such alternating direction frameworks like (7) (Cyffers et al., 2023; Lai & Wang, 2024; Wang et al., 2025). Hence there exists at least one fixed point $\boldsymbol{u}_*$ of $\mathcal{T}_F$ according to Theorem 3.1. Moreover, a feasible solution $(\boldsymbol{x}_*, \boldsymbol{y}_*)$ can be retrieved from $\boldsymbol{u}_*$ by $\boldsymbol{x}_* = \boldsymbol{V}\boldsymbol{u}_*$ and $\boldsymbol{y}_* = \boldsymbol{U}(\boldsymbol{c} - \boldsymbol{u}_*)$, with $\boldsymbol{V}$ and $\boldsymbol{U}$ defined in (11).

Besides being a fixed-point iteration, $\mathcal{T}_F$ needs to be a non-expansive operator in order to ensure convergence and amplify privacy (Lin et al., 2024a;b). We recall the definitions of firmly non-expansive and averaged non-expansive operators.

**Definition 4** (Firmly Non-expansive Operator)**.** An operator $\mathcal{T}_F : \mathbb{R}^m \to \mathbb{R}^m$ is firmly non-expansive if for all $\boldsymbol{u}, \boldsymbol{v} \in \mathbb{R}^m$,

$$\|\mathcal{T}_F(\boldsymbol{u}) - \mathcal{T}_F(\boldsymbol{v})\|_2^2 \leqslant \langle \mathcal{T}_F(\boldsymbol{u}) - \mathcal{T}_F(\boldsymbol{v}), \boldsymbol{u} - \boldsymbol{v} \rangle,$$

where $\langle \cdot, \cdot \rangle$ denotes the inner product operator.

**Definition 5** (Averaged Non-expansive Operator)**.** An operator $\mathcal{T}_F : \mathbb{R}^m \to \mathbb{R}^m$ is $\alpha$-averaged non-expansive if there exists a non-expansive operator $\mathcal{N} : \mathbb{R}^m \to \mathbb{R}^m$ and an $\alpha \in (0, 1)$ such that $\mathcal{T}_F = (1 - \alpha)\mathcal{I} + \alpha\mathcal{N}$, where $\mathcal{I}$ is the identity operator.

**Theorem 3.2.** *The operator $\mathcal{T}_F$ defined in* (14) *is $\frac{2}{3}$-averaged non-expansive.*

The detailed proof is provided in Appendix A.3, while we give a sketch here. As the feasible set $\mathcal{W}$ is convex, the projection operator $\Pi_{\mathcal{W}}$ is firmly non-expansive. Appendix A.2 proves that $\boldsymbol{A} \circ \mathcal{P}_{\beta, F, \boldsymbol{A}}$ and $\boldsymbol{B} \circ \mathcal{P}_{\beta, g, \boldsymbol{B}}$ are also firmly non-expansive. Hence $\mathcal{T}_F$ is a composition of two firmly non-expansive operators, which results in an $\alpha$-averaged non-expansive operator.

As explained above, $\mathcal{T}_F$ has at least one fixed point. Therefore, we can use Krasnoselskii-Mann (KM) theorem (Mann, 1953; Krasnosel'skii, 1955; Bauschke & Combettes, 2017) to deduce the convergence of PADM.

**Theorem 3.3.** *Let the sequence $\{\boldsymbol{u}_{(k)}\}_{k \in \mathbb{N}}$ be generated by* (14) *and $\boldsymbol{u}_*$ be a fixed point of $\mathcal{T}_F$. Define $\boldsymbol{x}_{(k)} := \boldsymbol{V}\boldsymbol{u}_{(k)}$ for all $k$ and $\boldsymbol{x}_* := \boldsymbol{V}\boldsymbol{u}_*$. Then the sequences $\{\boldsymbol{u}_{(k)}\}_{k \in \mathbb{N}}$ and $\{\boldsymbol{x}_{(k)}\}_{k \in \mathbb{N}}$ converge to $\boldsymbol{u}_*$ and $\boldsymbol{x}_*$, respectively.*

The proof is provided in Appendix A.4.

## 3.3 PRIVATE PROJECTED ALTERNATING DIRECTION METHOD

The next step is to establish the private PADM by introducing privacy preservation or even amplification mechanisms. Let $\mathcal{R}$ be the domain of data sets, and $P_{\mathcal{R}}$ be a distribution over $\mathcal{R}$. Given a private data set $\mathcal{D} := \{\boldsymbol{d}_{(1)}, \boldsymbol{d}_{(2)}, \ldots, \boldsymbol{d}_{(n)}\}$ with $n$ samples drawn i.i.d. from $P_{\mathcal{R}}$ and a function $f(\boldsymbol{x}, \boldsymbol{d}) : \mathbb{R}^p \times \mathcal{R} \to \mathbb{R}$ that is convex in $\boldsymbol{x}$ for every fixed $\boldsymbol{d}$, the objective is to solve model (3) with $F(\boldsymbol{x}) := \mathbb{E}_{\boldsymbol{d} \sim P_{\mathcal{R}}}[f(\boldsymbol{x}, \boldsymbol{d})]$ in a privacy-preserving manner.

In this task, each iteration $k \in [n] := \{1, 2, \ldots, n\}$ is associated with a user whose private data is contained in $\boldsymbol{d}_{(k)}$. Instead of directly accessing $F$, the $k$-th user can only access $f_k(\cdot) := f(\cdot, \boldsymbol{d}_{(k)})$ in the $k$-th iteration, while the function $g$ is publicly known. The $k$-th user can further generate a noise $\boldsymbol{z}_{(k)}$ with mean zero and variance $\delta^2$ to mask the variable for privacy preservation. Then the update rule of private PADM is

$$\tilde{\boldsymbol{u}}_{(k)} := \boldsymbol{H} \cdot \left( \Pi_{\mathcal{W}} \left( \begin{bmatrix} \boldsymbol{A} \circ \mathcal{P}_{\beta, f_k, \boldsymbol{A}}(\tilde{\boldsymbol{u}}_{(k-1)}) + \boldsymbol{z}_{(k)} \\ \boldsymbol{B} \circ \mathcal{P}_{\beta, g, \boldsymbol{B}}(\boldsymbol{c} - \tilde{\boldsymbol{u}}_{(k-1)}) \end{bmatrix} \right) \right) = \mathcal{T}_{f_k}(\tilde{\boldsymbol{u}}_{(k-1)}) + \frac{\boldsymbol{z}_{(k)}}{2}, \tag{16}$$

where $\mathcal{T}_{f_k}$ is the fixed-point algorithm (14) instantiated by $F := f_k$. If needed, one can obtain a feasible intermediate solution $(\tilde{\boldsymbol{x}}_{(k)}, \tilde{\boldsymbol{y}}_{(k)})$ in each iteration $k$ using $\tilde{\boldsymbol{x}}_{(k)} = \boldsymbol{V}\tilde{\boldsymbol{u}}_{(k)}$ and $\tilde{\boldsymbol{y}}_{(k)} = \boldsymbol{U}(\boldsymbol{c} - \tilde{\boldsymbol{u}}_{(k)})$ like (11). The entire private PADM can be summarized as Algorithm 1.

Next, we derive a general utility guarantee for private PADM with some common and mild conditions.

---

**Algorithm 1:** Private PADM

---

**Input:** Data set $\mathcal{D} := (\boldsymbol{d}_{(1)}, \boldsymbol{d}_{(2)}, \dots, \boldsymbol{d}_{(n)})$, objective functions $f : \mathbb{R}^p \times \mathcal{R} \to \mathbb{R}$ and $g : \mathbb{R}^l \to \mathbb{R}$, starting point $\tilde{\boldsymbol{x}}_{(0)} \in \mathbb{R}^p$.

1: Compute $\tilde{\boldsymbol{u}}_{(0)} = \boldsymbol{A}\tilde{\boldsymbol{x}}_{(0)}$.

2: **for** $k = 1, 2, \dots, n$ **do**

3:   Sample $\boldsymbol{z}_{(k)}$ from a noise distribution $P_{\boldsymbol{z}}$ on $\mathbb{R}^m$ with mean zero and variance $\delta^2$.

4:   $\tilde{\boldsymbol{u}}_{(k)} = \boldsymbol{H} \cdot \left( \Pi_{\mathcal{W}} \left( \begin{bmatrix} \boldsymbol{A} \circ \mathcal{P}_{\beta, f_k, \boldsymbol{A}}(\tilde{\boldsymbol{u}}_{(k-1)}) + \boldsymbol{z}_{(k)} \\ \boldsymbol{B} \circ \mathcal{P}_{\beta, g, \boldsymbol{B}}(\boldsymbol{c} - \tilde{\boldsymbol{u}}_{(k-1)}) \end{bmatrix} \right) \right)$.

5: **end for**

6: Obtain a feasible $\tilde{\boldsymbol{x}}_{(n)}$ by using $\tilde{\boldsymbol{x}}_{(n)} = \boldsymbol{V}\tilde{\boldsymbol{u}}_{(n)}$.

**Output:** Feasible solutions $\tilde{\boldsymbol{x}}_{(n)}$ and $\tilde{\boldsymbol{u}}_{(n)}$ in the preimage and image spaces, respectively.

---

**Condition 1.** *There exists a constant $\mu \geqslant 0$ such that for any $\boldsymbol{u} \in \mathbb{R}^m$ and $k$, $\mathbb{E}\big(\|\mathcal{T}_{f_k}(\boldsymbol{u}) - \mathcal{T}_F(\boldsymbol{u})\|_2^2\big) \leqslant \mu^2$.*

**Definition 6** (Sub-sigma-algebra Flow)**.** *Let $\Gamma$ be a sigma-algebra on $\mathbb{R}^m$. Define $\{\Gamma_k\}_{k=0}^n$ as a sub-sigma-algebra flow of $\Gamma$ such that $\varpi(\tilde{\boldsymbol{u}}_{(0)}, \dots, \tilde{\boldsymbol{u}}_{(k-1)}) \subseteq \Gamma_{k-1} \subseteq \Gamma_k$ for all $k$, where $\varpi(\tilde{\boldsymbol{u}}_{(0)}, \dots, \tilde{\boldsymbol{u}}_{(k-1)})$ denotes the sigma-algebra generated by $\tilde{\boldsymbol{u}}_{(0)}, \dots, \tilde{\boldsymbol{u}}_{(k-1)}$.*

**Theorem 3.4** (Utility guarantee for private PADM)**.** *Under Condition 1, suppose $\mathcal{T}_F$ is $\tau$-contractive with $0 < \tau < 1$ and has a unique fixed point $\boldsymbol{u}_* \in \mathbb{R}^m$. If $\delta \in \big(0, 2\sqrt{\frac{(1-\tau^2)^2}{\tau^2} - \mu^2}\big) \neq \emptyset$, then the iterates of private PADM satisfy that for any $k \in [n]$,*

$$\mathbb{E}(\|\tilde{\boldsymbol{x}}_{(k)} - \boldsymbol{x}_*\|_2^2 | \Gamma_0) \leqslant \tilde{\tau}^k \|\boldsymbol{V}\|_2^2 \|\boldsymbol{A}\|_2^2 \cdot \|\tilde{\boldsymbol{x}}_{(0)} - \boldsymbol{x}_*\|_2^2 + \frac{\|\boldsymbol{V}\|_2^2 (1 - \tilde{\tau}^k)}{1 - \tilde{\tau}} \left( \frac{\delta^2}{4} + \mu^2 + \tau \sqrt{\frac{\delta^2}{4} + \mu^2} \right), \quad (17)$$

*where $\boldsymbol{x}_* := \boldsymbol{V}\boldsymbol{u}_*$ and $\tilde{\tau} := \tau^2 + \tau\sqrt{\frac{\delta^2}{4} + \mu^2} < 1$. The utility guarantee for $\boldsymbol{u}$ is provided in (36) along the proof.*

The proof is provided in Appendix A.5. The utility bound (17) is conceptually divided into two distinct components. The first term characterizes the standard linear convergence achieved by PADM, reflecting the rate at which the solution approaches the unperturbed optimum, with a geometric decay factor $\tilde{\tau}^k$. The second term quantifies the additional stochastic error inherent in private PADM, which arises from both the magnitude of the added privacy noise and the distance between the stochastic sample objective functions and the true objective function. Importantly, the overall convergence rate meets or exceeds the most advanced existing results in private constrained optimization, achieving the advantage convergence rate achieved by Cyffers et al. (2023), which is a better rate than the slower convergence demonstrated by Chan et al. (2024).

Next, we provide a strategy to obtain a contractive $\mathcal{T}_F$ by appropriately setting the regularization parameter $\beta$.

**Definition 7** (Induced Strong Convexity)**.** *The function $F$ is $\boldsymbol{A}$-induced $\varrho_F$-strongly convex if there exists a constant $\varrho_F > 0$ such that for any $\boldsymbol{x}, \boldsymbol{x}' \in \mathbb{R}^p$, $\boldsymbol{v} \in \partial F(\boldsymbol{x})$, and $\boldsymbol{v}' \in \partial F(\boldsymbol{x}')$,*

$$(\boldsymbol{v} - \boldsymbol{v}')^\top (\boldsymbol{x} - \boldsymbol{x}') \geqslant \varrho_F \|\boldsymbol{A}(\boldsymbol{x} - \boldsymbol{x}')\|_2^2. \quad (18)$$

**Proposition 3.1.** *The operator $\mathcal{T}_F$ is $\frac{1}{2} \cdot \big(\frac{\beta}{\beta + \varrho_F} + 1\big)$-contractive, or $\frac{1}{2} \cdot \big(1 + \frac{\beta}{\beta + \varrho_g}\big)$-contractive, or $\frac{1}{2} \cdot \big(\frac{\beta}{\beta + \varrho_F} + \frac{\beta}{\beta + \varrho_g}\big)$-contractive if $F$ is $\boldsymbol{A}$-induced $\varrho_F$-strongly convex, or $g$ is $\boldsymbol{B}$-induced $\varrho_g$-strongly convex, or both $F$ and $g$ are induced strongly convex.*

Note that the induced strong convexity is weaker than the classic strong convexity defined in (Rockafellar & Wets, 2009). Hence Proposition 3.1 is applicable to more classes of functions for $F$ and $g$. For example, $F$ can be a positive semidefinite function, or $g$ can be a ridge regularization. Details and the proof are provided in Appendix A.6.

## 3.4 PRIVACY AMPLIFICATION WITH PRIVATE PROJECTED ALTERNATING DIRECTION METHOD

The following sensitivity bound is a crucial condition for deducing privacy amplification properties.

**Condition 2** (Sensitivity Bound for the Operators $\mathcal{T}_{f_k, k \in [n]}$). *There exists a constant $C > 0$ such that for any $\boldsymbol{u} \in \mathbb{R}^m$ and $k \in [n]$,*

$$\sup_{\mathcal{D} \sim \mathcal{D}'} \|\mathcal{T}_{f_k}(\boldsymbol{u}) - \mathcal{T}_{f_k'}(\boldsymbol{u})\|_2 \leqslant C, \tag{19}$$

*where $f_k'(\cdot) := f(\cdot, \boldsymbol{d}_{(k)}')$ with $\boldsymbol{d}_{(k)}' \in \mathcal{D}'$, and $\mathcal{D}'$ is any neighboring data set of $\mathcal{D}$ as defined in Section 2.1.*

Condition 2 can be directly obtained if $f(\cdot, \boldsymbol{d})$ is convex, $L$-Lipschitz and differentiable over $\mathbb{R}^p$ for any $\boldsymbol{d} \in \mathcal{R}$.

**Proposition 3.2.** *Suppose $\{f(\cdot, \boldsymbol{d})\}_{\boldsymbol{d} \in \mathcal{R}}$ is a family of convex, differentiable and $L$-Lipschitz functions over $\mathbb{R}^p$, the matrix $(\boldsymbol{A}^\top \boldsymbol{A})$ is positive definite with the smallest eigenvalue $\omega_A > 0$. Then for any $k \in [n]$ and $\boldsymbol{u} \in \mathbb{R}^m$,*

$$\sup_{\mathcal{D} \sim \mathcal{D}'} \|\mathcal{T}_{f_k}(\boldsymbol{u}) - \mathcal{T}_{f_k'}(\boldsymbol{u})\|_2 \leqslant \frac{\sqrt{2} L \|\boldsymbol{A}\|_2}{\beta \omega_A}. \tag{20}$$

The proof is provided in Appendix A.7.

With Condition 2, we can obtain privacy preservation properties for private PADM in either case: when the privacy noise is generated from a Gaussian distribution (denoted by PADM-$\mathcal{N}_\sigma$) or a joint Laplace distribution (denoted by PADM-$\mathcal{L}_\lambda$). In this subsection, we analyze PADM-$\mathcal{N}_\sigma$, in which the privacy noise is sampled from a Gaussian distribution $\mathcal{N}(\boldsymbol{0}_m, \sigma^2 \boldsymbol{I}_m)$ for some $\sigma > 0$. **The corresponding analysis for PADM-$\mathcal{L}_\lambda$ is provided in Appendix A.11.**

The following theorem indicates that increasing the number of processed iterations $(n - i)$ yields a tighter privacy guarantee.

**Theorem 3.5.** *Under Condition 2, the final ($n$-th) output of PADM-$\mathcal{N}_\sigma$ achieves $\left(\alpha, \frac{2\alpha C^2}{\sigma^2(n-i+1)}\right)$-RDP for its $i$-th input.*

The proof is provided in Appendix A.8. By setting $n = i$ in Theorem 3.5, we can obtain that PADM-$\mathcal{N}_\sigma$ achieves local $(\alpha, \frac{2\alpha C^2}{\sigma^2})$-RDP for any user $k \in [n]$. We also consider a variant of PADM-$\mathcal{N}_\sigma$ with random stopping, which provides a similar function to that of (Feldman et al., 2018), denoted by Stop-PADM-$\mathcal{N}_\sigma$. It randomly terminates and outputs $\tilde{\boldsymbol{u}}^K$, where $K$ is uniformly and randomly selected from $[n]$. It also achieves privacy amplification by iteration.

**Theorem 3.6.** *Under Condition 2, the random output of Stop-PADM-$\mathcal{N}_\sigma$ for $\sigma \geqslant C\sqrt{2\alpha(\alpha-1)}$ achieves $\left(\alpha, \frac{4\alpha C^2 \ln n}{n\sigma^2}\right)$-RDP.*

The proof is provided in Appendix A.9. We also examine the privacy-utility tradeoff of Stop-PADM-$\mathcal{N}_\sigma$ in Theorem 3.7. Specifically, as the privacy budget $\epsilon$ is increased, the expected distance between the random output of Stop-PADM-$\mathcal{N}_\sigma$ and the solution derived from $\mathcal{T}_F$ decreases.

**Theorem 3.7** (Privacy-utility Tradeoff of Stop-PADM-$\mathcal{N}_\sigma$). *Suppose the operator $\mathcal{T}_F$ is $\tau$-contractive with some $0 < \tau < 1$ and has a fixed point $\boldsymbol{u}_* \in \mathbb{R}^m$. Under Conditions 1 and 2, if $\mu = 0$ and $\sigma < \frac{2(1-\tau^2)}{\sqrt{m}\tau}$, then Stop-PADM-$\mathcal{N}_\sigma$ satisfies*

$$\mathbb{E}(\|\tilde{\boldsymbol{x}}_{(K)} - \boldsymbol{x}_*\|_2^2 | \Gamma_0) \leqslant \tilde{\mathcal{O}}\left(\frac{1}{1 - (\tau^2 + \tau C\sqrt{m\alpha})} \cdot \left(\frac{1}{n} + \frac{m\alpha C^2 + \sqrt{m\alpha}C}{n\epsilon}\right)\right), \tag{21}$$

*where $\boldsymbol{x}_* = \boldsymbol{V}\boldsymbol{u}_*$, $\alpha > 1$ and $\epsilon$ are the privacy parameters such that the output $\tilde{\boldsymbol{x}}_{(K)}$ of Stop-PADM-$\mathcal{N}_\sigma$ achieves $(\alpha, \epsilon)$-RDP.*

The proof is provided in Appendix A.10. **Note that the privacy-utility tradeoffs of private PADM for both Gaussian (Theorem 3.7) and Laplace (Theorem A.3) mechanisms are in the same order as those of federated ADMM (decentralized setting, Cyffers et al. 2023), while PNSGD (Feldman et al., 2018) and private ADMM (Chan et al., 2024) cannot have such privacy-utility tradeoffs for model (3). PNSGD does not consider the public function $g(\boldsymbol{y})$, and private ADMM can only have privacy-utility tradeoffs of the objective function instead of the argument. Moreover, federated ADMM cannot achieve exact feasibility, while PADM always can. Hence PADM is more advantageous over these baselines.**

## 4 Experiments

We evaluate the performance of PADM-$\mathcal{N}_\sigma$ and PADM-$\mathcal{L}_\lambda$ through two experiments: (1) a linear regression task on synthetic data following the benchmark settings of (Cyffers et al., 2023; Chan et al., 2024), and (2) a logistic regression task on real-world data from the UCI Machine Learning Repository. Further details on evaluation model, data processing and parameter settings are provided in Appendix A.12 and A.13.

The performance of PADM-$\mathcal{N}_\sigma$ and PADM-$\mathcal{L}_\lambda$ is examined under different privacy budgets, specifically focusing on the perspective of the first user with respect to (w.r.t.) the final output of the algorithm. The privacy budget is measured in terms of $(\epsilon_{DP}, \vartheta)$-DP to facilitate comparison with other methods. Eight distinct values of $\epsilon_{DP}$ are evenly spread from $0.01$ to $10$ in the logarithmic scale, while $\vartheta$ is fixed at $1e-6$. This range of $\epsilon_{DP}$ represents a wide range of privacy levels from rigorous (low $\epsilon_{DP}$) to more relaxed (high $\epsilon_{DP}$).

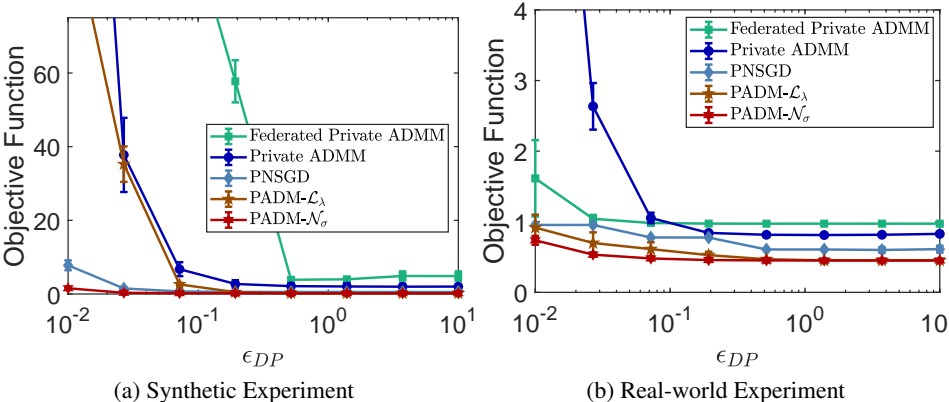

(a) Synthetic Experiment        (b) Real-world Experiment

Figure 2: Final objective function values (mean $\pm$ STD) of federated private ADMM, private ADMM, PNSGD, PADM-$\mathcal{L}_\lambda$ (ours), and PADM-$\mathcal{N}_\sigma$ (ours).

We compare the performance of PADM-$\mathcal{L}_\lambda$ and PADM-$\mathcal{N}_\sigma$ with federated private ADMM (Cyffers et al., 2023), private ADMM (Chan et al., 2024) and PNSGD (Feldman et al., 2018), using the average and standard deviation of the final objective function values with the same regularization parameters. For competitors, the noise added during the iterations is sampled from a Gaussian distribution, as specified in the original work. As shown in Figure 2, PADM-$\mathcal{N}_\sigma$ outperforms the three competitors across all the cases in both synthetic and real-world experiments. PADM-$\mathcal{L}_\lambda$ outperforms the three competitors in the real-world experiment while remaining competitive in the synthetic experiment. In contrast, the outputs of the two ADMM methods are even infeasible, and PNSGD directly drops the public variable $y$ and cannot solve the private-public joint optimization problem. Additional experimental results are provided in Appendix A.14.

## 5 Conclusions and future works

We establish a novel privacy amplification algorithm with the Projected Alternating Direction Method (PADM). It mainly addresses the limitations of existing ADMM based methods that they may have a large feasibility gap (primal residual) during or at the end of a learning process. By achieving exact feasibility in each iteration, PADM ensures meaningful outputs and allows each private user to supervise the objective value. The private version of PADM allows both Gaussian (PADM-$\mathcal{N}_\sigma$) and Laplace (PADM-$\mathcal{L}_\lambda$) noises to achieve the same order of privacy-utility tradeoff as that of the existing ADMM methods. In short, PADM solves more general privacy amplification problems, has more favorable properties, but requires no more conditions than the existing ADMM methods.

Experimental results on the synthetic and real-world data demonstrate that both PADM-$\mathcal{N}_\sigma$ and PADM-$\mathcal{L}_\lambda$ consistently outperform the private ADMM across different levels of privacy budgets,

and achieve lower objective function values under exact feasibility. This advantage is especially significant when the privacy budget is low. Both PADM methods also achieve better performance to PNSGD under moderate privacy budgets. The experiment results validate the effectiveness of PADM-$\mathcal{N}_\sigma$ and PADM-$\mathcal{L}_\lambda$ in balancing privacy and utility. Future works may lie in extending private PADM to federated learning scenarios or other types of feasibility constraints.

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

# A APPENDIX

## A.1 PROOF OF THEOREM 3.1

**Definition 8** (Subgradient, (Rockafellar & Wets, 2009)). Let $F : \mathbb{R}^p \to \mathbb{R}$ be a convex function. A vector $\boldsymbol{s} \in \mathbb{R}^p$ is called a subgradient of $F$ at $\boldsymbol{x} \in \mathbb{R}^p$ if for all $\boldsymbol{t} \in \mathbb{R}^p$,

$$F(\boldsymbol{t}) - F(\boldsymbol{x}) \geqslant \boldsymbol{s}^\top (\boldsymbol{t} - \boldsymbol{x}). \tag{22}$$

The set of all subgradients at $\boldsymbol{x}$ is denoted by $\partial F(\boldsymbol{x})$. If $F$ is differentiable at $\boldsymbol{x}$, then $\partial F(\boldsymbol{x})$ reduces to the gradient $\nabla F(\boldsymbol{x})$.

**Lemma 1** (Fermat's rule,(Rockafellar & Wets, 2009)). *The point $\boldsymbol{x}_* \in \mathbb{R}^p$ is a minimizer of $F$ if and only if $\boldsymbol{0}_p \in \partial F(\boldsymbol{x}_*)$.*

*Proof of Theorem 3.1.* If (15) holds, then from Fermat's rule,

$$\boldsymbol{0}_p \in \partial F(\hat{\boldsymbol{x}}_*) + \frac{\beta}{2} \boldsymbol{A}^\top (\boldsymbol{A}\hat{\boldsymbol{x}}_* + \boldsymbol{B}\hat{\boldsymbol{y}}_* - \boldsymbol{c}),$$

$$\boldsymbol{0}_l \in \partial g(\hat{\boldsymbol{y}}_*) + \frac{\beta}{2} \boldsymbol{B}^\top (\boldsymbol{A}\hat{\boldsymbol{x}}_* + \boldsymbol{B}\hat{\boldsymbol{y}}_* - \boldsymbol{c}). \tag{23}$$

It follows from (23) and the equation $\boldsymbol{u}_* = \frac{\boldsymbol{A}\hat{\boldsymbol{x}}_* - \boldsymbol{B}\hat{\boldsymbol{y}}_* + \boldsymbol{c}}{2}$ that

$$\boldsymbol{0}_p \in \partial F(\hat{\boldsymbol{x}}_*) + \beta \boldsymbol{A}^\top (\boldsymbol{A}\hat{\boldsymbol{x}}_* - \boldsymbol{u}_*),$$

$$\boldsymbol{0}_l \in \partial g(\hat{\boldsymbol{y}}_*) + \beta \boldsymbol{B}^\top (\boldsymbol{B}\hat{\boldsymbol{y}}_* - \boldsymbol{c}^* + \boldsymbol{u}_*). \tag{24}$$

Using the definitions of $\mathcal{P}_{\beta,F,\boldsymbol{A}}(\boldsymbol{u}_*)$ and $\mathcal{P}_{\beta,g,\boldsymbol{B}}(\boldsymbol{c} - \boldsymbol{u}_*)$ in (12), we have $\begin{bmatrix} \boldsymbol{A}\hat{\boldsymbol{x}}_* \\ \boldsymbol{B}\hat{\boldsymbol{y}}_* \end{bmatrix} = \begin{bmatrix} \boldsymbol{A} \circ \mathcal{P}_{\beta,F,\boldsymbol{A}}(\boldsymbol{u}_*) \\ \boldsymbol{B} \circ \mathcal{P}_{\beta,g,\boldsymbol{B}}(\boldsymbol{c} - \boldsymbol{u}_*) \end{bmatrix}$. Since $\boldsymbol{u}_* = \frac{\boldsymbol{A}\hat{\boldsymbol{x}}_* - \boldsymbol{B}\hat{\boldsymbol{y}}_* + \boldsymbol{c}}{2}$, together with (9) we have

$$\begin{bmatrix} \boldsymbol{u}_* \\ \boldsymbol{c} - \boldsymbol{u}_* \end{bmatrix} = \Pi_{\mathcal{W}} \left( \begin{bmatrix} \boldsymbol{A}\hat{\boldsymbol{x}}_* \\ \boldsymbol{B}\hat{\boldsymbol{y}}_* \end{bmatrix} \right),$$

$$\boldsymbol{u}_* = \boldsymbol{H} \cdot \left( \Pi_{\mathcal{W}} \left( \begin{bmatrix} \boldsymbol{A} \circ \mathcal{P}_{\beta,F,\boldsymbol{A}}(\boldsymbol{u}_*) \\ \boldsymbol{B} \circ \mathcal{P}_{\beta,g,\boldsymbol{B}}(\boldsymbol{c} - \boldsymbol{u}_*) \end{bmatrix} \right) \right), \tag{25}$$

which implies that $\boldsymbol{u}_*$ is a fixed point of $\mathcal{T}_F$.

Conversely, if $\boldsymbol{u}_*$ is a fixed point of $\mathcal{T}_F$, then the equation (25) holds, which implies that there exists $(\hat{\boldsymbol{x}}_*, \hat{\boldsymbol{y}}_*)$ such that $\hat{\boldsymbol{x}}_* = \mathcal{P}_{\beta,F,\boldsymbol{A}}(\boldsymbol{u}_*)$, $\hat{\boldsymbol{y}}_* = \mathcal{P}_{\beta,g,\boldsymbol{B}}(\boldsymbol{c} - \boldsymbol{u}_*)$ and $\boldsymbol{u}_* = \frac{\boldsymbol{A}\hat{\boldsymbol{x}}_* - \boldsymbol{B}\hat{\boldsymbol{y}}_* + \boldsymbol{c}}{2}$ hold. Then, using the definition of $\mathcal{P}_{\beta,F,\boldsymbol{A}}(\boldsymbol{u}_*)$ and $\mathcal{P}_{\beta,g,\boldsymbol{B}}(\boldsymbol{c} - \boldsymbol{u}_*)$, we know that (24) holds. Applying (24) with the equation $\boldsymbol{u}_* = \frac{\boldsymbol{A}\hat{\boldsymbol{x}}_* - \boldsymbol{B}\hat{\boldsymbol{y}}_* + \boldsymbol{c}}{2}$, we can conclude that (23) holds and therefore (15) holds, which completes the proof. $\qquad \square$

## A.2 FIRMLY NON-EXPANSIVENESS OF OPERATOR

**Proposition A.1.** *Let the operator $\mathcal{P}_{\beta,F,\boldsymbol{A}}$ be defined as in (12). Then the operator $\boldsymbol{A} \circ \mathcal{P}_{\beta,F,\boldsymbol{A}}$ is firmly non-expansive.*

*Proof.* For $\boldsymbol{u}, \boldsymbol{v} \in \mathbb{R}^m$, let $\boldsymbol{p} = \mathcal{P}_{\beta,F,\boldsymbol{A}}(\boldsymbol{u})$ and $\boldsymbol{q} = \mathcal{P}_{\beta,F,\boldsymbol{A}}(\boldsymbol{v})$. Since $F$ is convex and proper, then by the definition of $\mathcal{P}_{\beta,F,\boldsymbol{A}}$, we know that $\beta \boldsymbol{A}^\top (\boldsymbol{u} - \boldsymbol{A}\boldsymbol{p}) \in \partial F(\boldsymbol{p})$ and $\beta \boldsymbol{A}^\top (\boldsymbol{v} - \boldsymbol{A}\boldsymbol{q}) \in \partial F(\boldsymbol{q})$. Together with Definition 8,

$$F(\boldsymbol{q}) - F(\boldsymbol{p}) \geqslant \langle \beta \boldsymbol{A}^\top (\boldsymbol{u} - \boldsymbol{A}\boldsymbol{p}), \boldsymbol{q} - \boldsymbol{p} \rangle,$$

$$F(\boldsymbol{p}) - F(\boldsymbol{q}) \geqslant \langle \beta \boldsymbol{A}^\top (\boldsymbol{v} - \boldsymbol{A}\boldsymbol{q}), \boldsymbol{p} - \boldsymbol{q} \rangle.$$

Summing up both sides of the above inequalities yields

$$\langle \boldsymbol{v} - \boldsymbol{u}, \boldsymbol{A}(\boldsymbol{q} - \boldsymbol{p}) \rangle \geqslant \langle \boldsymbol{A}(\boldsymbol{q} - \boldsymbol{p}), \boldsymbol{A}(\boldsymbol{q} - \boldsymbol{p}) \rangle = \|\boldsymbol{A}(\boldsymbol{q} - \boldsymbol{p})\|_2^2, \tag{26}$$

which indicates that the operator $\boldsymbol{A} \circ \mathcal{P}_{\beta,F,\boldsymbol{A}}$ is firmly non-expansive. $\qquad \square$

### A.3 PROOF OF THEOREM 3.2

**Lemma 2** ((Combettes & Yamada, 2015), Proposition 2.4)**.** *Let the operators $\mathcal{T}_1 : \mathbb{R}^m \to \mathbb{R}^m$ and $\mathcal{T}_2 : \mathbb{R}^m \to \mathbb{R}^m$ be $\alpha_2$-averaged non-expansive, where $\alpha_1, \alpha_2 \in (0, 1)$. Then the operator $\mathcal{T}_1 \circ \mathcal{T}_2$ is $\frac{\alpha_1 + \alpha_2 - 2\alpha_1\alpha_2}{1 - \alpha_1\alpha_2}$-averaged non-expansive.*

**Lemma 3** ((Combettes & Yamada, 2015), Remark 2.7)**.** *The operator $\mathcal{T} : \mathbb{R}^m \to \mathbb{R}^m$ is firmly non-expansive if and only if it is $\frac{1}{2}$-averaged non-expansive.*

**Lemma 4** ((Combettes & Yamada, 2015), Proposition 2.1)**.** *The operator $\mathcal{T} : \mathbb{R}^m \to \mathbb{R}^m$ is $\alpha$-averaged nonexpansive $(0 < \alpha < 1)$ if and only if*

$$\|\mathcal{T}(\boldsymbol{x}) - \mathcal{T}(\boldsymbol{y})\|_2^2 \leqslant \|\boldsymbol{x} - \boldsymbol{y}\|_2^2 - \frac{1 - \alpha}{\alpha}\|(\mathcal{I} - \mathcal{T})\boldsymbol{x} - (\mathcal{I} - \mathcal{T})\boldsymbol{y}\|_2^2.$$

*Proof of Theorem 3.2.* From Proposition A.1, we know that the operators $\boldsymbol{A} \circ \mathcal{P}_{\beta, F, \boldsymbol{A}}$ and $\boldsymbol{B} \circ \mathcal{P}_{\beta, g, \boldsymbol{B}}$ are both firmly non-expansive. For $\boldsymbol{u}, \boldsymbol{v} \in \mathbb{R}^m$, define $\Omega\left(\begin{bmatrix} \boldsymbol{u} \\ \boldsymbol{v} \end{bmatrix}\right) := \begin{bmatrix} \boldsymbol{A} \circ \mathcal{P}_{\beta, F, \boldsymbol{A}}(\boldsymbol{u}) \\ \boldsymbol{B} \circ \mathcal{P}_{\beta, g, \boldsymbol{B}}(\boldsymbol{v}) \end{bmatrix}$. Then for $\boldsymbol{u}, \boldsymbol{v}, \boldsymbol{u}', \boldsymbol{v}' \in \mathbb{R}^m$,

$$
\begin{aligned}
&\left\langle \Omega\left(\begin{bmatrix} \boldsymbol{u} \\ \boldsymbol{v} \end{bmatrix}\right) - \Omega\left(\begin{bmatrix} \boldsymbol{u}' \\ \boldsymbol{v}' \end{bmatrix}\right), \begin{bmatrix} \boldsymbol{u} - \boldsymbol{u}' \\ \boldsymbol{v} - \boldsymbol{v}' \end{bmatrix} \right\rangle \\
=& \left\langle \begin{bmatrix} \boldsymbol{A} \circ \mathcal{P}_{\beta, F, \boldsymbol{A}}(\boldsymbol{u}) - \boldsymbol{A} \circ \mathcal{P}_{\beta, F, \boldsymbol{A}}(\boldsymbol{u}') \\ \boldsymbol{B} \circ \mathcal{P}_{\beta, g, \boldsymbol{B}}(\boldsymbol{v}) - \boldsymbol{B} \circ \mathcal{P}_{\beta, g, \boldsymbol{B}}(\boldsymbol{v}') \end{bmatrix}, \begin{bmatrix} \boldsymbol{u} - \boldsymbol{u}' \\ \boldsymbol{v} - \boldsymbol{v}' \end{bmatrix} \right\rangle \\
\geqslant& \|\boldsymbol{A} \circ \mathcal{P}_{\beta, F, \boldsymbol{A}}(\boldsymbol{u}) - \boldsymbol{A} \circ \mathcal{P}_{\beta, F, \boldsymbol{A}}(\boldsymbol{u}')\|_2^2 + \|\boldsymbol{B} \circ \mathcal{P}_{\beta, g, \boldsymbol{B}}(\boldsymbol{v}) - \boldsymbol{B} \circ \mathcal{P}_{\beta, g, \boldsymbol{B}}(\boldsymbol{v}')\|_2^2 \\
=& \left\| \begin{bmatrix} \boldsymbol{A} \circ \mathcal{P}_{\beta, F, \boldsymbol{A}}(\boldsymbol{u}) - \boldsymbol{A} \circ \mathcal{P}_{\beta, F, \boldsymbol{A}}(\boldsymbol{u}') \\ \boldsymbol{B} \circ \mathcal{P}_{\beta, g, \boldsymbol{B}}(\boldsymbol{v}) - \boldsymbol{B} \circ \mathcal{P}_{\beta, g, \boldsymbol{B}}(\boldsymbol{v}') \end{bmatrix} \right\|_2^2 \\
=& \left\| \Omega\left(\begin{bmatrix} \boldsymbol{u} \\ \boldsymbol{v} \end{bmatrix}\right) - \Omega\left(\begin{bmatrix} \boldsymbol{u}' \\ \boldsymbol{v}' \end{bmatrix}\right) \right\|_2^2,
\end{aligned}
$$

which implies that the operator $\Omega$ is also firmly non-expansive. Since the projection operator $\Pi_{\mathcal{W}}$ is firmly non-expansive, then from Lemma 2 and Lemma 3, the operator $\Pi_{\mathcal{W}} \circ \Omega$ is $\frac{2}{3}$-averaged non-expansive. Let $\begin{bmatrix} \hat{\boldsymbol{u}} \\ \boldsymbol{c} - \hat{\boldsymbol{u}} \end{bmatrix} := \Pi_{\mathcal{W}} \circ \Omega\left(\begin{bmatrix} \boldsymbol{u} \\ \boldsymbol{v} \end{bmatrix}\right)$ and $\begin{bmatrix} \hat{\boldsymbol{u}}' \\ \boldsymbol{c} - \hat{\boldsymbol{u}}' \end{bmatrix} := \Pi_{\mathcal{W}} \circ \Omega\left(\begin{bmatrix} \boldsymbol{u}' \\ \boldsymbol{v}' \end{bmatrix}\right)$. Then we know from Lemma 4 that

$$\left\| \begin{bmatrix} \hat{\boldsymbol{u}} - \hat{\boldsymbol{u}}' \\ \hat{\boldsymbol{u}}' - \hat{\boldsymbol{u}} \end{bmatrix} \right\|_2^2 \leqslant \left\| \begin{bmatrix} \boldsymbol{u} - \boldsymbol{u}' \\ \boldsymbol{v} - \boldsymbol{v}' \end{bmatrix} \right\|_2^2 - \frac{1}{2}\left\| \begin{bmatrix} \hat{\boldsymbol{u}}' - \hat{\boldsymbol{u}} + \boldsymbol{u} - \boldsymbol{u}' \\ \hat{\boldsymbol{u}} - \hat{\boldsymbol{u}}' + \boldsymbol{v} - \boldsymbol{v}' \end{bmatrix} \right\|_2^2. \tag{27}$$

Let $\boldsymbol{v} := \boldsymbol{c} - \boldsymbol{u}$ and $\boldsymbol{v}' := \boldsymbol{c} - \boldsymbol{u}'$. Then it follows from (27) that

$$2\|\hat{\boldsymbol{u}} - \hat{\boldsymbol{u}}'\|_2^2 \leqslant 2\|\boldsymbol{u} - \boldsymbol{u}'\|_2^2 - \|\hat{\boldsymbol{u}}' - \hat{\boldsymbol{u}} + \boldsymbol{u} - \boldsymbol{u}'\|_2^2,$$

which implies that the operator $\mathcal{T}_F$ is $\frac{2}{3}$-averaged non-expansive. This completes the proof. □

### A.4 PROOF OF THEOREM 3.3

**Lemma 5** (Krasnoselskii-Mann (KM) Theorem)**.** *Let $\mathcal{T}_F : \mathbb{R}^m \to \mathbb{R}^m$ be a $\alpha$-averaged non-expansive operator for some $\alpha > 0$. For an initial point $\boldsymbol{u}_{(0)} \in \mathbb{R}^m$, let $\boldsymbol{u}_{(k)} = \mathcal{T}_F(\boldsymbol{u}_{(k-1)})$, $k \in \mathbb{N}$. If the fixed-point set of $\mathcal{T}_F$ is nonempty, then the sequence $\{\boldsymbol{u}_{(k)}\}_{k \in \mathbb{N}}$ converges to a fixed point of $\mathcal{T}_F$.*

*Proof of Theorem 3.3.* By the general assumption that at least one minimum point of (15) exists, Lemma 3.1 indicates that there exists a fixed point of $\mathcal{T}_F$. Then from Theorem 3.2 and the KM theorem, the sequence $\{\boldsymbol{u}_{(k)}\}_{k \in \mathbb{N}}$ generated by (14) converges to a fixed point of $\mathcal{T}_F$. □

### A.5 PROOF OF THEOREM 3.4

*Proof.* The update rule defined in (16) can be reformulated as:

$$\tilde{\boldsymbol{u}}_{(k)} = \mathcal{T}_F(\tilde{\boldsymbol{u}}_{(k-1)}) + \boldsymbol{e}_{(k)} + \frac{\boldsymbol{z}_{(k)}}{2}, \ k \in [n], \tag{28}$$

where $\boldsymbol{e}_{(k)} := \mathcal{T}_{f_k}(\tilde{\boldsymbol{u}}_{(k-1)}) - \mathcal{T}_F(\tilde{\boldsymbol{u}}_{(k-1)})$ represents the estimation error and $\boldsymbol{z}_{(k)}$ represents the generated noise with mean $\mathbb{E}(\boldsymbol{z}_{(k)}) = \boldsymbol{0}_m$ and variance $\mathbb{E}(\|\boldsymbol{z}_{(k)}\|_2^2) = \delta^2$. From Condition 1, we know that $\mathbb{E}(\|\boldsymbol{e}_{(k)}\|_2^2) \leqslant \mu^2$. Since $\mathbb{E}(\|\boldsymbol{e}_{(k)}\|_2)$ is an expectation on a probability measure (which is a finite measure), it is dominated by $\mathbb{E}(\|\boldsymbol{e}_{(k)}\|_2^2)$ and thus also finite. Because $\boldsymbol{e}_{(k)}$ is independent of $\boldsymbol{z}_{(k)}$, the squared-$\ell_2$-norm of the total error term

$$\mathbb{E}(\|\boldsymbol{e}_{(k)} + \frac{\boldsymbol{z}_{(k)}}{2}\|_2^2)$$

$$=\mathbb{E}(\|\boldsymbol{e}_{(k)}\|_2^2) + \mathbb{E}(\|\frac{\boldsymbol{z}_{(k)}}{2}\|_2^2) + 2\mathbb{E}(\langle \boldsymbol{e}_{(k)}, \frac{\boldsymbol{z}_{(k)}}{2}\rangle)$$

$$=\mathbb{E}(\|\boldsymbol{e}_{(k)}\|_2^2) + \mathbb{E}(\|\frac{\boldsymbol{z}_{(k)}}{2}\|_2^2) + 2\langle \mathbb{E}(\boldsymbol{e}_{(k)}), \mathbb{E}(\frac{\boldsymbol{z}_{(k)}}{2})\rangle$$

$$=\mathbb{E}(\|\boldsymbol{e}_{(k)}\|_2^2) + \mathbb{E}(\|\frac{\boldsymbol{z}_{(k)}}{2}\|_2^2)$$

$$\leqslant \mu^2 + \frac{\delta^2}{4} =: \varsigma^2 \tag{29}$$

is bounded. According to the assumption that the operator $\mathcal{T}_F$ is $\tau$-contractive,

$$\|\mathcal{T}_F(\tilde{\boldsymbol{u}}_{(k)}) - \boldsymbol{u}_*\|_2^2 \leqslant \tau^2\|\tilde{\boldsymbol{u}}_{(k)} - \boldsymbol{u}_*\|_2^2 \tag{30}$$

holds for all $k$. Combining (29) and (30), we can extend the methodology for the one-error-term case in ((Combettes & Pesquet, 2019), Theorem 2.3) to the two-error-term case in (28). Let $\boldsymbol{\phi}_{(k)} := \boldsymbol{e}_{(k)} + \frac{\boldsymbol{z}_{(k)}}{2}$. Then (28) indicates that for any $k \in [n]$,

$$\|\tilde{\boldsymbol{u}}_{(k)} - \boldsymbol{u}_*\|_2^2 = \|\mathcal{T}_F(\tilde{\boldsymbol{u}}_{(k-1)}) - \boldsymbol{u}_*\|_2^2 + 2\langle \mathcal{T}_F(\tilde{\boldsymbol{u}}_{(k-1)}) - \boldsymbol{u}_*, \boldsymbol{\phi}_{(k)}\rangle + \|\boldsymbol{\phi}_{(k)}\|_2^2. \tag{31}$$

Combining (29), (30) and (31),

$$\mathbb{E}(\|\tilde{\boldsymbol{u}}_{(k)} - \boldsymbol{u}_*\|_2^2|\Gamma_{k-1})$$

$$\leqslant \mathbb{E}(\|\mathcal{T}_F(\tilde{\boldsymbol{u}}_{(k-1)}) - \boldsymbol{u}_*\|_2^2|\Gamma_{k-1}) + 2\sqrt{\mathbb{E}(\|\mathcal{T}_F(\tilde{\boldsymbol{u}}_{(k-1)}) - \boldsymbol{u}_*\|_2^2|\Gamma_{k-1})} \cdot \sqrt{\mathbb{E}(\|\boldsymbol{\phi}_{(k)}\|_2^2|\Gamma_{k-1})}$$

$$+ \mathbb{E}(\|\boldsymbol{\phi}_{(k)}\|_2^2|\Gamma_{k-1})$$

$$\leqslant \tau^2\|\tilde{\boldsymbol{u}}_{(k-1)} - \boldsymbol{u}_*\|_2^2 + 2\tau\|\tilde{\boldsymbol{u}}_{(k-1)} - \boldsymbol{u}_*\|_2\sqrt{\mathbb{E}(\|\boldsymbol{\phi}_{(k)}\|_2^2|\Gamma_{k-1})} + \mathbb{E}(\|\boldsymbol{\phi}_{(k)}\|_2^2|\Gamma_{k-1})$$

$$\leqslant \tau^2\|\tilde{\boldsymbol{u}}_{(k-1)} - \boldsymbol{u}_*\|_2^2 + \tau(\|\tilde{\boldsymbol{u}}_{(k-1)} - \boldsymbol{u}_*\|_2^2 + 1)\sqrt{\mathbb{E}(\|\boldsymbol{\phi}_{(k)}\|_2^2|\Gamma_{k-1})} + \mathbb{E}(\|\boldsymbol{\phi}_{(k)}\|_2^2|\Gamma_{k-1})$$

$$\leqslant \chi\|\tilde{\boldsymbol{u}}_{(k-1)} - \boldsymbol{u}_*\|_2^2 + \tau\varsigma + \varsigma^2, \tag{32}$$

where $\chi := \tau^2 + \tau\varsigma$. Taking expectations conditioned on $\Gamma_{k-2}$ in both sides of (32) yields

$$\mathbb{E}\big(\mathbb{E}(\|\tilde{\boldsymbol{u}}_{(k)} - \boldsymbol{u}_*\|_2^2|\Gamma_{k-1})|\Gamma_{k-2}\big) \leqslant \chi\mathbb{E}(\|\tilde{\boldsymbol{u}}_{(k-1)} - \boldsymbol{u}_*\|_2^2|\Gamma_{k-2}) + \tau\varsigma + \varsigma^2. \tag{33}$$

Since $\Gamma_{k-2} \subseteq \Gamma_{k-1}$, $\mathbb{E}\big(\mathbb{E}(\|\tilde{\boldsymbol{u}}_{(k)} - \boldsymbol{u}_*\|_2^2|\Gamma_{k-1})|\Gamma_{k-2}\big) = \mathbb{E}(\|\tilde{\boldsymbol{u}}_{(k)} - \boldsymbol{u}_*\|_2^2|\Gamma_{k-2})$. Then (33) is simplified as

$$\mathbb{E}(\|\tilde{\boldsymbol{u}}_{(k)} - \boldsymbol{u}_*\|_2^2|\Gamma_{k-2}) \leqslant \chi\mathbb{E}(\|\tilde{\boldsymbol{u}}_{(k-1)} - \boldsymbol{u}_*\|_2^2|\Gamma_{k-2}) + \tau\varsigma + \varsigma^2. \tag{34}$$

Similarly, taking expectations conditioned on $\Gamma_0$ in both sides of (32) yields

$$\mathbb{E}(\|\tilde{\boldsymbol{u}}_{(k)} - \boldsymbol{u}_*\|_2^2|\Gamma_0) \leqslant \chi\mathbb{E}(\|\tilde{\boldsymbol{u}}_{(k-1)} - \boldsymbol{u}_*\|_2^2|\Gamma_0) + \tau\varsigma + \varsigma^2.$$

The assumption $\delta < 2\sqrt{\frac{(1-\tau^2)^2}{\tau^2} - \mu^2}$ indicates that $0 < \chi < 1$. Therefore,

$$\mathbb{E}(\|\tilde{\boldsymbol{u}}_{(k)} - \boldsymbol{u}_*\|_2^2|\Gamma_0)$$

$$\leqslant \chi\mathbb{E}(\|\tilde{\boldsymbol{u}}_{(k-1)} - \boldsymbol{u}_*\|_2^2|\Gamma_0) + \tau\varsigma + \varsigma^2$$

$$\leqslant \chi^2 \mathbb{E}(\|\tilde{\boldsymbol{u}}_{(k-2)} - \boldsymbol{u}_*\|_2^2 | \Gamma_0) + (1 + \chi)(\tau\varsigma + \varsigma^2)$$

$$\leqslant \ldots$$

$$\leqslant \chi^k \|\tilde{\boldsymbol{u}}_{(0)} - \boldsymbol{u}_*\|_2^2 + (\tau\varsigma + \varsigma^2) \sum_{j=0}^{k-1} \chi^j$$

$$= \chi^k \|\tilde{\boldsymbol{u}}_{(0)} - \boldsymbol{u}_*\|_2^2 + (\tau\varsigma + \varsigma^2) \frac{1 - \chi^k}{1 - \chi}. \tag{35}$$

Inserting $\chi = \tau^2 + \tau\varsigma$ and $\varsigma = \sqrt{\frac{\delta^2}{4} + \mu^2}$ into (35) yields

$$\mathbb{E}(\|\tilde{\boldsymbol{u}}_{(k)} - \boldsymbol{u}_*\|_2^2 | \Gamma_0)$$

$$\leqslant \left(\tau^2 + \tau\sqrt{\frac{\delta^2}{4} + \mu^2}\right)^k \|\tilde{\boldsymbol{u}}_{(0)} - \boldsymbol{u}_*\|_2^2$$

$$+ \frac{1 - \left(\tau^2 + \tau\sqrt{\frac{\delta^2}{4} + \mu^2}\right)^k}{1 - \left(\tau^2 + \tau\sqrt{\frac{\delta^2}{4} + \mu^2}\right)} \left(\frac{\delta^2}{4} + \mu^2 + \tau\sqrt{\frac{\delta^2}{4} + \mu^2}\right). \tag{36}$$

Then we have

$$\mathbb{E}(\|\tilde{\boldsymbol{x}}_{(k)} - \boldsymbol{x}_*\|_2^2 | \Gamma_0)$$

$$= \mathbb{E}(\|\boldsymbol{V}(\tilde{\boldsymbol{u}}_{(k)} - \boldsymbol{u}_*)\|_2^2 | \Gamma_0)$$

$$\leqslant \|\boldsymbol{V}\|_2^2 \mathbb{E}(\|\tilde{\boldsymbol{u}}_{(k)} - \boldsymbol{u}_*\|_2^2 | \Gamma_0)$$

$$\leqslant \left(\tau^2 + \tau\sqrt{\frac{\delta^2}{4} + \mu^2}\right)^k \|\boldsymbol{V}\|_2^2 \|\tilde{\boldsymbol{u}}_{(0)} - \boldsymbol{u}_*\|_2^2$$

$$+ \frac{\|\boldsymbol{V}\|_2^2 \left[1 - \left(\tau^2 + \tau\sqrt{\frac{\delta^2}{4} + \mu^2}\right)^k\right]}{1 - \left(\tau^2 + \tau\sqrt{\frac{\delta^2}{4} + \mu^2}\right)} \left(\frac{\delta^2}{4} + \mu^2 + \tau\sqrt{\frac{\delta^2}{4} + \mu^2}\right)$$

$$= \left(\tau^2 + \tau\sqrt{\frac{\delta^2}{4} + \mu^2}\right)^k \|\boldsymbol{V}\|_2^2 \cdot \|\boldsymbol{A}(\tilde{\boldsymbol{x}}_{(0)} - \boldsymbol{x}_*)\|_2^2$$

$$+ \frac{\|\boldsymbol{V}\|_2^2 \left[1 - \left(\tau^2 + \tau\sqrt{\frac{\delta^2}{4} + \mu^2}\right)^k\right]}{1 - \left(\tau^2 + \tau\sqrt{\frac{\delta^2}{4} + \mu^2}\right)} \left(\frac{\delta^2}{4} + \mu^2 + \tau\sqrt{\frac{\delta^2}{4} + \mu^2}\right)$$

$$\leqslant \left(\tau^2 + \tau\sqrt{\frac{\delta^2}{4} + \mu^2}\right)^k \|\boldsymbol{V}\|_2^2 \|\boldsymbol{A}\|_2^2 \cdot \|\tilde{\boldsymbol{x}}_{(0)} - \boldsymbol{x}_*\|_2^2$$

$$+ \frac{\|\boldsymbol{V}\|_2^2 \left[1 - \left(\tau^2 + \tau\sqrt{\frac{\delta^2}{4} + \mu^2}\right)^k\right]}{1 - \left(\tau^2 + \tau\sqrt{\frac{\delta^2}{4} + \mu^2}\right)} \left(\frac{\delta^2}{4} + \mu^2 + \tau\sqrt{\frac{\delta^2}{4} + \mu^2}\right),$$

which completes the proof. $\qquad \square$

### A.6 PROOF OF PROPOSITION 3.1

*Proof.* For any $\boldsymbol{u} \in \mathbb{R}^m$ (or $\boldsymbol{u}' \in \mathbb{R}^m$), define $\hat{\boldsymbol{x}} := \mathcal{P}_{\beta,F,\boldsymbol{A}}(\boldsymbol{u})$ and $\hat{\boldsymbol{y}} := \mathcal{P}_{\beta,g,\boldsymbol{B}}(\boldsymbol{c} - \boldsymbol{u})$ (or $\hat{\boldsymbol{x}}' := \mathcal{P}_{\beta,F,\boldsymbol{A}}(\boldsymbol{u}')$ and $\hat{\boldsymbol{y}}' := \mathcal{P}_{\beta,g,\boldsymbol{B}}(\boldsymbol{c} - \boldsymbol{u}'))$. Then $\mathcal{T}_F(\boldsymbol{u}) = \frac{\boldsymbol{A}\hat{\boldsymbol{x}} - \boldsymbol{B}\hat{\boldsymbol{y}} + \boldsymbol{c}}{2}$ (or $\mathcal{T}_F(\boldsymbol{u}') = \frac{\boldsymbol{A}\hat{\boldsymbol{x}}' - \boldsymbol{B}\hat{\boldsymbol{y}}' + \boldsymbol{c}}{2}$).

From the definition of $\mathcal{P}_{\beta,F,\boldsymbol{A}}$,

$$\beta\boldsymbol{A}^\top\boldsymbol{u} - \beta\boldsymbol{A}^\top\boldsymbol{A}\hat{\boldsymbol{x}} \in \partial F(\hat{\boldsymbol{x}}), \tag{37}$$

$$\beta \boldsymbol{A}^\top \boldsymbol{u}' - \beta \boldsymbol{A}^\top \boldsymbol{A}\hat{\boldsymbol{x}}' \in \partial F(\hat{\boldsymbol{x}}'). \tag{38}$$

Since $F$ is $\boldsymbol{A}$-induced $\varrho_F$-strongly convex, combining (37) and (38) yields

$$(\beta(\boldsymbol{u} - \boldsymbol{u}') - \beta \boldsymbol{A}(\hat{\boldsymbol{x}} - \hat{\boldsymbol{x}}'))^\top \boldsymbol{A}(\hat{\boldsymbol{x}} - \hat{\boldsymbol{x}}') \geqslant \varrho_F \|\boldsymbol{A}(\hat{\boldsymbol{x}} - \hat{\boldsymbol{x}}')\|_2^2,$$
$$\beta(\boldsymbol{u} - \boldsymbol{u}')^\top \boldsymbol{A}(\hat{\boldsymbol{x}} - \hat{\boldsymbol{x}}') \geqslant (\beta + \varrho_F)\|\boldsymbol{A}(\hat{\boldsymbol{x}} - \hat{\boldsymbol{x}}')\|_2^2. \tag{39}$$

From Cauchy's inequality,

$$\beta(\boldsymbol{u} - \boldsymbol{u}')^\top \boldsymbol{A}(\hat{\boldsymbol{x}} - \hat{\boldsymbol{x}}') \leqslant \|\beta(\boldsymbol{u} - \boldsymbol{u}')\|_2 \|\boldsymbol{A}(\hat{\boldsymbol{x}} - \hat{\boldsymbol{x}}')\|_2. \tag{40}$$

Combining (39) and (40), we have

$$\|\boldsymbol{A}\hat{\boldsymbol{x}} - \boldsymbol{A}\hat{\boldsymbol{x}}'\|_2 \leqslant \frac{\beta}{\beta + \varrho_F}\|\boldsymbol{u} - \boldsymbol{u}'\|_2 \quad \text{or} \quad \|\boldsymbol{A}\hat{\boldsymbol{x}} - \boldsymbol{A}\hat{\boldsymbol{x}}'\|_2 = 0. \tag{41}$$

It is easy to see that (41) is equivalent to the former case

$$\|\boldsymbol{A}\hat{\boldsymbol{x}} - \boldsymbol{A}\hat{\boldsymbol{x}}'\|_2 \leqslant \frac{\beta}{\beta + \varrho_F}\|\boldsymbol{u} - \boldsymbol{u}'\|_2. \tag{42}$$

On the other hand, Appendix A.2 indicates that the operator $\boldsymbol{B} \circ \mathcal{P}_{\beta,g,\boldsymbol{B}}$ is firmly non-expansive, which implies that $\boldsymbol{B} \circ \mathcal{P}_{\beta,g,\boldsymbol{B}}$ is non-expansive. Therefore,

$$\|\boldsymbol{B}\hat{\boldsymbol{y}} - \boldsymbol{B}\hat{\boldsymbol{y}}'\|_2 \leqslant \|\boldsymbol{u} - \boldsymbol{u}'\|_2. \tag{43}$$

Combining (42) and (43), we conclude that

$$\|\mathcal{T}_F(\boldsymbol{u}) - \mathcal{T}_F(\boldsymbol{u}')\|_2 = \left\|\frac{\boldsymbol{A}\hat{\boldsymbol{x}} - \boldsymbol{A}\hat{\boldsymbol{x}}' - (\boldsymbol{B}\hat{\boldsymbol{y}} - \boldsymbol{B}\hat{\boldsymbol{y}}')}{2}\right\|_2 \leqslant \frac{1}{2} \cdot (\frac{\beta}{\beta + \varrho_F} + 1)\|\boldsymbol{u} - \boldsymbol{u}'\|_2.$$

The $\frac{1}{2} \cdot (1 + \frac{\beta}{\beta + \varrho_g})$-contractive and $\frac{1}{2} \cdot (\frac{\beta}{\beta + \varrho_F} + \frac{\beta}{\beta + \varrho_g})$-contractive cases can be deduced in a similar way.

We further verify that the induced strong convexity defined in Definition 7 is weaker than the classic strong convexity defined in (Rockafellar & Wets, 2009). Recall the latter as follows: there exists a constant $\iota_F > 0$ such that for any $\boldsymbol{x}, \boldsymbol{x}' \in \mathbb{R}^p$, $\boldsymbol{v} \in \partial F(\boldsymbol{x})$, and $\boldsymbol{v}' \in \partial F(\boldsymbol{x}')$,

$$(\boldsymbol{v} - \boldsymbol{v}')^\top(\boldsymbol{x} - \boldsymbol{x}') \geqslant \iota_F \|\boldsymbol{x} - \boldsymbol{x}'\|_2^2. \tag{44}$$

Denote the largest eigenvalue of $\boldsymbol{A}^\top \boldsymbol{A}$ as $\iota_{\boldsymbol{A}}$. Then if $\iota_{\boldsymbol{A}} > 0$ and $\varrho_F \leqslant \frac{\iota_F}{\iota_{\boldsymbol{A}}}$, Definition 7 holds. If $\iota_{\boldsymbol{A}} = 0$, Definition 7 also holds trivially.

On the other hand, Definition 7 does not necessarily yield (44). For example, if $\boldsymbol{A}^\top \boldsymbol{A}$ is singular, then its smallest eigenvalue is 0, which cannot dominate a positive $\iota_F$ in (44). $\qquad\square$

## A.7 PROOF OF PROPOSITION 3.2

We recall the following lemma from ((Chaudhuri et al., 2011), Lemma 7).

**Lemma 6.** *Let the functions $\mathscr{F} : \mathbb{R}^p \to \mathbb{R}$ and $\mathscr{F}' : \mathbb{R}^p \to \mathbb{R}$ be $\varrho$-strongly convex and differentiable. If $\boldsymbol{x}_\bullet = \operatorname{argmin}_{\boldsymbol{x} \in \mathbb{R}^p} \mathscr{F}(\boldsymbol{x})$ and $\boldsymbol{x}'_\bullet = \operatorname{argmin}_{\boldsymbol{x} \in \mathbb{R}^p} \mathscr{F}'(\boldsymbol{x})$ (the solution is unique due to strong convexity, if it exists), then*

$$\left\|\boldsymbol{x}_\bullet - \boldsymbol{x}'_\bullet\right\|_2 \leqslant \frac{1}{\varrho} \max_{\boldsymbol{x} \in \mathbb{R}^p} \left\|\nabla \mathscr{F}'(\boldsymbol{x}) - \nabla \mathscr{F}(\boldsymbol{x})\right\|_2. \tag{45}$$

*Proof of Proposition 3.2.* Let $\hat{\boldsymbol{u}} := \mathcal{T}_{f_k}(\boldsymbol{u})$ and $\hat{\boldsymbol{u}}' := \mathcal{T}_{f'_k}(\boldsymbol{u})$. Then by the definition of $\mathcal{T}_{f_k}$ and $\mathcal{T}_{f'_k}$ in (16), we know that

$$\begin{bmatrix} \hat{\boldsymbol{u}} \\ \boldsymbol{c} - \hat{\boldsymbol{u}} \end{bmatrix} - \begin{bmatrix} \hat{\boldsymbol{u}}' \\ \boldsymbol{c} - \hat{\boldsymbol{u}}' \end{bmatrix} = \Pi_{\mathcal{W}}\left(\begin{bmatrix} \boldsymbol{A} \circ \mathcal{P}_{\beta,f_k \circ \boldsymbol{A}}(\boldsymbol{u}) \\ \boldsymbol{B} \circ \mathcal{P}_{\beta,g,\boldsymbol{B}}(\boldsymbol{c} - \boldsymbol{u}) \end{bmatrix}\right) - \Pi_{\mathcal{W}}\left(\begin{bmatrix} \boldsymbol{A} \circ \mathcal{P}_{\beta,f'_k \circ \boldsymbol{A}}(\boldsymbol{u}) \\ \boldsymbol{B} \circ \mathcal{P}_{\beta,g,\boldsymbol{B}}(\boldsymbol{c} - \boldsymbol{u}) \end{bmatrix}\right). \tag{46}$$

The left side of (46) satisfies

$$\left\|\begin{bmatrix} \hat{\boldsymbol{u}} \\ \boldsymbol{c} - \hat{\boldsymbol{u}} \end{bmatrix} - \begin{bmatrix} \hat{\boldsymbol{u}}' \\ \boldsymbol{c} - \hat{\boldsymbol{u}}' \end{bmatrix}\right\|_2^2 = \left\|\begin{bmatrix} \hat{\boldsymbol{u}} - \hat{\boldsymbol{u}}' \\ \hat{\boldsymbol{u}}' - \hat{\boldsymbol{u}} \end{bmatrix}\right\|_2^2 = 2\|\hat{\boldsymbol{u}} - \hat{\boldsymbol{u}}'\|_2^2. \tag{47}$$

Since the operator $\Pi_{\mathcal{W}}$ is non-expansive, the right side of (46) satisfies

$$\left\| \Pi_{\mathcal{W}}\left( \begin{bmatrix} \boldsymbol{A} \circ \mathcal{P}_{\beta, f_k \circ \boldsymbol{A}}(\boldsymbol{u}) \\ \boldsymbol{B} \circ \mathcal{P}_{\beta, g, \boldsymbol{B}}(\boldsymbol{c} - \boldsymbol{u}) \end{bmatrix} \right) - \Pi_{\mathcal{W}}\left( \begin{bmatrix} \boldsymbol{A} \circ \mathcal{P}_{\beta, f_k' \circ \boldsymbol{A}}(\boldsymbol{u}) \\ \boldsymbol{B} \circ \mathcal{P}_{\beta, g, \boldsymbol{B}}(\boldsymbol{c} - \boldsymbol{u}) \end{bmatrix} \right) \right\|_2^2$$

$$\leqslant \left\| \begin{bmatrix} \boldsymbol{A} \circ \mathcal{P}_{\beta, f_k \circ \boldsymbol{A}}(\boldsymbol{u}) \\ \boldsymbol{B} \circ \mathcal{P}_{\beta, g, \boldsymbol{B}}(\boldsymbol{c} - \boldsymbol{u}) \end{bmatrix} - \begin{bmatrix} \boldsymbol{A} \circ \mathcal{P}_{\beta, f_k' \circ \boldsymbol{A}}(\boldsymbol{u}) \\ \boldsymbol{B} \circ \mathcal{P}_{\beta, g, \boldsymbol{B}}(\boldsymbol{c} - \boldsymbol{u}) \end{bmatrix} \right\|_2^2$$

$$= \| \boldsymbol{A} \circ \left( \mathcal{P}_{\beta, f_k \circ \boldsymbol{A}}(\boldsymbol{u}) - \mathcal{P}_{\beta, f_k' \circ \boldsymbol{A}}(\boldsymbol{u}) \right) \|_2^2. \tag{48}$$

Combining (46), (47) and (48),

$$\|\mathcal{T}_{f_k}(\boldsymbol{u}) - \mathcal{T}_{f_k'}(\boldsymbol{u})\|_2$$

$$= \|\hat{\boldsymbol{u}} - \hat{\boldsymbol{u}}'\|_2$$

$$\leqslant \frac{1}{\sqrt{2}} \|\boldsymbol{A} \circ \left( \mathcal{P}_{\beta, f_k \circ \boldsymbol{A}}(\boldsymbol{u}) - \mathcal{P}_{\beta, f_k' \circ \boldsymbol{A}}(\boldsymbol{u}) \right) \|_2$$

$$\leqslant \frac{\|\boldsymbol{A}\|_2}{\sqrt{2}} \|\mathcal{P}_{\beta, f_k \circ \boldsymbol{A}}(\boldsymbol{u}) - \mathcal{P}_{\beta, f_k' \circ \boldsymbol{A}}(\boldsymbol{u})\|_2. \tag{49}$$

We apply Lemma 6 to derive a bound for $\|\mathcal{P}_{\beta, f_k \circ \boldsymbol{A}}(\boldsymbol{u}) - \mathcal{P}_{\beta, f_k' \circ \boldsymbol{A}}(\boldsymbol{u})\|_2$. Let $\mathscr{F}(\boldsymbol{x}) := f_k(\boldsymbol{x}) + \frac{\beta}{2}\|\boldsymbol{A}\boldsymbol{x} - \boldsymbol{u}\|_2^2$ and $\mathscr{F}'(\boldsymbol{x}) := f_k'(\boldsymbol{x}) + \frac{\beta}{2}\|\boldsymbol{A}\boldsymbol{x} - \boldsymbol{u}\|_2^2$. Then $\mathscr{F}(\boldsymbol{x})$ and $\mathscr{F}'(\boldsymbol{x})$ are differentiable. If $\boldsymbol{A}^\top \boldsymbol{A}$ is positive definite with the smallest eigenvalue $\omega_A > 0$, then both $\mathscr{F}(\boldsymbol{x})$ and $\mathscr{F}'(\boldsymbol{x})$ are $\beta\omega_A$-strongly convex. Thus from Lemma 6 and the definitions of $\mathcal{P}_{\beta, f_k \circ \boldsymbol{A}}$ and $\mathcal{P}_{\beta, f_k' \circ \boldsymbol{A}}$,

$$\|\mathcal{P}_{\beta, f_k \circ \boldsymbol{A}}(\boldsymbol{u}) - \mathcal{P}_{\beta, f_k' \circ \boldsymbol{A}}(\boldsymbol{u})\|_2$$

$$\leqslant \frac{1}{\beta\omega_A} \max_{\boldsymbol{x} \in \mathbb{R}^p} \|\nabla\mathscr{F}(\boldsymbol{x}) - \nabla\mathscr{F}'(\boldsymbol{x})\|_2$$

$$= \frac{1}{\beta\omega_A} \max_{\boldsymbol{x} \in \mathbb{R}^p} \|\nabla f_k(\boldsymbol{x}) - \nabla f_k'(\boldsymbol{x})\|_2$$

$$\leqslant \frac{2L}{\beta\omega_A}. \tag{50}$$

The last inequality holds because $\max_{\boldsymbol{x} \in \mathbb{R}^p} \|\nabla f_k(\boldsymbol{x})\|_2 \leqslant L$ and $\max_{\boldsymbol{x} \in \mathbb{R}^p} \|\nabla f_k'(\boldsymbol{x})\|_2 \leqslant L$ due to $L$-Lipschitz continuity and differentiability of $f_k(\boldsymbol{x})$ and $f_k'(\boldsymbol{x})$. Combining (50) and (49), (20) holds. $\qquad\square$

### A.8    PROOF OF THEOREM 3.5

**Definition 9.** For a distribution $P_{\boldsymbol{z}}$ over $\mathbb{R}^m$, define

$$R_\alpha(P_{\boldsymbol{z}}, a) = \sup_{\|\boldsymbol{r}\|_2 \leqslant a} D_\alpha(\boldsymbol{z} + \boldsymbol{r} \| \boldsymbol{z}), \tag{51}$$

where $a > 0$, $\boldsymbol{r} \in \mathbb{R}^m$, and $\boldsymbol{z} \sim P_{\boldsymbol{z}}$.

**Lemma 7** (Privacy amplification by iteration, (Feldman et al., 2018))**.** *Let* $\mathcal{T}_{f_1}, \ldots, \mathcal{T}_{f_n}$, $\mathcal{T}_{f_1'}, \ldots, \mathcal{T}_{f_n'}$ *be non-expansive operators. For an initial point* $\tilde{\boldsymbol{u}}_{(0)} \in \mathbb{R}^m$ *and a noise distributions* $P_{\boldsymbol{z}}$, *consider the noisy iterations* $\tilde{\boldsymbol{u}}_{(k)} = \mathcal{T}_{f_k}(\tilde{\boldsymbol{u}}_{(k-1)}) + \boldsymbol{z}_{(k)}$ *and* $\tilde{\boldsymbol{u}}_{(k)}' = \mathcal{T}_{f_k'}(\tilde{\boldsymbol{u}}_{(k-1)}') + \boldsymbol{z}_{(k)}$, *where* $\boldsymbol{z}_{(k)} \sim P_{\boldsymbol{z}}$. *Let* $s_{(k)} := \sup_{\boldsymbol{u} \in \mathbb{R}^m} \|\mathcal{T}_{f_k}(\boldsymbol{u}) - \mathcal{T}_{f_k'}(\boldsymbol{u})\|$ *and* $\{a_{(k)}\}_{k=1}^n$ *be a sequence of real numbers such that for any* $k$,

$$\sum_{i \leqslant k} s_{(i)} - \sum_{i \leqslant k} a_{(i)} \geqslant 0, \quad \sum_{i \leqslant n} s_{(i)} = \sum_{i \leqslant n} a_{(i)}. \tag{52}$$

*Then*

$$D_\alpha(\tilde{\boldsymbol{u}}_{(n)} \| \tilde{\boldsymbol{u}}_{(n)}') \leqslant \sum_{k=1}^n R_\alpha(P_{\boldsymbol{z}}, a_{(k)}). \tag{53}$$

*Proof of Theorem 3.5.* Recall the fact that $R_\alpha(\mathcal{N}(\mathbf{0}_m, \frac{\sigma^2}{4}\mathbf{I}_m), a) = \frac{2\alpha a^2}{\sigma^2}$, where $a > 0$. We know from Proposition 3.2 that the operators $\mathcal{T}_{f_1}, \ldots, \mathcal{T}_{f_n}$ are non-expansive. According to Condition 2, we can compute that $s_{(i)} = C$ and $s_{(k)} = 0$ for all $k \neq i$ in Lemma 7. Let $a_{(1)}, \ldots, a_{(i-1)} = 0$ and $a_{(i)}, \ldots, a_{(n)} = \frac{C}{n-i+1}$, then (52) holds. Thus Lemma 7 indicates that

$$D_\alpha(\tilde{\boldsymbol{x}}_{(n)} \| \tilde{\boldsymbol{x}}'_{(x)}) = D_\alpha(\tilde{\boldsymbol{u}}_{(n)} \| \tilde{\boldsymbol{u}}'_{(n)})$$

$$\leqslant \sum_{k=1}^n R_\alpha(\mathcal{N}(\mathbf{0}_m, \frac{\sigma^2}{4}\mathbf{I}_m), a_{(k)}) \leqslant \frac{2\alpha}{\sigma^2} \sum_{k=1}^n a_{(k)}^2 \leqslant \frac{2\alpha C^2}{\sigma^2(n-i+1)}, \tag{54}$$

which completes the proof. $\qquad\square$

### A.9 PROOF OF THEOREM 3.6

We can apply the following lemma to prove Theorem 3.6.

**Lemma 8** ((Feldman et al., 2018), Lemma 25). *Let $\{\tilde{\boldsymbol{u}}_{(k)}\}_{k\in[n]}$ and $\{\tilde{\boldsymbol{u}}'_{(k)}\}_{k\in[n]}$ be two sequences of random variables over some domain such that for all $k \in [n]$, $D_\alpha(\tilde{\boldsymbol{u}}_{(k)} \| \tilde{\boldsymbol{u}}'_{(k)}) \leqslant \frac{h}{\alpha-1}$ for some $h \in (0,1]$. Sample $K$ from $[n]$ with a probability distribution $\rho$ and denote the $K$-th variables by $\tilde{\boldsymbol{u}}_{(K)}$ and $\tilde{\boldsymbol{u}}'_{(K)}$, respectively. Then*

$$D_\alpha(\tilde{\boldsymbol{u}}_{(K)} \| \tilde{\boldsymbol{u}}'_{(K)}) \leqslant (1+h) \cdot \mathbb{E}_{k\sim\rho}\left[D_\alpha(\tilde{\boldsymbol{u}}_{(k)} \| \tilde{\boldsymbol{u}}'_{(k)})\right]. \tag{55}$$

*Proof of Theorem 3.6.* Suppose $\mathcal{D}$ and $\mathcal{D}'$ are neighboring data sets that differ in $\boldsymbol{d}_{(i)}$. Then for $k < i$, we can directly know that $D_\alpha(\tilde{\boldsymbol{u}}_{(k)} \| \tilde{\boldsymbol{u}}'_{(k)}) = 0$. For $k \geqslant i$, Theorem 3.5 indicates that $D_\alpha(\tilde{\boldsymbol{u}}_{(k)} \| \tilde{\boldsymbol{u}}'_{(k)}) \leqslant \frac{2\alpha C^2}{\sigma^2(k-i+1)}$. If $\sigma \geqslant C\sqrt{2\alpha(\alpha-1)}$, then for all $k \in [n]$,

$$D_\alpha(\tilde{\boldsymbol{u}}_{(k)} \| \tilde{\boldsymbol{u}}'_{(k)}) \leqslant \frac{1}{\alpha-1}. \tag{56}$$

Applying Lemma 8 with $h = 1$,

$$\begin{aligned}
&D_\alpha(\tilde{\boldsymbol{x}}_{(K)} \| \tilde{\boldsymbol{x}}'_{(K)}) \\
=&D_\alpha(\tilde{\boldsymbol{u}}_{(K)} \| \tilde{\boldsymbol{u}}'_{(K)}) \\
\leqslant&\frac{2}{n} \sum_{k=1}^n D_\alpha(\tilde{\boldsymbol{u}}_{(k)} \| \tilde{\boldsymbol{u}}'_{(k)}) \\
\leqslant&\frac{2}{n} \sum_{k=i}^n \frac{2\alpha C^2}{\sigma^2(k-i+1)} \\
\leqslant&\frac{4\alpha C^2 \ln(n-i+1)}{n\sigma^2} \\
\leqslant&\frac{4\alpha C^2 \ln n}{n\sigma^2}.
\end{aligned}$$

$\qquad\square$

### A.10 PROOF OF THEOREM 3.7

*Proof.* Since $0 < \tau < 1$, if $\delta < 2\sqrt{\frac{(1-\tau^2)^2}{\tau^2} - \mu^2}$, then $0 < \left(\tau^2 + \tau\sqrt{\frac{\delta^2}{4} + \mu^2}\right)^k < 1$ for all $k \in [n]$. Thus

$$\frac{1 - \left(\tau^2 + \tau\sqrt{\frac{\delta^2}{4} + \mu^2}\right)^k}{1 - \left(\tau^2 + \tau\sqrt{\frac{\delta^2}{4} + \mu^2}\right)} < \frac{1}{1 - \left(\tau^2 + \tau\sqrt{\frac{\delta^2}{4} + \mu^2}\right)}. \tag{57}$$

On the other hand, since $\delta, \mu \geqslant 0$,

$$\sqrt{\frac{\delta^2}{4} + \mu^2} \leqslant \frac{\delta}{2} + \mu. \tag{58}$$

Combining (57), (58), and Theorem 3.4,

$$\frac{1}{n} \sum_{k=1}^{n} \mathbb{E}(\|\tilde{\boldsymbol{x}}_{(k)} - \boldsymbol{x}_*\|_2^2 | \Gamma_0)$$

$$\leqslant \frac{\|\boldsymbol{V}\|_2^2 \|\boldsymbol{A}\|_2^2}{n} \sum_{k=1}^{n} \left( \tau^2 + \tau \sqrt{\frac{\delta^2}{4} + \mu^2} \right)^k \|\tilde{\boldsymbol{x}}_{(0)} - \boldsymbol{x}_*\|_2^2$$

$$+ \frac{\|\boldsymbol{V}\|_2^2}{1 - \left( \tau^2 + \tau \sqrt{\frac{\delta^2}{4} + \mu^2} \right)} \left( \frac{\delta^2}{4} + \mu^2 + \tau \left( \frac{\delta}{2} + \mu \right) \right).$$

It indicates that

$$\mathbb{E}(\|\tilde{\boldsymbol{x}}_{(K)} - \boldsymbol{x}_*\|_2^2 | \Gamma_0)$$

$$\leqslant \frac{\|\boldsymbol{V}\|_2^2 \|\boldsymbol{A}\|_2^2 \|\tilde{\boldsymbol{x}}_{(0)} - \boldsymbol{x}_*\|_2^2}{n \left( 1 - \left( \tau^2 + \tau \sqrt{\frac{\delta^2}{4} + \mu^2} \right) \right)}$$

$$+ \frac{\|\boldsymbol{V}\|_2^2}{1 - \left( \tau^2 + \tau \sqrt{\frac{\delta^2}{4} + \mu^2} \right)} \left( \frac{\delta^2}{4} + \mu^2 + \tau \left( \frac{\delta}{2} + \mu \right) \right). \tag{59}$$

For some $a > 0$, let $\boldsymbol{A}_a := a\boldsymbol{A}$, $\boldsymbol{B}_a := a\boldsymbol{B}$, $\boldsymbol{c}_a := a\boldsymbol{c}$, $\boldsymbol{V}_a := \frac{1}{a}\boldsymbol{V}$, $\beta_a := \frac{\beta}{a^2}$, $\delta_a := a\delta$ and $\mathcal{W}_a := \{\boldsymbol{w} \in \mathbb{R}^{2m} : [\boldsymbol{I}_m \, \boldsymbol{I}_m] \cdot \boldsymbol{w} = \boldsymbol{c}_a\}$. For $\boldsymbol{v} \in \mathbb{R}^m$, define

$$\mathcal{T}_{a,f_k}(\boldsymbol{v}) := \boldsymbol{H} \cdot \left( \Pi_{\mathcal{W}_a} \begin{bmatrix} \boldsymbol{A}_a \circ \mathcal{P}_{\beta_a, f_k, \boldsymbol{A}_a}(\boldsymbol{v}) \\ \boldsymbol{B}_a \circ \mathcal{P}_{\beta_a, g, \boldsymbol{B}_a}(\boldsymbol{c}_a - \boldsymbol{v}) \end{bmatrix} \right).$$

Then for any $K \in [n]$, the solution $\tilde{\boldsymbol{x}}_{(K)}$ obtained from Algorithm 1 with the initial point $\tilde{\boldsymbol{x}}_{(0)}$ can be equivalently obtained through the following iteration:

$$\tilde{\boldsymbol{v}}_{(0)} = \boldsymbol{A}_a \tilde{\boldsymbol{x}}_{(0)}$$

$$\tilde{\boldsymbol{v}}_{(k)} = \mathcal{T}_{a,f_k}(\tilde{\boldsymbol{v}}_{(k-1)}) + \frac{a\boldsymbol{z}_{(k)}}{2}, \; k = 1, 2, \ldots, K,$$

$$\tilde{\boldsymbol{x}}_{(K)} = \boldsymbol{V}_a \tilde{\boldsymbol{v}}_{(K)}.$$

From the definition of the operator $\mathcal{T}_{a,f_k}$, the sensitivity bound of $\mathcal{T}_{a,f_k}$ scales as $C_a := aC$, and the bound of $\mathbb{E}(\|\mathcal{T}_{a,f_k}(\boldsymbol{u}) - \mathcal{T}_{a,F}(\boldsymbol{u})\|_2^2)$ scales as $\mu_a^2 := a^2\mu^2$. It can be verified that the contraction parameter of $\mathcal{T}_{a,F}$ is identical to that of $\mathcal{T}_F$, both equal to $\tau$. From Theorem 3.6, by setting the variance of the Gaussian noise as $\sigma^2 = \frac{4\alpha C^2 \ln n}{n\epsilon}$, the random output $\tilde{\boldsymbol{x}}_{(K)}$ of Stop-PADM-$\mathcal{N}_\sigma$ achieves $(\alpha, \epsilon)$-RDP. Then $\delta_a = a\sqrt{m}\sigma = 2aC\sqrt{\frac{m\alpha \ln n}{n\epsilon}}$.

Then from (59),

$$\mathbb{E}(\|\tilde{\boldsymbol{x}}_{(K)} - \boldsymbol{x}_*\|_2^2 | \Gamma_0)$$

$$\leqslant \frac{\|\boldsymbol{V}_a\|_2^2 \|\boldsymbol{A}_a\|_2^2 \|\tilde{\boldsymbol{x}}_{(0)} - \boldsymbol{x}_*\|_2^2}{n \left( 1 - \left( \tau^2 + \tau \sqrt{\frac{\delta_a^2}{4} + \mu_a^2} \right) \right)} + \frac{\|\boldsymbol{V}_a\|_2^2}{1 - \left( \tau^2 + \tau \sqrt{\frac{\delta_a^2}{4} + \mu_a^2} \right)} \left( \frac{\delta_a^2}{4} + \mu_a^2 + \tau \left( \frac{\delta_a}{2} + \mu_a \right) \right)$$

$$= \frac{\|\boldsymbol{V}\|_2^2 \|\boldsymbol{A}\|_2^2 \|\tilde{\boldsymbol{x}}_{(0)} - \boldsymbol{x}_*\|_2^2}{n \left( 1 - \left( \tau^2 + \tau \sqrt{\frac{\delta_a^2}{4} + \mu_a^2} \right) \right)} + \frac{1}{a^2} \cdot \frac{\|\boldsymbol{V}\|_2^2}{1 - \left( \tau^2 + \tau \sqrt{\frac{\delta_a^2}{4} + \mu_a^2} \right)} \left( \frac{\delta_a^2}{4} + \mu_a^2 + \tau \left( \frac{\delta_a}{2} + \mu_a \right) \right)$$

$$\leqslant \frac{\|\boldsymbol{V}\|_2^2 \|\boldsymbol{A}\|_2^2 \|\tilde{\boldsymbol{x}}_{(0)} - \boldsymbol{x}_*\|_2^2}{n \left( 1 - \left( \tau^2 + \tau \sqrt{\frac{\delta_a^2}{4} + \mu_a^2} \right) \right)}$$

$$+ \frac{1}{a^2} \cdot \frac{\|\boldsymbol{V}\|_2^2}{1 - \left(\tau^2 + \tau\sqrt{\frac{\delta_a^2}{4} + \mu_a^2}\right)} \left(\frac{m\alpha C_a^2 \ln n}{n\epsilon} + \mu_a^2 + \sqrt{\frac{m\alpha C_a^2 \ln n}{n\epsilon}} + \mu_a\right)$$

$$= \frac{\|\boldsymbol{V}\|_2^2 \|\boldsymbol{A}\|_2^2 \|\tilde{\boldsymbol{x}}_{(0)} - \boldsymbol{x}_*\|_2^2}{n\left(1 - \left(\tau^2 + \tau\sqrt{\frac{m\alpha a^2 C^2 \ln n}{n\epsilon} + \mu_a^2}\right)\right)}$$

$$+ \frac{\|\boldsymbol{V}\|_2^2}{1 - \left(\tau^2 + \tau\sqrt{\frac{m\alpha a^2 C^2 \ln n}{n\epsilon} + \mu_a^2}\right)} \left(\frac{m\alpha C^2 \ln n}{n\epsilon} + \mu^2 + \sqrt{\frac{m\alpha C^2 \ln n}{a^2 n\epsilon}} + \frac{\mu}{a}\right).$$

Suppose that $\mu = 0$. Then by setting $a = \sqrt{\frac{n\epsilon}{\ln n}}$ in the above inequation, the random output $\tilde{\boldsymbol{x}}_{(K)}$ of Stop-PADM-$\mathcal{N}_\sigma$ satisfies

$$\mathbb{E}(\|\tilde{\boldsymbol{x}}_{(K)} - \boldsymbol{x}_*\|_2^2 | \Gamma_0) \leqslant \frac{\|\boldsymbol{V}\|_2^2 \|\boldsymbol{A}\|_2^2 \|\tilde{\boldsymbol{x}}_{(0)} - \boldsymbol{x}_*\|_2^2}{n\left(1 - \left(\tau^2 + \tau C\sqrt{m\alpha}\right)\right)} + \frac{\|\boldsymbol{V}\|_2^2 (m\alpha C^2 + C\sqrt{m\alpha})}{1 - \left(\tau^2 + \tau C\sqrt{m\alpha}\right)} \cdot \frac{\ln n}{n\epsilon}$$

$$= \tilde{\mathcal{O}}\left(\frac{1}{1 - \left(\tau^2 + \tau C\sqrt{m\alpha}\right)} \cdot \left(\frac{1}{n} + \frac{m\alpha C^2 + \sqrt{m\alpha}C}{n\epsilon}\right)\right),$$

which completes the proof. $\qquad\square$

### A.11 Noise from joint Laplace distribution

In this section, we analyze the scenario where the noise in private PADM is sampled from a joint distribution of a Laplace distribution $\mathcal{L}^m(\boldsymbol{0}_m, \lambda)$. Specifically, for any $k$, all components of $\boldsymbol{z}_{(k)}$ are independently drawn from a Laplace distribution $\mathcal{L}(0, \lambda)$ with some $\lambda > 0$. **The private PADM with noise $\boldsymbol{z}_{(k)} \sim \mathcal{L}^m(\boldsymbol{0}_m, \lambda)$ is denoted by PADM-$\mathcal{L}_\lambda$.**

We first prove that PADM-$\mathcal{L}_\lambda$ preserves privacy for each user in each iteration. To this end, we give and prove a related lemma.

**Lemma 9.** *Given $\boldsymbol{u}, \boldsymbol{u}' \in \mathbb{R}^m$, define random vectors $\boldsymbol{U} := \boldsymbol{u} + \boldsymbol{w}$ and $\boldsymbol{U}' := \boldsymbol{u}' + \boldsymbol{w}$, respectively, where $\boldsymbol{w} \sim \mathcal{L}^m(\boldsymbol{0}_m, b)$ for some $b > 0$. Then $D_\infty(\boldsymbol{U}\|\boldsymbol{U}') \leqslant \frac{\sqrt{m}\|\boldsymbol{u} - \boldsymbol{u}'\|_2}{b}$.*

*Proof.* Denote $P_{\boldsymbol{U}}(\boldsymbol{a})$ and $P_{\boldsymbol{U}'}(\boldsymbol{a})$ as the probability density functions w.r.t. $\boldsymbol{U}$ and $\boldsymbol{U}'$. Then from the definition of $\mathcal{L}^m(\boldsymbol{0}_m, b)$,

$$\frac{P_{\boldsymbol{U}}(\boldsymbol{a})}{P_{\boldsymbol{U}'}(\boldsymbol{a})}$$

$$= \frac{\prod_{t=1}^m e^{-\frac{|a^{(t)} - u^{(t)}|}{b}}}{\prod_{t=1}^m e^{-\frac{|a^{(t)} - u'^{(t)}|}{b}}} = \frac{e^{-\frac{\sum_{t=1}^m |a^{(t)} - u^{(t)}|}{b}}}{e^{-\frac{\sum_{t=1}^m |a^{(t)} - u'^{(t)}|}{b}}}$$

$$= e^{\frac{\|\boldsymbol{a} - \boldsymbol{u}'\|_1 - \|\boldsymbol{a} - \boldsymbol{u}\|_1}{b}}$$

$$\leqslant e^{\frac{\|\boldsymbol{u} - \boldsymbol{u}'\|_1}{b}} \leqslant e^{\frac{\sqrt{m}\|\boldsymbol{u} - \boldsymbol{u}'\|_2}{b}}. \tag{60}$$

It corresponds to the definition of $D_\infty$ that $D_\infty(\boldsymbol{U}\|\boldsymbol{U}') \leqslant \frac{\sqrt{m}\|\boldsymbol{u} - \boldsymbol{u}'\|_2}{b}$. $\qquad\square$

**Proposition A.2.** *Under Condition 2, PADM-$\mathcal{L}_\lambda$ achieves local $\frac{2\sqrt{m}C}{\lambda}$-DP for any user $k \in [n]$.*

*Proof.* Recall that $\tilde{\boldsymbol{u}}_{(k)} = \mathcal{T}_{f_k}(\boldsymbol{u}_{(k-1)}) + \frac{\boldsymbol{z}_{(k)}}{2}$ and $\tilde{\boldsymbol{u}}'_{(k)} = \mathcal{T}_{f'_k}(\boldsymbol{u}_{(k-1)}) + \frac{\boldsymbol{z}_{(k)}}{2}$ for any $k \in [n]$, where $\boldsymbol{z}_{(k)} \sim \mathcal{L}^m(\boldsymbol{0}_m, \lambda)$ with some $\lambda > 0$. We apply Lemma 9 with $\boldsymbol{w}_{(k)} := \frac{\boldsymbol{z}_{(k)}}{2}$. Then $\boldsymbol{w}_{(k)} \sim \mathcal{L}^m(\boldsymbol{0}_m, \frac{\lambda}{2})$. From Lemma 9 and the sensitivity bound of $\mathcal{T}_{f_k}$, we have

$$D_\infty\left(\tilde{\boldsymbol{u}}_{(k)}\|\tilde{\boldsymbol{u}}'_{(k)}\right) \leqslant \frac{2\sqrt{m}\|\mathcal{T}_{f_k}(\boldsymbol{u}_{(k-1)}) - \mathcal{T}_{f'_k}(\boldsymbol{u}_{(k-1)})\|_2}{\lambda} \leqslant \frac{2\sqrt{m}C}{\lambda}. \tag{61}$$

Since $\boldsymbol{V}$ is a injection, it corresponds to the definition of $D_\infty$ that

$$D_\infty\big(\tilde{\boldsymbol{x}}_{(k)}\|\tilde{\boldsymbol{x}}'_{(k)}\big) = D_\infty\big(\boldsymbol{V}\tilde{\boldsymbol{u}}_{(k)}\|\boldsymbol{V}\tilde{\boldsymbol{u}}'_{(k)}\big) = D_\infty\big(\tilde{\boldsymbol{u}}_{(k)}\|\tilde{\boldsymbol{u}}'_{(k)}\big) \leqslant \frac{2\sqrt{m}C}{\lambda}. \tag{62}$$

This completes the proof. $\qquad\qquad\square$

In addition, PADM-$\mathcal{L}_\lambda$ has stronger privacy guarantees that amplifies privacy by iteration.

**Theorem A.1.** *Under Condition 2, the final ($n$-th) output of PADM-$\mathcal{L}_\lambda$ achieves $\big(\alpha, \frac{2m\alpha C^2}{\lambda^2(n-i+1)}\big)$-RDP for its $i$-th input.*

To prove Theorem A.1, we begin with the following lemma from (Bun & Steinke, 2016) that establishes a relationship between $D_\alpha$ and $D_\infty$.

**Lemma 10.** *Let $\tilde{\boldsymbol{u}}_{(n)}$ and $\tilde{\boldsymbol{u}}'_{(n)}$ be two random variables. If $D_\infty(\tilde{\boldsymbol{u}}_{(n)}\|\tilde{\boldsymbol{u}}'_{(n)}) \leqslant \epsilon$ and $D_\infty(\tilde{\boldsymbol{u}}'_{(n)}\|\tilde{\boldsymbol{u}}_{(n)}) \leqslant \epsilon$, then $D_\alpha(\tilde{\boldsymbol{u}}_{(n)}\|\tilde{\boldsymbol{u}}'_{(n)}) \leqslant \frac{\alpha\epsilon^2}{2}$ for all $\alpha > 1$.*

*Proof of Theorem A.1.* We can deduce from the definition of $R_\alpha(\mathcal{L}(\boldsymbol{0}_m, \frac{\lambda}{2}), a)$, Lemma 9 and Lemma 10 that $R_\alpha(\mathcal{L}^m(\boldsymbol{0}_m, \frac{\lambda}{2}), a) \leqslant \frac{2m\alpha a^2}{\lambda^2}$. From Theorem 3.2, the operators $\mathcal{T}_{f_1}, \ldots, \mathcal{T}_{f_n}$ are non-expansive. According to Condition 2, we can compute that $s_{(i)} = C$ and $s_{(k)} = 0$ for all $k \neq i$ in Lemma 7. Let $a_{(1)}, \ldots, a_{(i-1)} = 0$ and $a_{(i)}, \ldots, a_{(n)} = \frac{C}{n-i+1}$, then (52) holds. Thus Lemma 7 indicates that

$$D_\alpha(\tilde{\boldsymbol{x}}_{(n)}\|\tilde{\boldsymbol{x}}'_{(n)})$$
$$=D_\alpha(\tilde{\boldsymbol{u}}_{(n)}\|\tilde{\boldsymbol{u}}'_{(n)}) \leqslant \sum_{k=1}^n R_\alpha(\mathcal{L}^m(\boldsymbol{0}_m, \frac{\lambda}{2}), a_{(k)}) \leqslant \frac{2m\alpha}{\lambda^2}\sum_{k=1}^n a_{(k)}^2 \leqslant \frac{2m\alpha C^2}{\lambda^2(n-i+1)}. \tag{63}$$

$\square$

Similar to Stop-PADM-$\mathcal{N}_\sigma$ described in Section 3.4, we denote the random stopping variant of PADM-$\mathcal{L}_\lambda$ as Stop-PADM-$\mathcal{L}_\lambda$. Then we derive the privacy parameters of Stop-PADM-$\mathcal{L}_\lambda$ as follows.

**Theorem A.2.** *Under Condition 2, the random output of Stop-PADM-$\mathcal{L}_\lambda$ for $\lambda \geqslant C\sqrt{2m\alpha(\alpha-1)}$ achieves $\big(\alpha, \frac{4m\alpha C^2 \ln n}{\lambda^2 n}\big)$-RDP.*

*Proof.* Suppose $\mathcal{D}$ and $\mathcal{D}'$ are neighboring data sets that differ in $\boldsymbol{d}_{(i)}$. Then for $k < i$, we can directly know that $D_\alpha(\tilde{\boldsymbol{u}}_{(k)}\|\tilde{\boldsymbol{u}}'_{(k)}) = 0$. For $k \geqslant i$, Theorem A.1 indicates that $D_\alpha(\tilde{\boldsymbol{u}}_{(k)}\|\tilde{\boldsymbol{u}}'_{(k)}) \leqslant \frac{2m\alpha C^2}{\lambda^2(k-i+1)}$. If $\lambda \geqslant C\sqrt{2m\alpha(\alpha-1)}$, then for all $k \in [n]$,

$$D_\alpha(\tilde{\boldsymbol{u}}_{(k)}\|\tilde{\boldsymbol{u}}'_{(k)}) \leqslant \frac{1}{\alpha-1}. \tag{64}$$

We apply Lemma 8 with $h = 1$, then

$$D_\alpha(\tilde{\boldsymbol{x}}_{(K)}\|\tilde{\boldsymbol{x}}'_{(K)})$$
$$=D_\alpha(\tilde{\boldsymbol{u}}_{(K)}\|\tilde{\boldsymbol{u}}'_{(K)})$$
$$\leqslant\frac{2}{n}\sum_{k=1}^n D_\alpha(\tilde{\boldsymbol{u}}_{(k)}\|\tilde{\boldsymbol{u}}'_{(k)})$$
$$\leqslant\frac{2}{n}\sum_{k=i}^n \frac{2m\alpha C^2}{\lambda^2(k-i+1)}$$
$$\leqslant\frac{4m\alpha C^2 \ln(n-i+1)}{n\lambda^2}$$
$$\leqslant\frac{4m\alpha C^2 \ln n}{n\lambda^2},$$

which completes the proof. $\qquad\qquad\square$

The following theorem examines the privacy-utility tradeoff of Stop-PADM-$\mathcal{L}_\lambda$.

**Theorem A.3** (Privacy-utility Tradeoff of Stop-PADM-$\mathcal{L}_\lambda$). *Suppose the operator $\mathcal{T}_F$ is $\tau$-contractive with some $0 < \tau < 1$ and has a fixed point $\boldsymbol{u}_* \in \mathbb{R}^m$. Under Conditions 1 and 2, if $\mu = 0$ and $\lambda < \sqrt{\frac{2}{m}\left(\frac{(1-\tau^2)^2}{\tau^2} - \mu^2\right)}$, then Stop-PADM-$\mathcal{L}_\lambda$ satisfies*

$$\mathbb{E}(\|\tilde{\boldsymbol{x}}_{(K)} - \boldsymbol{x}_*\|_2^2 | \Gamma_0) \leqslant \tilde{\mathcal{O}}\left(\frac{1}{1 - \left(\tau^2 + \tau m C \sqrt{2\alpha}\right)} \cdot \left(\frac{1}{n} + \frac{2\alpha m^2 C^2 + \sqrt{2\alpha} m C}{n\epsilon}\right)\right), \qquad (65)$$

*where $\boldsymbol{x}_* = \boldsymbol{V}\boldsymbol{u}_*$, $\alpha > 1$ and $\epsilon$ are the privacy parameters such that the output $\tilde{\boldsymbol{x}}_{(K)}$ of Stop-PADM-$\mathcal{L}_\lambda$ achieves $(\alpha, \epsilon)$-RDP.*

*Proof.* From Theorem A.2, by setting the Laplace noise parameter $\lambda = C\sqrt{\frac{4m\alpha \ln n}{n\epsilon}}$, the random output $\tilde{\boldsymbol{x}}_{(K)}$ of Stop-PADM-$\mathcal{L}_\lambda$ achieves $(\alpha, \epsilon)$-RDP. Following a similar derivation as in Appendix A.10, by setting $\delta_a = a\sqrt{2m}\lambda = amC\sqrt{\frac{8\alpha \ln n}{n\epsilon}}$, we can conclude that the random output $\tilde{\boldsymbol{x}}_{(K)}$ of Stop-PADM-$\mathcal{L}_\lambda$ satisfies

$$\mathbb{E}(\|\tilde{\boldsymbol{x}}_{(K)} - \boldsymbol{x}_*\|_2^2 | \Gamma_0) \leqslant \frac{\|\boldsymbol{V}\|_2^2 \|\boldsymbol{A}\|_2^2 \|\tilde{\boldsymbol{x}}_{(0)} - \boldsymbol{x}_*\|_2^2}{n\left(1 - \left(\tau^2 + \tau m C \sqrt{2\alpha}\right)\right)} + \frac{\|\boldsymbol{V}\|_2^2 (2\alpha m^2 C^2 + m C \sqrt{2\alpha})}{1 - \left(\tau^2 + \tau m C \sqrt{2\alpha}\right)} \cdot \frac{\ln n}{n\epsilon}$$

$$= \tilde{\mathcal{O}}\left(\frac{1}{1 - \left(\tau^2 + \tau m C \sqrt{2\alpha}\right)} \cdot \left(\frac{1}{n} + \frac{2\alpha m^2 C^2 + \sqrt{2\alpha} m C}{n\epsilon}\right)\right),$$

which completes the proof. $\qquad\square$

### A.12  SYNTHETIC EXPERIMENT

In this scenario, the experimental settings basically follow the evaluation benchmark of (Cyffers et al., 2023; Chan et al., 2024). The evaluation model is an elastic-net regularized linear regression model (Zou & Hastie, 2005):

$$\min_{\boldsymbol{x} \in \mathbb{R}^p} \{F(\boldsymbol{x}) + g(\boldsymbol{y})\} \quad s.t. \ \boldsymbol{x} = \boldsymbol{y}, \qquad (66)$$

$$\text{where } F(\boldsymbol{x}) := \frac{1}{n}\|\boldsymbol{R}\boldsymbol{x} - \boldsymbol{b}\|_2^2 = \frac{1}{n}\sum_{k=1}^n \left(\boldsymbol{R}^{(k)}\boldsymbol{x} - b^{(k)}\right)^2 =: \frac{1}{n}\sum_{k=1}^n f_k(\boldsymbol{x}), \qquad (67)$$

$$g(\boldsymbol{y}) := \kappa_1\|\boldsymbol{y}\|_1 + \frac{\kappa_2}{2}\|\boldsymbol{y}\|_2^2. \qquad (68)$$

In this model, $F(\boldsymbol{x})$ and $g(\boldsymbol{y})$ are the fidelity term and the regularization term, respectively. $\kappa_1, \kappa_2 \geqslant 0$ control the strengths of $\ell_1$-norm and ridge regularizations, respectively. For each $k \in [n]$, the private information $\boldsymbol{d}_{(k)}$ of the $k$-th user refers to the data item $(\boldsymbol{R}^{(k)}, b^{(k)})$, where $\boldsymbol{R}^{(k)}$ or $b^{(k)}$ denotes the $k$-th row or component of $\boldsymbol{R} \in \mathbb{R}^{n \times p}$ or $\boldsymbol{b} \in \mathbb{R}^n$, respectively. $f_k(\boldsymbol{x})$ represents the private function for the $k$-th user.

We use the private PADM defined in (16) to solve model (66):

$$\tilde{\boldsymbol{x}}_{(k)} = \boldsymbol{H} \cdot \left(\Pi_{\mathcal{W}}\left(\begin{bmatrix} \mathcal{P}_{\beta, f_k, \boldsymbol{I}_p}(\tilde{\boldsymbol{x}}_{(k-1)}) + \boldsymbol{z}_{(k)} \\ -\mathcal{P}_{\beta, g, (-\boldsymbol{I}_p)}(-\tilde{\boldsymbol{x}}_{(k-1)}) \end{bmatrix}\right)\right), \qquad (69)$$

where $\boldsymbol{z}_{(k)}$ denotes the noise vector and the final output $\tilde{\boldsymbol{x}}_{(n)}$ is the resulting feasible solution obtained from private PADM. The following two propositions provide the closed-forms of the operators $\mathcal{P}_{\beta, f_k, \boldsymbol{I}_p}$ and $\mathcal{P}_{\beta, g, (-\boldsymbol{I}_p)}$, respectively.

**Proposition A.3.** *For any $k \in [n]$,*

$$\mathcal{P}_{\beta, f_k, \boldsymbol{I}_p}(\tilde{\boldsymbol{x}}_{(k-1)}) = \left(\beta \boldsymbol{I}_p + 2(\boldsymbol{R}^{(k)})^\top \boldsymbol{R}^{(k)}\right)^{-1}\left(\beta\tilde{\boldsymbol{x}}_{(k-1)} + 2b^{(k)}(\boldsymbol{R}^{(k)})^\top\right). \qquad (70)$$

*Proof.* Let $\mathscr{P}_k(\boldsymbol{x}) := f_k(\boldsymbol{x}) + \frac{\beta}{2}\|\boldsymbol{x} - \tilde{\boldsymbol{x}}_{(k-1)}\|_2^2$, which is convex. Let

$$\hat{\boldsymbol{x}}_{(k)} := \left(\beta \boldsymbol{I}_p + 2(\boldsymbol{R}^{(k)})^\top \boldsymbol{R}^{(k)}\right)^{-1}\left(\beta\tilde{\boldsymbol{x}}_{(k-1)} + 2b^{(k)}(\boldsymbol{R}^{(k)})^\top\right).$$

Then $\nabla \mathscr{P}_k(\hat{\boldsymbol{x}}_{(k)}) = 0$, which implies that $\hat{\boldsymbol{x}}_{(k)}$ minimizes $\mathscr{P}_k$. Then we can obtain (70) from the definition of $\mathcal{P}_{\beta, f_k, \boldsymbol{I}_p}(\tilde{\boldsymbol{x}}_{(k-1)})$. $\qquad\square$

Since $\beta > 0$, the matrix $\left(\beta \boldsymbol{I}_p + 2(\boldsymbol{R}^{(k)})^\top \boldsymbol{R}^{(k)}\right)$ is positive definite and thus (70) is well-defined.

**Proposition A.4.** *For any $k \in [n]$ and $i \in [p]$,*

$$\mathcal{P}_{\beta, g, (-\boldsymbol{I}_p)}(-\tilde{\boldsymbol{x}}_{(k-1)}) = \hat{\boldsymbol{y}}_{(k)},$$

$$\hat{y}_{(k)}^{(i)} := \begin{cases} \frac{\beta \tilde{x}_{(k-1)}^{(i)} - \kappa_1}{\beta + \kappa_2} & \text{if } \tilde{x}_{(k-1)}^{(i)} > \frac{\kappa_1}{\beta}, \\ 0 & \text{else if } -\frac{\kappa_1}{\beta} \leqslant \tilde{x}_{(k-1)}^{(i)} \leqslant \frac{\kappa_1}{\beta}, \\ \frac{\beta \tilde{x}_{(k-1)}^{(i)} + \kappa_1}{\beta + \kappa_2} & \text{else.} \end{cases} \tag{71}$$

*Proof.* Let $G(\boldsymbol{y}) := \kappa_1 \|\boldsymbol{y}\|_1 + \frac{\kappa_2}{2} \|\boldsymbol{y}\|_2^2 + \frac{\beta}{2} \|\boldsymbol{y} - \tilde{\boldsymbol{x}}_{(k-1)}\|_2^2$, which is convex. We can compute that

$$\partial G(\boldsymbol{y}) = \kappa_1 \partial(\|\boldsymbol{y}\|_1) + \kappa_2 \boldsymbol{y} + \beta(\boldsymbol{y} - \tilde{\boldsymbol{x}}_{(k-1)}).$$

From the definition of $\hat{\boldsymbol{y}}_{(k)}$, we can verify that $\boldsymbol{0}_p \in \partial G(\hat{\boldsymbol{y}}_{(k)})$. Then (71) holds from the definition of $\mathcal{P}_{\beta, g, (-\boldsymbol{I}_p)}(-\tilde{\boldsymbol{x}}_{(k-1)})$. $\qquad\square$

**Proposition A.5.** *Let the functions $F$ and $g$ be defined in (67) and (68), respectively. Let $\boldsymbol{A} := \boldsymbol{I}_p$ and $\boldsymbol{B} := -\boldsymbol{I}_p$. If either the matrix $(\boldsymbol{R}^\top \boldsymbol{R})$ is positive definite or $\kappa_2 > 0$, then the operator $\mathcal{T}_F$ defined in (14) is $\tau$-contractive, where $\tau := \frac{1}{2} \cdot \frac{\beta}{\beta + \frac{2}{n}\omega_{\boldsymbol{R}}} + \frac{1}{2} \cdot \frac{\beta}{\beta + \kappa_2}$ and $\omega_{\boldsymbol{R}}$ denotes the smallest eigenvalue of $(\boldsymbol{R}^\top \boldsymbol{R})$.*

*Proof.* For any $\boldsymbol{x} \in \mathbb{R}^p$ (or $\boldsymbol{x}' \in \mathbb{R}^p$), define $\hat{\boldsymbol{x}} := \mathcal{P}_{\beta, F, \boldsymbol{I}_p}(\boldsymbol{x})$ and $\hat{\boldsymbol{y}} := \mathcal{P}_{\beta, g, (-\boldsymbol{I}_p)}(-\boldsymbol{x})$ (or $\hat{\boldsymbol{x}}' := \mathcal{P}_{\beta, F, \boldsymbol{I}_p}(\boldsymbol{x}')$ and $\hat{\boldsymbol{y}}' := \mathcal{P}_{\beta, g, (-\boldsymbol{I}_p)}(-\boldsymbol{x}')$). Then $\mathcal{T}_F(\boldsymbol{x}) = \frac{\hat{\boldsymbol{x}} + \hat{\boldsymbol{y}}}{2}$ (or $\mathcal{T}_F(\boldsymbol{x}') = \frac{\hat{\boldsymbol{x}}' + \hat{\boldsymbol{y}}'}{2}$).

First, we need to prove that

$$\|\hat{\boldsymbol{x}} - \hat{\boldsymbol{x}}'\|_2 \leqslant \frac{\beta}{\beta + \frac{2}{n}\omega_{\boldsymbol{R}}} \|\boldsymbol{x} - \boldsymbol{x}'\|_2. \tag{72}$$

By the definition of $\mathcal{P}_{\beta, F, \boldsymbol{I}_p}$,

$$\mathcal{P}_{\beta, F, \boldsymbol{I}_p}(\boldsymbol{x}) = \left(\beta \boldsymbol{I}_p + \frac{2}{n} \boldsymbol{R}^\top \boldsymbol{R}\right)^{-1} \left(\beta \boldsymbol{x} + \frac{2}{n} \boldsymbol{R}^\top \boldsymbol{b}\right). \tag{73}$$

We can compute $\left\|\left(\beta \boldsymbol{I}_p + \frac{2}{n} \boldsymbol{R}^\top \boldsymbol{R}\right)^{-1}\right\|_2 = \frac{1}{\beta + \frac{2}{n}\omega_{\boldsymbol{R}}}$, which indicates that

$$\|\hat{\boldsymbol{x}} - \hat{\boldsymbol{x}}'\|_2 \leqslant \beta \left\|\left(\beta \boldsymbol{I}_p + 2\boldsymbol{R}^\top \boldsymbol{R}\right)^{-1}\right\|_2 \|\boldsymbol{x} - \boldsymbol{x}'\|_2 = \frac{\beta \|\boldsymbol{x} - \boldsymbol{x}'\|_2}{\beta + \frac{2}{n}\omega_{\boldsymbol{R}}}.$$

Next, we need to prove that

$$\|\hat{\boldsymbol{y}} - \hat{\boldsymbol{y}}'\|_2 \leqslant \frac{\beta}{\beta + \kappa_2} \|\boldsymbol{x} - \boldsymbol{x}'\|_2. \tag{74}$$

Recall the definition of $\mathcal{P}_{\beta, g, (-\boldsymbol{I}_p)}$ in (71) that for all $i \in [p]$,

$$[\mathcal{P}_{\beta, g, (-\boldsymbol{I}_p)}(-\boldsymbol{x})]^{(i)} = \hat{y}^{(i)} = \begin{cases} \frac{\beta x^{(i)} - \kappa_1}{\beta + \kappa_2} & \text{if } x^{(i)} > \frac{\kappa_1}{\beta}, \\ 0 & \text{else if } -\frac{\kappa_1}{\beta} \leqslant x^{(i)} \leqslant \frac{\kappa_1}{\beta}, \\ \frac{\beta x^{(i)} + \kappa_1}{\beta + \kappa_2} & \text{else.} \end{cases} \tag{75}$$

We then prove (74) by showing that for all $i \in [p]$,

$$|\hat{y}^{(i)} - \hat{y}'^{(i)}| \leqslant \frac{\beta}{\beta + \kappa_2} |x^{(i)} - x'^{(i)}|. \tag{76}$$

For any $i \in [p]$, the values of $x^{(i)}$ and $x'^{(i)}$ can be classified into 6 cases:

(i) $x^{(i)}, x'^{(i)} > \frac{\kappa_1}{\beta}$,

(ii) $x^{(i)}, x'^{(i)} \in [-\frac{\kappa_1}{\beta}, \frac{\kappa_1}{\beta}]$,

(iii) $x^{(i)}, x'^{(i)} < -\frac{\kappa_1}{\beta}$,

(iv) $x^{(i)} > \frac{\kappa_1}{\beta}, x'^{(i)} \in [-\frac{\kappa_1}{\beta}, \frac{\kappa_1}{\beta}]$,

(v) $x^{(i)} > \frac{\kappa_1}{\beta}, x'^{(i)} < -\frac{\kappa_1}{\beta}$,

(vi) $x^{(i)} \in [-\frac{\kappa_1}{\beta}, \frac{\kappa_1}{\beta}], x'^{(i)} < -\frac{\kappa_1}{\beta}$.

Other cases are identical to one of the above by switching $x^{(i)}$ and $x'^{(i)}$. For cases $(i)$, $(ii)$ and $(iii)$, it can be directly seen that (76) holds. For case $(iv)$, since $\frac{\beta x^{(i)} - \kappa_1}{\beta + \kappa_2} > 0$ and $\frac{\beta x'^{(i)} - \kappa_1}{\beta + \kappa_2} \leqslant 0$, then $|\hat{y}^{(i)} - \hat{y}'^{(i)}| = |\frac{\beta x^{(i)} - \kappa_1}{\beta + \kappa_2}| \leqslant |\frac{\beta x^{(i)} - \kappa_1}{\beta + \kappa_2} - \frac{\beta x'^{(i)} - \kappa_1}{\beta + \kappa_2}| = \frac{\beta}{\beta + \kappa_2}|x^{(i)} - x'^{(i)}|$. For case $(v)$, since $\frac{\beta x^{(i)}}{\beta + \kappa_2} > \frac{\kappa_1}{\beta + \kappa_2}$ and $\frac{\beta x'^{(i)}}{\beta + \kappa_2} < \frac{-\kappa_1}{\beta + \kappa_2}$, then $|\hat{y}^{(i)} - \hat{y}'^{(i)}| = |\frac{\beta x^{(i)} - \kappa_1}{\beta + \kappa_2} - \frac{\beta x'^{(i)} + \kappa_1}{\beta + \kappa_2}| < |\frac{\beta x^{(i)}}{\beta + \kappa_2} - \frac{\beta x'^{(i)}}{\beta + \kappa_2}| = \frac{\beta}{\beta + \kappa_2}|x^{(i)} - x'^{(i)}|$. For case (vi), since $\frac{\beta x^{(i)} + \kappa_1}{\beta + \kappa_2} \geqslant 0$ and $\frac{\beta x'^{(i)} + \kappa_1}{\beta + \kappa_2} < 0$, then $|\hat{y}^{(i)} - \hat{y}'^{(i)}| = |\frac{\beta x'^{(i)} + \kappa_1}{\beta + \kappa_2}| \leqslant |\frac{\beta x^{(i)} + \kappa_1}{\beta + \kappa_2} - \frac{\beta x'^{(i)} + \kappa_1}{\beta + \kappa_2}| = \frac{\beta}{\beta + \kappa_2}|x^{(i)} - x'^{(i)}|$. Summarizing these cases, we can conclude that (76) holds, which implies that $\|\hat{\boldsymbol{y}} - \hat{\boldsymbol{y}}'\|_2^2 = \sum_{i=1}^p |\hat{y}^{(i)} - \hat{y}'^{(i)}|^2 \leqslant (\frac{\beta}{\beta + \kappa_2})^2 \sum_{i=1}^p |x^{(i)} - x'^{(i)}|^2 = (\frac{\beta}{\beta + \kappa_2})^2 \|\boldsymbol{x} - \boldsymbol{x}'\|_2^2$. Therefore, (74) holds.

Combining (72) and (74), we know that $\|\mathcal{T}_F(\boldsymbol{x}) - \mathcal{T}_F(\boldsymbol{x}')\|_2 \leqslant \frac{1}{2}\|\hat{\boldsymbol{x}} - \hat{\boldsymbol{x}}'\|_2 + \frac{1}{2}\|\hat{\boldsymbol{y}} - \hat{\boldsymbol{y}}'\|_2 \leqslant \tau\|\boldsymbol{x} - \boldsymbol{x}'\|_2$ with $\tau := \frac{1}{2} \cdot \frac{\beta}{\beta + \frac{2}{n}\omega_{\boldsymbol{R}}} + \frac{1}{2} \cdot \frac{\beta}{\beta + \kappa_2}$, which completes the proof. $\square$

This proposition indicates that the contraction of $\mathcal{T}_F$ depends on the strong convexity of either $F$ or $g$. Once $\mathcal{T}_F$ is contractive, the value of $\tau$ can be tuned by $\beta$. The entire algorithm for solving model (66) is summarized in Algorithm 2.

---

**Algorithm 2:** Private PADM for solving the elastic net model

**Input:** The data set $\mathcal{D} = (\boldsymbol{R}, \boldsymbol{b})$, the regularization parameters $\kappa_1 > 0$ and $\kappa_2 > 0$, the parameter $\beta > 0$, and a starting point $\tilde{\boldsymbol{x}}_{(0)} \in \mathbb{R}^p$.

1: **for** $k \in [n]$ **do**
2:     Sample $\boldsymbol{z}_{(k)}$ from a noise distribution $P_{\boldsymbol{z}}$ on $\mathbb{R}^p$ with mean zero and variance $\delta^2$.
3:     Compute $\hat{\boldsymbol{x}}_k = \mathcal{P}_{\beta, f_k, \boldsymbol{I}_p}(\tilde{\boldsymbol{x}}_{(k-1)})$ by (70).
4:     Compute $\hat{\boldsymbol{y}}_k = \mathcal{P}_{\beta, g, (-\boldsymbol{I}_p)}(-\tilde{\boldsymbol{x}}_{(k-1)})$ by (71).
5:     Compute $\tilde{\boldsymbol{x}}_{(k)} = \frac{\hat{\boldsymbol{x}}_k + \hat{\boldsymbol{y}}_k + \boldsymbol{z}_{(k)}}{2}$.
6: **end for**
**Output:** A feasible solution $\tilde{\boldsymbol{x}}_{(n)}$.

---

**Data generation:** The data dimensionalities are set to $n = 1000$ and $p = 64$. The matrix $\boldsymbol{R}$ is generated with entries drawn independently from a standard Gaussian distribution $\mathcal{N}(0, 1)$. The true coefficient vector $\boldsymbol{x}_*$ is defined as follows: $x_*^{(i)} = 3$ for $i \in \lfloor p/5 \rfloor$ and 0 otherwise, where $\lfloor \cdot \rfloor$ denotes the floor operation. The response vector $\boldsymbol{b}$ is computed by $\boldsymbol{b} = \boldsymbol{R}\boldsymbol{x}_* + \boldsymbol{\varsigma}$, where $\boldsymbol{\varsigma} \sim \mathcal{N}(\boldsymbol{0}_n, 0.01\boldsymbol{I}_n)$ is a standard Gaussian noise term that simulates real-world data perturbations.

**Parameter setup:** The regularization parameters $\kappa_1$ and $\kappa_2$ are both set to $1e - 3$. The bound $\mu$ defined in Condition 1 is approximately set to zero. We ensure a precise evaluation of the sensitivity bound defined in Condition 2 by clipping, as widely-used in privacy-preserving machine learning. The sensitivity bound is set to $C = 1/(20 \cdot p)$. We convert each privacy budget $(\epsilon_{DP}, \vartheta)$-DP to the corresponding RDP, and then set the noise parameters $\lambda$ and $\sigma$ in PADM-$\mathcal{L}_\lambda$ and PADM-$\mathcal{N}_\sigma$ according to Theorems A.1 and 3.5 with $i = 1$, respectively. The parameter $\beta$ of the operators $\mathcal{P}_{\beta, f_k, \boldsymbol{I}_p}$ and $\mathcal{P}_{\beta, g, -\boldsymbol{I}_p}$ can be set according to Theorem 3.4 to ensure convergence. For statistical reliability, we repeat the experiment for 100 times on each privacy budget.

### A.13 REAL-WORLD EXPERIMENT

In this scenario, the evaluation model is an elastic-net regularized logistic regression model given by (66), with the fidelity term $F(\boldsymbol{x})$ defined as:

$$F(\boldsymbol{x}) := \frac{1}{n} \sum_{k=1}^{n} \ln\left(1 + e^{-b^{(k)} \cdot \boldsymbol{R}^{(k)} \boldsymbol{x}}\right) := \frac{1}{n} \sum_{k=1}^{n} f_k(\boldsymbol{x}), \tag{77}$$

and the regularization term $g(\boldsymbol{y})$ defined as (68).

Similar to the synthetic experiment, the private ADMM for solving this model can be formulated as (69). To implement private PADM, we require a closed-form of the operator $\mathcal{P}_{\beta, f_k, \boldsymbol{I}_p}$. For computational efficiency, we approximate $f_k$ by its first-order Taylor expansion, denoted by $\hat{f}_k$, and derive the closed-form of the operator $\mathcal{P}_{\beta, \hat{f}_k, \boldsymbol{I}_p}$.

**Proposition A.6.** *For any $k \in [n]$,*

$$\mathcal{P}_{\beta, \hat{f}_k, \boldsymbol{I}_p}(\tilde{\boldsymbol{x}}_{(k-1)}) = \tilde{\boldsymbol{x}}_{(k-1)} + \frac{b^{(k)}}{\beta} \mathbf{R}^{(k)\top} \left(\frac{1}{1 + e^{b^{(k)} \mathbf{R}^{(k)} \tilde{\boldsymbol{x}}_{(k-1)}}}\right), \tag{78}$$

*where $\hat{f}_k(\boldsymbol{x}) := \ln\left(1 + e^{-b^{(k)} \cdot \boldsymbol{R}^{(k)} \tilde{\boldsymbol{x}}_{(k-1)}}\right) - \frac{b^{(k)}}{1 + e^{b^{(k)} \cdot \boldsymbol{R}^{(k)} \tilde{\boldsymbol{x}}_{(k-1)}}} \boldsymbol{R}^{(k)}(\boldsymbol{x} - \tilde{\boldsymbol{x}}_{(k-1)})$ is the first-order Taylor expansion of $f$ at $\tilde{\boldsymbol{x}}_{(k-1)}$.*

*Proof.* Let $\hat{\mathscr{P}}_k(\boldsymbol{x}) := \hat{f}_k(\boldsymbol{x}) + \frac{\beta}{2}\|\boldsymbol{x} - \tilde{\boldsymbol{x}}_{(k-1)}\|_2^2$, which is convex. Let

$$\hat{\boldsymbol{x}}_{(k)} := \tilde{\boldsymbol{x}}_{(k-1)} + \frac{b^{(k)}}{\beta} \mathbf{R}^{(k)\top} \left(\frac{1}{1 + e^{b^{(k)} \mathbf{R}^{(k)} \tilde{\boldsymbol{x}}_{(k-1)}}}\right).$$

Then $\nabla\hat{\mathscr{P}}_k(\hat{\boldsymbol{x}}_{(k)}) = 0$, which implies that $\hat{\boldsymbol{x}}_{(k)}$ minimizes $\hat{\mathscr{P}}_k$. Then we can obtain (78) from the definition of $\mathcal{P}_{\beta, \hat{f}_k, \boldsymbol{I}_p}(\tilde{\boldsymbol{x}}_{(k-1)})$. $\square$

The closed-form of the operator $\mathcal{P}_{\beta, g, (-\boldsymbol{I}_p)}$ is given in Proposition A.4. The contraction parameter for the operator $\mathcal{T}_F$ can be derived following the same method as in Proposition A.5, and is therefore omitted here. The complete algorithm for solving the logistic regression model is similar to Algorithm 2, with the only modification that we compute $\hat{\boldsymbol{x}}_k = \mathcal{P}_{\beta, \hat{f}_k, \boldsymbol{I}_p}(\tilde{\boldsymbol{x}}_{(k-1)})$ by (78) in step 3.

**Data processing:** We conduct the experiments on the Adult data set from the UCI Machine Learning Repository: https://archive.ics.uci.edu/dataset/2/adult. The data set comprises 48842 instances, each containing 14 personal attributes including age, sex, education level, and native country. The binary classification task predicts whether the income of individual exceeds \$50,000. Following the preprocessing procedure described in (Huang et al., 2020), we transform categorical attributes into binary vectors, normalize all features, and map the original labels $\{> 50000, \leq 50000\}$ to $\{+1, -1\}$. After preprocessing, each instance is represented by a 96-dimensional feature vector and a corresponding binary label. In the experiments, we randomly select 2,000 instances for training and another 2,000 for testing, with both sets containing 50% positive and 50% negative samples.

**Parameter setup:** The regularization parameters $\kappa_1$ and $\kappa_2$ are both set to $1e-3$. The bound $\mu$ defined in Condition 1 is approximately set to zero. We ensure a precise evaluation of the sensitivity bound defined in Condition 2 by clipping, as widely-used in privacy-preserving machine learning. The sensitivity bound is set to $C = 1/p$. We convert each privacy budget $(\epsilon_{DP}, \vartheta)$-DP to the corresponding RDP, and then set the noise parameters $\lambda$ and $\sigma$ in PADM-$\mathcal{L}_\lambda$ and PADM-$\mathcal{N}_\sigma$ according to Theorems A.1 and 3.5 with $i = 1$, respectively. The parameter $\beta$ of the operators $\mathcal{P}_{\beta, f_k, \boldsymbol{I}_p}$ and $\mathcal{P}_{\beta, g, -\boldsymbol{I}_p}$ can be set according to Theorem 3.4 to ensure convergence.

### A.14 ADDITIONAL EXPERIMENTAL RESULTS

The experiments are carried out on a desktop workstation with an Intel Core i9-14900KF CPU, 64GB DDR5 6000-MHZ memory cards, and an Nvidia RTX-4080 graphics card with 16-GB independent memory.

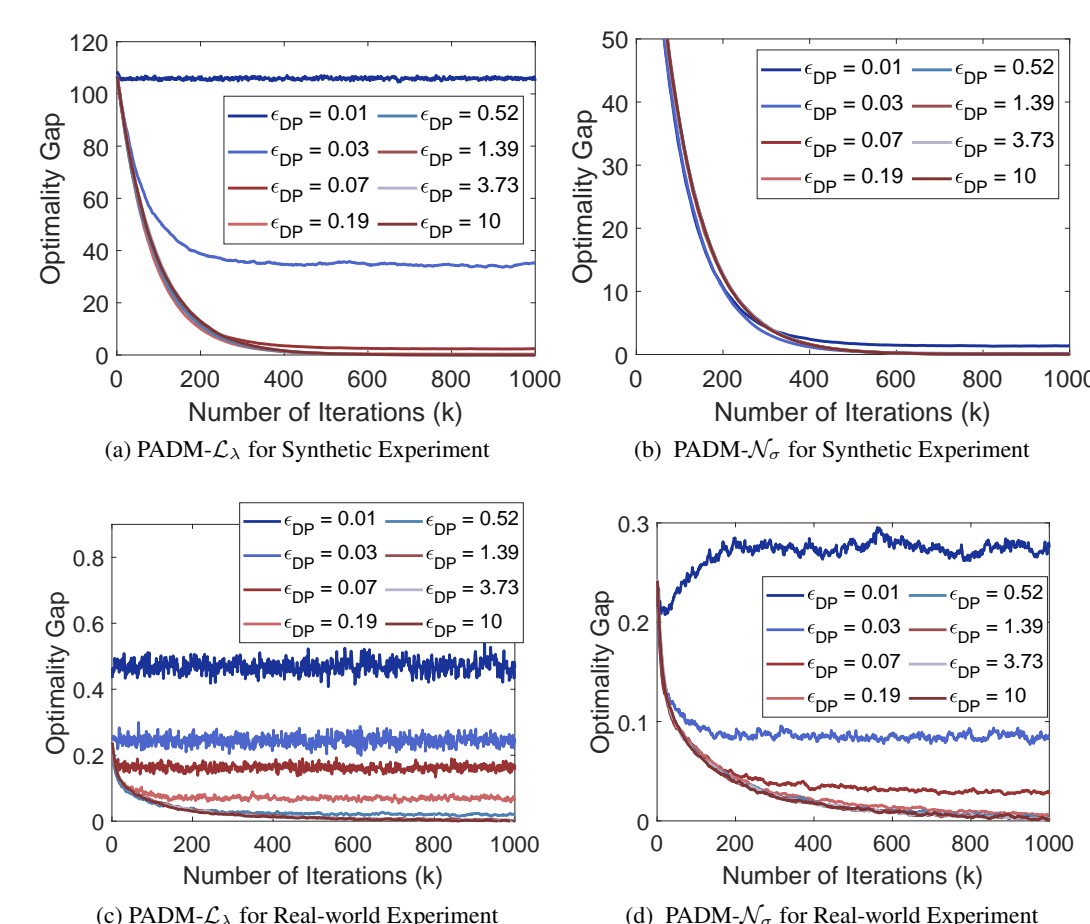

(a) PADM-$\mathcal{L}_\lambda$ for Synthetic Experiment

(b) PADM-$\mathcal{N}_\sigma$ for Synthetic Experiment

(c) PADM-$\mathcal{L}_\lambda$ for Real-world Experiment

(d) PADM-$\mathcal{N}_\sigma$ for Real-world Experiment

Figure 3: Optimality gaps of PADM-$\mathcal{L}_\lambda$ (left) and PADM-$\mathcal{N}_\sigma$ (right) at 1000-th iteration under eight privacy budgets $\epsilon_{DP}$.

**The convergence of PADM-$\mathcal{L}_\lambda$ and PADM-$\mathcal{N}_\sigma$:** We verify the convergence of PADM-$\mathcal{L}_\lambda$ and PADM-$\mathcal{N}_\sigma$ for different privacy budgets by the optimality gap (Chan et al., 2024), which is defined as the difference between the objective function values of PADM-$\mathcal{L}_\lambda$ or PADM-$\mathcal{N}_\sigma$ in each iteration and pure PADM. Results for PADM-$\mathcal{L}_\lambda$ and PADM-$\mathcal{N}_\sigma$ are shown in Figure 3. For very small privacy budgets, larger noise variances are required to meet the privacy budget, which affects the convergence performance.

In the synthetic experiment, the optimality gaps of PADM-$\mathcal{L}_\lambda$ for the budget $\epsilon_{DP} \leqslant 0.03$ decrease only a little at the beginning of the iterations, resulting in a considerable optimality gap. This behavior reflects the inherent trade-off between privacy and utility, where stricter privacy requirements lead to slower convergence and larger optimality gaps. However, for moderate privacy budgets ($\epsilon_{DP} > 0.03$), PADM-$\mathcal{L}_\lambda$ converges within 400 iterations, and the optimality gap approaches zero. In contrast, for all privacy budgets, the optimality gap of PADM-$\mathcal{N}_\sigma$ converges to zero. This indicates that even for very small privacy budgets, PADM-$\mathcal{N}_\sigma$ can achieve satisfactory solutions. In the real-world experiment, PADM-$\mathcal{L}_\lambda$ decreases the optimality gaps and converges for moderate privacy budgets $\epsilon_{DP} > 0.19$, while PADM-$\mathcal{N}_\sigma$ decreases the optimality gaps for $\epsilon_{DP} > 0.01$, and converges for $\epsilon_{DP} > 0.3$. The results demonstrate the effectiveness of the proposed methods on the real-world data.

The optimality gaps of PADM-$\mathcal{L}_\lambda$ and PADM-$\mathcal{N}_\sigma$ at 1000-th iteration under eight privacy budgets $\epsilon_{DP}$ are shown in Table 1. PADM achieves optimality with sufficiently large privacy budgets.

**Final objective function values:** Final objective function values (mean $\pm$ STD) of federated private ADMM, private ADMM, PNSGD, PADM-$\mathcal{L}_\lambda$, and PADM-$\mathcal{N}_\sigma$ are shown in Table 2. PADM-$\mathcal{N}_\sigma$

Table 1: Optimality gaps of PADM-$\mathcal{L}_\lambda$ and PADM-$\mathcal{N}_\sigma$ at 1000-th iteration under eight privacy budgets $\epsilon_{DP}$.

| Synthetic Experiment | | | | | | | | |
|---|---|---|---|---|---|---|---|---|
| $\epsilon_{DP}$ | 0.01 | 0.03 | 0.07 | 0.19 | 0.52 | 1.39 | 3.73 | 10 |
| PADM-$\mathcal{L}_\lambda$ | 105.0202 | 35.0743 | 2.4273 | 0.3246 | 0.0682 | 0.0287 | 0.0227 | 0.0220 |
| PADM-$\mathcal{N}_\sigma$ | 1.3808 | 0.1470 | 0.0417 | 0.0245 | 0.0220 | 0.0221 | 0.0214 | 0.0227 |
| Real-world Experiment | | | | | | | | |
| $\epsilon_{DP}$ | 0.01 | 0.03 | 0.07 | 0.19 | 0.52 | 1.39 | 3.73 | 10 |
| PADM-$\mathcal{L}_\lambda$ | 0.4669 | 0.2379 | 0.1559 | 0.0641 | 0.0221 | 0.0024 | 0.0031 | 0.0006 |
| PADM-$\mathcal{N}_\sigma$ | 0.2751 | 0.0867 | 0.0290 | 0.0059 | 0.0029 | 0.0033 | 0.0021 | 0.0022 |

outperforms the three competitors in all the cases. PADM-$\mathcal{L}_\lambda$ outperforms the three competitors in the real-world experiment while remaining competitive in the synthetic experiment.

Table 2: Final objective function values (mean $\pm$ STD) of federated private ADMM, private ADMM, PNSGD, PADM-$\mathcal{L}_\lambda$ (ours), and PADM-$\mathcal{N}_\sigma$ (ours).

| Synthetic Experiment | | | | |
|---|---|---|---|---|
| $\epsilon_{DP}$ | 0.01 | 0.03 | 0.07 | 0.19 |
| Federated Private ADMM | $27785.87 \pm 5126.50$ | $617.35 \pm 109.51$ | $110.23 \pm 9.86$ | $57.74 \pm 5.72$ |
| Private ADMM | $253.29 \pm 58.27$ | $37.99 \pm 9.85$ | $6.80 \pm 1.94$ | $2.70 \pm 1.02$ |
| PNSGD | $7.78 \pm 1.34$ | $1.51 \pm 0.22$ | $0.70 \pm 0.07$ | $0.58 \pm 0.04$ |
| **PADM-$\mathcal{L}_\lambda$** | $105.20 \pm 4.39$ | $35.25 \pm 4.81$ | $2.61 \pm 0.45$ | $\mathbf{0.50 \pm 0.06}$ |
| **PADM-$\mathcal{N}_\sigma$** | $\mathbf{1.56 \pm 0.32}$ | $\mathbf{0.33 \pm 0.03}$ | $\mathbf{0.22 \pm 0.007}$ | $\mathbf{0.20 \pm 0.004}$ |
| $\epsilon_{DP}$ | 0.52 | 1.39 | 3.73 | 10 |
| Federated Private ADMM | $3.87 \pm 0.51$ | $3.99 \pm 0.66$ | $4.89 \pm 1.25$ | $4.87 \pm 1.27$ |
| Private ADMM | $2.12 \pm 0.76$ | $2.04 \pm 0.87$ | $1.99 \pm 0.81$ | $2.03 \pm 0.70$ |
| PNSGD | $0.56 \pm 0.02$ | $0.57 \pm 0.02$ | $0.56 \pm 0.02$ | $0.56 \pm 0.02$ |
| **PADM-$\mathcal{L}_\lambda$** | $\mathbf{0.25 \pm 0.01}$ | $\mathbf{0.21 \pm 0.005}$ | $\mathbf{0.20 \pm 0.004}$ | $\mathbf{0.20 \pm 0.004}$ |
| **PADM-$\mathcal{N}_\sigma$** | $\mathbf{0.20 \pm 0.004}$ | $\mathbf{0.20 \pm 0.004}$ | $\mathbf{0.20 \pm 0.004}$ | $\mathbf{0.20 \pm 0.004}$ |

| Real-world Experiment | | | | |
|---|---|---|---|---|
| $\epsilon_{DP}$ | 0.01 | 0.03 | 0.07 | 0.19 |
| Federated Private ADMM | $1.61 \pm 0.54$ | $1.04 \pm 0.05$ | $0.98 \pm 0.01$ | $0.97 \pm 0.00$ |
| Private ADMM | $11.83 \pm 1.43$ | $2.60 \pm 0.28$ | $1.07 \pm 0.07$ | $0.85 \pm 0.03$ |
| PNSGD | $0.95 \pm 0.00$ | $0.95 \pm 0.00$ | $0.78 \pm 0.00$ | $0.78 \pm 0.00$ |
| **PADM-$\mathcal{L}_\lambda$** | $\mathbf{0.92 \pm 0.20}$ | $\mathbf{0.69 \pm 0.14}$ | $\mathbf{0.61 \pm 0.09}$ | $\mathbf{0.51 \pm 0.05}$ |
| **PADM-$\mathcal{N}_\sigma$** | $\mathbf{0.73 \pm 0.06}$ | $\mathbf{0.54 \pm 0.04}$ | $\mathbf{0.48 \pm 0.01}$ | $\mathbf{0.46 \pm 0.01}$ |
| $\epsilon_{DP}$ | 0.52 | 1.39 | 3.73 | 10 |
| Federated Private ADMM | $0.97 \pm 0.00$ | $0.97 \pm 0.00$ | $0.97 \pm 0.00$ | $0.97 \pm 0.00$ |
| Private ADMM | $0.82 \pm 0.02$ | $0.82 \pm 0.02$ | $0.81 \pm 0.03$ | $0.82 \pm 0.02$ |
| PNSGD | $0.61 \pm 0.01$ | $0.61 \pm 0.01$ | $0.60 \pm 0.04$ | $0.62 \pm 0.04$ |
| **PADM-$\mathcal{L}_\lambda$** | $\mathbf{0.47 \pm 0.02}$ | $\mathbf{0.45 \pm 0.01}$ | $\mathbf{0.45 \pm 0.02}$ | $\mathbf{0.45 \pm 0.01}$ |
| **PADM-$\mathcal{N}_\sigma$** | $\mathbf{0.45 \pm 0.01}$ | $\mathbf{0.45 \pm 0.01}$ | $\mathbf{0.45 \pm 0.01}$ | $\mathbf{0.45 \pm 0.01}$ |

**Accuracies for the real-world experiment on the Adult data set:** Accuracies (mean $\pm$ STD) of federated private ADMM, private ADMM, PNSGD, PADM-$\mathcal{L}_\lambda$, and PADM-$\mathcal{N}_\sigma$ for the real-world experiment on the Adult data set are shown in Table 3. The model is trained on the training set and the accuracy is obtained on the test set, which is the ratio of correctly classified samples to the total test samples. PADM-$\mathcal{L}_\lambda$ and PADM-$\mathcal{N}_\sigma$ achieve highest accuracies in most cases.

Table 3: Accuracies (mean $\pm$ STD) of federated private ADMM, private ADMM, PNSGD, PADM-$\mathcal{L}_\lambda$ (ours), and PADM-$\mathcal{N}_\sigma$ (ours) for real-world experiment on the Adult data set.

| | Classification Accuracy (%) | | | |
|---|---|---|---|---|
| $\epsilon_{DP}$ | 0.01 | 0.03 | 0.07 | 0.19 |
| Federated Private ADMM | $39.94 \pm 15.34\%$ | $41.04 \pm 15.76\%$ | $72.99 \pm 3.85\%$ | $75.00 \pm 0.00\%$ |
| Private ADMM | $68.52 \pm 3.99\%$ | $68.75 \pm 2.42\%$ | $74.61 \pm 1.01\%$ | $76.54 \pm 1.83\%$ |
| PNSGD | $25.00 \pm 0.00\%$ | $25.00 \pm 0.00\%$ | $73.59 \pm 0.46\%$ | $73.81 \pm 0.36\%$ |
| **PADM-$\mathcal{L}_\lambda$** | $53.74 \pm 12.88\%$ | $\mathbf{71.96 \pm 6.30}\%$ | $74.27 \pm 1.60\%$ | $\mathbf{77.64 \pm 1.76}\%$ |
| **PADM-$\mathcal{N}_\sigma$** | $\mathbf{73.43 \pm 3.71}\%$ | $\mathbf{75.65 \pm 0.80}\%$ | $\mathbf{77.44 \pm 2.16}\%$ | $\mathbf{78.95 \pm 1.71}\%$ |
| $\epsilon_{DP}$ | 0.52 | 1.39 | 3.73 | 10 |
| Federated Private ADMM | $75.00 \pm 0.00\%$ | $75.00 \pm 0.00\%$ | $75.00 \pm 0.00\%$ | $75.00 \pm 0.00\%$ |
| Private ADMM | $77.45 \pm 1.29\%$ | $78.03 \pm 0.82\%$ | $77.15 \pm 1.08\%$ | $76.91 \pm 2.61\%$ |
| PNSGD | $78.46 \pm 1.29\%$ | $78.48 \pm 1.29\%$ | $76.23 \pm 4.00\%$ | $76.08 \pm 5.62\%$ |
| **PADM-$\mathcal{L}_\lambda$** | $\mathbf{79.16 \pm 1.41}\%$ | $\mathbf{79.85 \pm 1.17}\%$ | $\mathbf{80.00 \pm 1.06}\%$ | $\mathbf{78.79 \pm 2.25}\%$ |
| **PADM-$\mathcal{N}_\sigma$** | $\mathbf{79.47 \pm 1.51}\%$ | $\mathbf{79.03 \pm 1.81}\%$ | $\mathbf{79.77 \pm 1.04}\%$ | $\mathbf{78.47 \pm 1.62}\%$ |

