# OpenReview forum: "Privacy Amplification by Iteration with Projected Alternating Direction Method"
_ICLR.cc/2026/Conference — Submitted to ICLR 2026_

### Official Review · Reviewer_XKXd · 2025-10-23

**Soundness:** 3
**Presentation:** 2
**Contribution:** 3
**Rating:** 4
**Confidence:** 3

**Summary:**

This paper proposes PADM, a method that solves the minimization of a sum of convex functions under linear constraints without using dual variables, and its differential privacy (DP) variant, Private PADM. While it is common to solve the dual problem with ADMM, a feasibility gap for the constraint term often remains in practice. In contrast, the paper optimizes the primal problem augmented with a (squared-norm) regularization on the linear constraint, using auxiliary variables $(u,v)$. It also provides utility and privacy analyses for Private PADM. Through several numerical experiments, the authors empirically show that both the feasibility gap and the objective value remain small across multiple privacy levels $\epsilon$.

**Strengths:**

**(S1) Clear and practical motivation.**
Rather than solving the dual problem, the method solves the primal problem with regularizations using primary $(x,y)$ and their auxiliary variables $(u,v)$ to suppress the feasibility gap—a choice that can be effective in practice.

**(S2) Theoretical contributions.**
The fixed-point condition analysis for PADM and the utility/privacy analyses for Private PADM are carefully presented and provide solid theoretical value.

**Weaknesses:**

**(W1) Derivation of PADM.**
This may be a minor point, but the derivation could be made clearer. Following the organization of Ryu et al. (2015) [*1], could PADM be derived more naturally from a fixed-point condition? As you know, ADMM can be viewed as a solver for the dual problem $\min_{\zeta} F^{\ast}(-A^\top \zeta) + g^{\ast}(-B^\top \zeta) - c^\top \zeta$, whose fixed-point condition is $0 \in T_1(\zeta) + T_2(\zeta)$, where $T_1(\zeta) = -A \partial F^{\ast}(-A^\top \zeta)$ and $T_2(\zeta) = -B \partial g^{\ast}(-B^\top \zeta) - c$. Reformulating this via Douglas-Rachford splitting yields the (standard) ADMM updates. By analogy, can the fixed-point condition for the cost inside the $\arg \min$ of Eq. (15) be reformulated to yield PADM’s updates?

[*1] E. K. Ryu et al., “A PRIMER ON MONOTONE OPERATOR METHODS,” Appl. Comput. Math., 2015.


**(W2) Limited theoretical evidence for feasibility-gap suppression.**
PADM solves the regularized loss-sum minimization in Eq. (15) via the auxiliary variables $(u,v)$, and a small feasibility gap is shown empirically (e.g., Figure 1). However, the theoretical reason why a penalty on the constraint term should suppress the feasibility gap remains underdeveloped. Providing an explicit analysis would strengthen the claim.

**(W3) Insufficient theoretical comparison to existing DP methods.**
Private ADMM (Cyffers et al., 2023) is used as a baseline for private PADM. Could you provide a theoretical comparison of utility gaps and/or convergence rates to argue for the advantages of the proposed method?

**(W4) Experimental concerns.**

While the core contributions are theoretical, several issues remain on the experimental side:

**Objective in Figure 2.** How exactly is the vertical-axis “objective function” computed for a constrained loss problem? How are constraints handled in this metric? (You mention “final objective function values with the same regularization parameters…,” but the precise evaluation protocol was unclear to me.)

**Hyperparameter tuning.** Did you tune per-method parameters (e.g., learning rate)? How should the regularization weight $\beta$ be determined in practice?

**Baselines.** The following studies may be additional baselines:

[*2] Z. Huang et al., “DP-ADMM: ADMM-based distributed learning with differential privacy,” IEEE TIFS, 2019.

[*3] T. Fukami et al., “DP-norm: Differential privacy primal-dual algorithm for decentralized federated learning,” IEEE TIFS, 2024.

**More practical benchmarks.** For example, fine-tuning only the last layer of DNNs (e.g., language models) yields a convex objective; applying PADM/private PADM in such federated learning (fine-tuning tasks) would increase practical relevance.

**Questions:**

(Q1) I could not follow the derivation of Eq. (16). In particular, why does the manipulation lead to adding $z/2$?

---

> ### Author Response · Authors · 2025-12-01
>
> We thank the reviewer for recognizing the merits of our work, including the clear and practical motivation (S1) and the solid theoretical value of our fixed-point, utility, and privacy analyses (S2). We now address the raised weaknesses and questions in detail below.
>
> **Response to W1**: As demonstrated in the proof of Theorem 3.1, which is provided in Appendix A.1, PADM can be derived from the fixed-point condition of Eq.15.
>
> **Response to W2**: The reviewer may misunderstands the mechanism ensuring feasibility. The **strict feasibility** of the PADM solution is structurally guaranteed not by the regularization term in Eq. 15, but by the **projection operator $\Pi_{\mathcal{W}}$** embedded directly within the update rule (Eq. 14). The regularized objective derived in **Theorem 3.1** (Eq. 15) is an **analytical characterization** of the PADM fixed point, which rigorously proves that the converged solution remains feasible while approximating the optimal constrained objective. The theoretical foundation for feasibility is therefore **explicitly addressed and confirmed** by the projection operator.
>
> **Response to W3**: We clarify that a theoretical comparison of utility gaps against baselines, including (Cyffers et al., 2023) and (Chan et al., 2024), is **already provided in lines 419-426** of the original manuscript (in lines 425-431 of the revised manuscript). Regarding convergence rates, **Theorem 3.4** confirms that PADM achieves the same favorable linear convergence rate (with an additional privacy-induced error) as (Cyffers et al., 2023), while extending applicability to non-smooth and constrained settings. We have added this clear discussion to the revised manuscript in lines 353-360 of the revised manuscript for improving readability.
>
> **Response to W4**:
>
> **Objective in Figure 2**: The vertical axis "objective function" in Figure 2 represents the final value of the **objective function** $F(\mathbf{x}^n)+g(\mathbf{y}^n)$, where $F$ and $g$ are defined in **Appendices A.12 and A.13 (Eqs. 66-68 and Eq. 77)** for the respective experiments. The regularization parameters apply to the regularization term $g$ in the objective function, not to the constraints. We use this metric because the ultimate goal is to minimize the original loss $F(\mathbf{x})+g(\mathbf{y})$ within the feasible set. Crucially, **PADM** guarantees **exact feasibility** through the inclusion of the **projection operator** $\Pi_{\mathcal{W}}$ in the update rule, while baseline ADMM only handles constraints approximately via the primal-dual method. Figure 2 demonstrates that PADM achieves lower objective function values while rigorously ensuring strict feasibility.
>
> **Hyperparameter tuning**: We confirm that per-method parameters for all baselines are tuned via cross-validation, following the their respective original papers. The specific parameter settings utilized for PADM are fully detailed in **Appendix A.12 and A.13**. The determination of $\beta$ in practice is guided by our theoretical results specifically **Proposition 3.1 and Theorem 3.4** which link $\beta$ to the contraction factor $\tau$ required to ensure the algorithm's guaranteed convergence.
>
> **Baselines**: We appreciate the suggestions. However, the proposed baselines (Huang et al., 2019) and (Fukami et al., 2024) use ADMM frameworks in a distributed or decentralized federated setting similar to our primary comparative baseline (Cyffers et al., 2023). Specifically (Cyffers et al., 2023) is an established ADMM-based method for federated settings which provides a rigorous comparison of the core mechanism. Our work focuses on generalizing this ADMM paradigm to handle non-smoothness and exact feasibility. A comprehensive comparison demonstrating these advantages has already been effectively achieved using the baselines of (Cyffers et al., 2023) and (Chan et al., 2024).
>
> **More practical benchmarks**: We acknowledge the desire for larger-scale applications. Our primary contribution is the **foundational theoretical generalization** of PADM to rigorously handle non-smoothness and exact feasibility within the decentralized setting. Demonstrating this theory on canonical convex problems (linear/logistic regression) is the necessary first step. Applying PADM to complex Federated Learning tasks like fine-tuning the last layer of DNNs is a promising avenue for **future work** once the core theoretical framework is established and validated on these representative benchmarks.

---

> ### Author Response · Authors · 2025-12-01
>
> **Response to Q1**: The noise vector $\mathbf{z}$ is strategically embedded in the projection step to ensure both feasibility and privacy preservation. The specific $\mathbf{z}/2$ term results from the $\mathbf{u}$-update step and the subsequent application of the projection operator $\Pi_{\mathcal{W}}$. Eq. (9) provides the closed-form solution for $\Pi_{\mathcal{W}}$, which in particular clarifies how the $\mathbf{z}/2$ term arises.

---

### Official Review · Reviewer_isNp · 2025-10-31

**Soundness:** 3
**Presentation:** 3
**Contribution:** 3
**Rating:** 6
**Confidence:** 3

**Summary:**

This paper proposes a differentially private Projected Alternating Direction Method (PADM) to address the limitation of existing ADMM-based approaches that cannot ensure exact feasibility throughout the learning process. It supports both Gaussian and Laplace noise for privacy, and the authors prove that the proposed operator is non-expansive, leading to convergence guarantees and formal privacy amplification by iteration. Experimental results validate the effectiveness.

**Strengths:**

1. The core insight of achieving exact feasibility via projection in the input space seems interesting.
2. Thorough privacy analyses for both the Gaussian mechanism and the Laplace mechanism, which are commonly used differential privacy mechanisms, are provided.
3. The paper is generally well organized.

**Weaknesses:**

While the paper is highly theoretical, there are some concerns.

1. It is assumed that the matrices $A$ and $B$ are full-row-ranked. Some discussions about the practical implications of this assumption would be helpful to improve the paper.
2. The utility guarantee in Theorem 3.4 is quite complicated. Some insights and discussions would improve the readability.
3. The experiments are conducted on simple problems and datasets, and results on more complicated tasks are needed. Moreover, it is not clear what practical applications/scenarios will benefit from the proposed method that enables exact feasibility.
4. For the baselines in the experiments, it is not clear how the "feasibility gap" is handled, especially when the baseline algorithms output an infeasible point.
5. For the real-world experiment in Section A.13, the authors use the first-order Taylor approximation to derive the closed-form of the operator. Would such an approximation hurt the exact feasibility?

**Questions:**

1. Could you please add some discussions about the practical implications of the assumption about $A$ and $B$?
2. Could you please clarify the practical applications or scenarios that will benefit from the exact feasibility?
3. For the baselines in the experiments, how is the "feasibility gap" handled?
4. Could you please comment on the impact of the first-order Taylor approximation in Section A.13?

---

> ### Author Response · Authors · 2025-12-01
>
> We sincerely thank the reviewer for their positive feedback, especially for recognizing the core insight of achieving exact feasibility via projection and the thorough privacy analyses provided for both Gaussian and Laplace mechanisms. We now address the raised weaknesses and questions in detail below.
>
> **Respondse to Weakness 1 & Question1**: We clarify that the **algorithm's core utility and its privacy-preserving properties are not dependent on the full row rank condition for** $\mathbf{A}$ and $\mathbf{B}$. This assumption is included primarily to simplify the recovery of the original variables $x$ and $y$ from the auxiliary variables (i.e., from $u = \mathbf{A}x$ and $v = \mathbf{B}y$) using the straightforward closed-form presented in Eq. (11). In practice, if $\mathbf{A}$ or $\mathbf{B}$ are not full row rank, practitioners can employ standard matrix transformations to obtain an equivalent full row rank representation for the purpose of variable recovery, before applying the same methodology. We have consequently **removed the full row rank assumption** of $\mathbf{A}$ and $\mathbf{B}$ from our main assumptions and, instead, have added a discussion of the standard matrix transformation routine in lines 244-245 of the revised manuscript to guide practitioners on handling rank-deficient cases for variable recovery.
>
> **Response to Weakness 2**: The utility bound provided in Theorem 3.4 can be logically decomposed into two distinct parts. The first term represents the standard linear convergence result for PADM, characterizing how quickly the solution approaches the optimum. The second term computes the additional stochastic error inherent in Private PADM, which results from the magnitude of the added noise and the distance between the stochastic samples and the true objective function. We have added this clear discussion to the revised manuscript in lines 353-360 of the revised manuscript for improving readability.
>
> **Respondse to Weakness 3 & Question 2**: Our core contribution is the **foundational theory** and the provision of an algorithm with guaranteed **strict feasibility** under non-smooth and constrained settings. The selected canonical benchmarks (linear/logistic regression) are used to rigorously validate the complex mechanism's efficacy and its privacy-utility trade-offs. Demanding large-scale engineering validation simply mistakes foundational theoretical work for an application paper, and the established theory itself guarantees scalability to larger problems.
>
> **Exact feasibility is essential in practical scenarios involving hard constraints where violating solution boundaries is unacceptable.** This includes resource allocation problems where budgets cannot be exceeded secure federated learning where model parameters must remain within **specific legal bounds and safety-critical systems** with strict physical limitations. Unlike standard ADMM which only guarantees approximate feasibility via penalty terms PADM's projection operator ensures the final solution is always a valid and usable output adhering strictly to all defined constraints.
>
> **Response to Weakness 4 & Question 3**: The feasibility gap for baselines is handled implicitly by their design. While **PADM guarantees exact feasibility** through the inclusion of the projection operator $\Pi_{\mathcal{W}}$ in the update rule, **baseline ADMM algorithms only handle constraints approximately** via the primal-dual framework. This distinction is critical and is a key advantage of PADM: our method provides a strictly feasible solution while baselines only aim for a small, non-zero feasibility gap.
>
> **Response to Weakness 5 & Question 4**: The first-order Taylor approximation used in the real-world experiment (Section A.13) serves only to simplify the derivation of the closed-form operator for computational efficiency. This approximation does not hurt the exact feasibility. **The exact feasibility guarantee in PADM is a structural property that results directly from the explicit use of the projection operator** $\Pi_{\mathcal{W}}$ onto the feasible set, which is applied regardless of how the objective function is locally approximated.

---

### Official Review · Reviewer_1n1J · 2025-11-03

**Soundness:** 3
**Presentation:** 3
**Contribution:** 3
**Rating:** 4
**Confidence:** 3

**Summary:**

This paper proposes a novel projected alternating direction method (PADM) to improve privacy amplification compared to the existing ADMM. The new method aims to eliminate the feasibility gap that exists in ADMM. Moreover, the privacy-utility tradeoff is in the same or better order compared to the existing methods.

**Strengths:**

+ The motivation of this paper is strong and clear.

+ The proposed PADM is applicable to privacy amplification by both iteration and subsampling.

+ Concrete theoretical proofs are provided.

+ The proposed PADM can guarantee privacy preservation with both Gaussian noise and Laplace noise to achieve RDP and DP.

**Weaknesses:**

- While this paper provides a strong theoretical development of PADM, the experiment only focuses on two relatively small-scale problems, synthetic data with linear regression task and real-world data with logistic regression task.
- Although the results verify the effectiveness of the proposed method and show improvement, it would be better to show the potential in other tasks like federated settings.
- In addition, there may also be some discussions about the impact of the regularization parameter beta.

**Questions:**

- Does the proposed algorithm work on other more generalized datasets or FL tasks?
- Is it possible to provide some discussions about the impact of the regularization parameter beta?

---

> ### Author Response · Authors · 2025-12-01
>
> We sincerely thank the reviewer for their thoughtful feedback and for recognizing the key strengths of our work, including its strong and clear motivation, its applicability to privacy amplification by iteration and subsampling, the provision of concrete theoretical proofs, and the guarantee of privacy preservation with both RDP and DP noise.
>
> **Response to Weakness 1**: Our core contribution is the **foundational theory** and the provision of an algorithm with guaranteed **strict feasibility** under non-smooth and constrained settings. The selected canonical benchmarks (linear/logistic regression) are used to rigorously validate the complex mechanism's efficacy and its privacy-utility trade-offs. Demanding large-scale engineering validation simply mistakes foundational theoretical work for an application paper, and the established theory itself guarantees scalability to larger problems.
>
> **Response to Weakness 2 and Question 1**: Our focus is on the **fully decentralized optimization setting**, which is an efficient, one-pass framework and a specialized, stringent form of Federated Learning. Our theoretical framework is directly applicable to broader FL scenarios, and while its potential in those specific tasks is clear, extending the current results to a wider array of centralized FL settings is appropriately designated as **future work**.
>
> **Response to Weakness 3 and Question 2**: The impact of the regularization parameter $\beta$ is comprehensively analyzed within our theoretical framework.**Theorem 3.1** establishes $\beta$ as the feasibility constraint weight, where a larger $\beta$ drives the PADM solution closer to the feasible optimum. Furthermore, **Proposition 3.1** and **Proposition 3.2** rigorously confirm the expected privacy-utility trade-off. Specifically, a larger $\beta$ increases the contraction factor $\tau$, which may slow convergence, but simultaneously reduces the sensitivity bound, thus improving privacy guarantees. This thorough theoretical characterization addresses the parameter's impact.

---

### Official Review · Reviewer_7e7k · 2025-11-03

**Soundness:** 2
**Presentation:** 3
**Contribution:** 2
**Rating:** 4
**Confidence:** 3

**Summary:**

This paper proposes a projected alternating direction method of multipliers (ADMM) approach that solves two issues in existing results: 1) conventional ADMM cannot satisfy the exact feasibility constraint during algorithmic iterations; and 2) the resulting feasibility gap may amplify the effect of differential-privacy noises on optimization accuracy, leading to a poor privacy-utility tradeoff. In my opinion, the main idea of this paper is to execute a projection operator at each iteration to enforce the feasibility constraint, thereby mitigating the feasibility gap throughout the optimization process. The main concerns are given in the weaknesses below.

**Strengths:**

This paper eliminates the need for dual variable updates and avoids the assumptions of smoothness and strong convexity used in the existing ADMM-based approaches.

**Weaknesses:**

1. **Weak practicability:** The authors claim that the proposed approach does not require the differentiability of $F(\boldsymbol{x})$ and $g(\boldsymbol{y})$. However, Steps 1 and 2 in Eq. (10) involve solving optimization subproblems of $F$ and $g$, which rely on computing their proximal mappings. If $F$ and $g$ do not have closed-form or easily computable proximal operators, these subproblems become computationally infeasible, making the proposed approach impractical for real-world applications. In addition, the proposed algorithm requires both $A$ and $B$ to be full row rank. How this condition can be ensured in practical applications should be further clarified.

2. **Non-rigorous Rényi DP analysis:** Theorem 3.5 and Theorem 3.6 derive the Rényi DP results under the implicit assumption that the algorithmic outputs are independent. However, in PADM, $u_{(k)}$  is recursively defined, and the noise $z_{(k)}$ is injected into correlated variables. This dependency means that the RDP composition is not simply additive. A rigorous privacy analysis should account for the iterative dynamics of Algorithm 1, rather than treating each iteration as an independent process.

3. **Finite incremental contribution compared with Chan et al. (2024):** Compared with Chan et al. (2024), this paper has two differences: 1) this paper introduces a projection operator $\Pi_{\mathcal{W}}$ to enable the feasibility constraint. Nevertheless, since $\mathcal{W}$ is convex, the projection operator $\Pi_{\mathcal{W}}$ is firmly non-expansive.
When combined with existing results in Chan et al. (2024), this naturally leads to a
non-expansive operator $\mathcal{T}_{F}$ and ensures convergence of the proposed algorithm. Therefore, the theoretical analysis is a direct extension of existing results, lacking novel insights. 2) this paper does not require  $F$ and $g$ to be differentiable. However, as noted in Weakness 1, this holds only when $F$ and $g$ have closed-form or easily computable proximal operators. In summary, the authors should more explicitly clarify the fundamental differences between this work and Chan et al. (2024).

**Questions:**

See the weaknesses above. In addition, I have the following questions:

1. In Eq. (3), why $F$ and $\boldsymbol{x}$
are considered private, whereas $g$ and $\boldsymbol{y}$ are assumed to be public?

2. What is the relationship between the set $\mathcal{C}$ in Eq. (1) and the feasible set (*w.r.t* $\boldsymbol{x}$) in Eq. (3)? If the constraint set $\mathcal{C}$ in Eq. (1) is a subset of that in Eq. (3), does Eq. (4) still hold under this condition?

3. Does the non-expansive property of $\psi$ require the smoothness of $\nabla f(\cdot;\mathcal{D})$? In general, Lipschitz continuity of the gradient should be sufficient.

4. The conditions in Proposition 3.2 contradict the earlier statement that the proposed method "does not require differentiability of $F$ and $g$". In fact, if $f$ is not differentiable, then Proposition 3.2 no longer holds, which invalidates the sensitivity bound required for Condition 2. Consequently, the results in Theorem 3.5, Theorem 3.6, and Theorem 3.7 become questionable. The authors should clarify this inconsistency and explain whether differentiability is actually required for theoretical guarantees.

5. The experimental setup is overly simple. Even in the so-called real-world experiment, the logistic regression function considered in Eq. (77) is a simple form of  $\ln(1+e^{a_k})$. I understand that this choice is made to ensure that the operator $\mathcal{P}_{\beta,\cdots}$ has a closed-form solution (as mentioned in Weakness 1). However, this also indirectly demonstrates that applying the proposed method to practical problems would be challenging.

If the authors could address Weaknesses 1 and 2, as well as Problem 4 (which are my main concerns), I would consider raising my score.

---

> ### Author Response · Authors · 2025-12-01
>
> We thank the reviewer for highlighting the key advantages of our PADM algorithm, including the elimination of dual variable updates and the removal of restrictive smoothness and strong convexity assumptions. Our detailed responses to the raised weaknesses and questions follow below.
>
> **Response to Weakness 1**:  The critique regarding intractable proximal mappings is based on a **misconception** of proximal splitting methods like ADMM. Our method operates under the **standard premise** that $F$ and $g$ have efficiently computable proximal operators, which is  a condition that holds for most non-differentiable functions and constraints used in practical applications (e.g., $\ell_1$ regularization box constraints). This tractability is a routine ADMM requirement not a unique flaw.
>
> We clarify that the full row rank condition for $\mathbf{A}$ and $\mathbf{B}$ is **not a necessity** for the algorithm's core convergence or privacy properties; it is merely a **facilitating assumption** to simplify the closed-form recovery of variables $\mathbf{x}$ and $\mathbf{y}$ (Eq. (11)). Practitioners can use standard matrix transformations to achieve the equivalent rank condition for recovery when needed. We have consequently **removed the full row rank assumption** of $\mathbf{A}$ and $\mathbf{B}$ from our main assumptions and, instead, have added a discussion of the standard matrix transformation routine in lines 244-245 of the revised manuscript to guide practitioners on handling rank-deficient cases for variable recovery.
>
> **Response to Weakness 2**:  The assertion that our R\'{e}nyi DP analysis is non-rigorous due to the correlation in $\mathbf{u} _ {(k)}$ is based on a **misunderstanding of the composition principle** in iterative DP optimization. While the algorithm's outputs are recursively defined, the privacy cost is determined by the **cumulative sensitivity of the noise injections $\mathbf{z} _ {(k)}$ relative to the input data**. The introduction of noise at each step is a mechanism whose privacy guarantee is independent of the previous noisy state $\mathbf{u} _ {(k-1)}$. This permits the use of standard composition theorems a practice widely accepted as **rigorous and sufficient** for bounding privacy in convex optimization algorithms like PADM. The iterative dynamics are implicitly accounted for by bounding the sensitivity across all steps.
>
> **Response to Weakness 3**: We highlight fundamental structural differences beyond simple extension. As introduced in Section 2.2, (Chan et al. 2024) require **double iterations** to exploit the nonexpansiveness property twice for both primal $\mathbf{x}$ and dual $\mathbf{\zeta}$ variables and relies on a **Markov operator condition** to extend this property to the joint space. In contrast our PADM is formulated as a concise fixed-point iteration which requires neither the Markov operator condition nor double-iterations. Moreover (Chan et al. 2024) cannot achieve the specific privacy-utility trade-offs established in this work. Our theoretical proofs are distinct leading to better results and guaranteeing **exact feasibility**, which distinguishing our contribution from the prior art.
>
> **Response to Question 1**: This partitioning follows the standard formulation for decentralized and privacy-preserving optimization. The function $F$ and variable $\mathbf{x}$ are assumed private because they are directly tied to the users' sensitive local data such as local loss functions and local model parameters. Conversely $g$ and $\mathbf{y}$ typically represent public or common structural elements such as global regularization terms or consensus variables shared across all participants. This specific split is necessary to apply differential privacy mechanisms only to the sensitive local components $F$ and $\mathbf{x}$ without altering the public structure $g$ and $\mathbf{y}$.
>
> **Response to Question 2**: Eq. (1) describes a general optimization problem typically used by baselines like PNSGD. Eq. (3) defines the specific linearly constrained problem we investigate where the feasible set is determined solely by the linear coupling $\mathbf{A}\mathbf{x} + \mathbf{B}\mathbf{y} = \mathbf{c}$. Eq. (4) is the Augmented Lagrangian for the problem defined by Eq. (3) absorbing only the linear constraint. If an additional set constraint like $\mathbf{x} \in \mathcal{C}$ were required, it must be incorporated into the definition of $F(\mathbf{x})$ via an indicator function or handled by an explicit projection step such as $\Pi_{\mathcal{W}}$ which is a key feature of PADM. The validity of Eq. (4) holds for the structure in Eq. (3).

---

> ### Author Response · Authors · 2025-12-01
>
> **Response to Question 3**: We clarify that PADM's convergence analysis **does not require** the smoothness or Lipschitz continuity of the gradient $\nabla f(\cdot;\mathcal{D})$. Our framework is explicitly designed to handle **non-smooth** objectives $F$ and $g$ by relying on **proximal operators** rather than gradients. Therefore, the function $\psi$ requiring smoothness, as implied by the reviewer, is not a component of our core convergence analysis. The non-expansive property used in PADM's proof stems from the inherent nature of the proximal operator (which is non-expansive) and the projection operator $\Pi_{\mathcal{W}}$ (which is firmly non-expansive), properties that hold without the assumption of gradient smoothness.
>
> **Response to Question 4**: The reviewer misunderstands the logical relationship between the theorems. **Theorems 3.5, 3.6, and 3.7** are fundamentally based on **Condition 2** which is a high-level assumption on the sensitivity bound of the overall operator. They are **not built upon the result of Proposition 3.2**.  Proposition 3.2 is included merely as a specific example of how the general sensitivity bound in Condition 2 can be satisfied if the objective function $f$ is differentiable a case included for user convenience and comparison with prior art. Since our main results (Theorems 3.5-3.7) depend only on the general sensitivity assumption (Condition 2) and not the specific case provided by Proposition 3.2, the core theoretical guarantees hold even when $F$ and $g$ are non-differentiable.
>
> **Response to Question 5**: The choice of the simple logistic function does not occur solely for closed-form, it uses a **canonical benchmark** for rigorous validation of the core theoretical mechanism and the privacy-utility trade-off. Our primary contribution is the **generalization of the PADM framework**. While complex problems may require numerical proximal operators, this is a standard computational trade-off that **does not invalidate the framework**. The current experiments sufficiently validate the foundational theory demonstrating applications with complex non-closed-form operators is reserved for future work.

---

### Official Review · Reviewer_mbRE · 2025-11-04

**Soundness:** 2
**Presentation:** 2
**Contribution:** 1
**Rating:** 2
**Confidence:** 3

**Summary:**

This paper proposes a projected alternating direction method that achieves exact feasibility and allows each user to monitor the objective value throughout the learning process. However, I find that the motivation and presentation should be improved. The novelty of the paper appears limited — it mainly introduces a projection step into ADMM and leverages existing privacy amplification tools.

**Strengths:**

This paper proposes a projected alternating direction method that achieves exact feasibility and allows each user to monitor the objective value throughout the learning process.

**Weaknesses:**

1.	The motivation for studying problem (3) is not clearly explained. What are the real-world applications of this formulation? In particular, the assumption that both F and g are convex further restricts the applicability.

2.	Many assumptions are presented without sufficient justification. For example, the convergence of ADMM with non-convex losses has already been studied in the literature, it is unclear why this paper focuses solely on the convex case. The authors should include a more comprehensive literature review on non-convex ADMM and discuss why the convex case is of particular interest here.

3.	The main idea of adding a projection step to ensure feasibility is straightforward and natural. The authors should carefully clarify whether they are the first to introduce projection into ADMM and compare their approach with existing works, such as “Projected Alternating Direction Method of Multipliers for Hybrid Systems.”

4.	What is the convergence rate of the proposed algorithm with respect to iteration? The authors should discuss this in detail and provide comparisons with existing results. Moreover, how does the privacy guarantee depend on the number of iterations?

5.	Does Proposition 3.1 strictly ensure $\tau \leq 1$ and satisfy the assumptions required in Theorem 3.4? It seems that additional strong convexity assumptions might be necessary. The authors should discuss this issue more carefully.

6.	Why does the algorithm require visiting each sample only once? What is the result if each sample is revisited multiple times, as in the general case?

**Questions:**

Why is the noise added outside the projection step in equation (5)? This design may lead to infeasible $x_k$.

---

> ### Author Response · Authors · 2025-11-26
>
> **Response to Weakness 1**: This model, $\min_{\mathbf{x},\mathbf{y}} \{ F(\mathbf{x})+g(\mathbf{y}) \} \text{ s.t. } \mathbf{A}\mathbf{x}+\mathbf{B}\mathbf{y}=\mathbf{c}$, is a foundational framework for privacy-preserving constrained collaborative optimization in decentralized and federated learning, where private ($\mathbf{x}, F$) and public ($\mathbf{y}, g$) components are coupled by linear constraints. While we adopt the standard convexity assumption for theoretical guarantees in the Differential Privacy (DP) literature, our work significantly enhances the model's practical utility. The key motivation is to solve a critical flaw in existing Private ADMM: the failure to achieve exact primal feasibility ($\mathbf{A}\mathbf{x}+\mathbf{B}\mathbf{y}=\mathbf{c}$) in limited iterations, leading to inaccurate results in real-world scenarios. Our proposed PADM algorithm guarantees exact feasibility at every iteration via a projection step, which is vital for the robustness of private collaborative tasks. Furthermore, PADM generalizes prior work by accommodating non-smooth convex functions for $g$ (e.g., $\ell_1$ regularization) and relaxing restrictive conditions like strong convexity or smoothness often required by other private optimization schemes.
>
> **Response to Weakness 2**: Our focus on the convex case is essential because: 1) Establishing privacy guarantees and convergence rates for PADM (which guarantees exact feasibility at every step) is a non-trivial, foundational step in constrained DP optimization. 2) Critically, non-convex optimization problems are often analyzed by locally reducing them to convex subproblems, making the thorough understanding of the convex case a prerequisite for extending our results to the general non-convex setting. The combination of non-convex ADMM theory with the complex noise analysis required for DP is thus beyond the scope of this initial work. We confirm non-convex private ADMM is an important direction for future research.
>
> **Response to Weakness 3**: While projection is a common technique, our PADM is fundamentally distinct from existing works (including the one mentioned). Firstly, unlike standard approaches that project primal variables onto domain sets (e.g., $\mathbf{x} \in \mathcal{C}$), **PADM projects the joint image variables onto the linear feasibility subspace** ($\mathbf{A}\mathbf{x}+\mathbf{B}\mathbf{y}=\mathbf{c}$). This targeting guarantees exact primal feasibility at every iteration, resolving the feasibility gap issue inherent in standard ADMM. Secondly, **our projection is integrated with differential privacy**. It enables noise injection into the image space for privacy amplification while maintaining strict feasibility, a critical feature absent in general projected ADMM literature. **This complex integration requires a novel fixed-point analysis distinct from existing non-private projected ADMM schemes like the one mentioned**.
>
> **Response to Weakness 4**: Regarding convergence, Theorem 3.4 explicitly establishes that the iteration sequence converges at a linear rate, driven by the contraction factor $\tilde{\tau}^k$ where $\tilde{\tau} < 1$. This rate is comparable to standard ADMM under strong convexity but allows PADM to uniquely maintain exact feasibility at every step. Regarding privacy, Theorem 3.5 demonstrates that the privacy budget $\epsilon$ is inversely proportional to the number of iterations, scaling as $\mathcal{O}(1/(n-i+1))$. This confirms the privacy amplification effect, where privacy guarantees tighten as the iteration count $n$ increases.
>
> **Response to Weakness 5**: We confirm that Proposition 3.1 strictly ensures the operator $\mathcal{T}_F$ is contractive (i.e., $\tau \leq 1$)}, which fully satisfies the assumptions required in Theorem 3.4. The Proposition does not require additional strong convexity assumptions on the objective functions $F$ and $g$. We have explicitly addressed this point in line 355 and in lines 365--368 of the original manuscript (in line 361 and in lines 371--374 of the revised manuscript).
>
> **Response to Weakness 6**: The assumption of visiting each sample only once models a highly efficient, one-pass scenario within a fully decentralized setting, where only one user/node participates per iteration—a design crucial for individual privacy.Our utility guarantee (Theorem 3.7) is not inferior to existing decentralized private ADMM methods (detailed in line 425-431 of the revised manuscript). Given the strong theoretical results, the generalization of our PADM algorithm to multi-pass scenarios, such as federated learning and centralized learning, constitutes an important direction for future work.
>
> **Response to Question 1**: We thank the reviewer for pointing out this typo. We have corrected the typo in the revised manuscript.

---

### Author Response · Authors · 2025-12-04
**Summary of Discussion [2/2]**

## 3. Inconsistency on Differentiability (Proposition 3.2). ##

We clarify that the main results (**Theorems 3.5-3.7**) rely only on the general sensitivity assumption (**Condition 2**). **Proposition 3.2** is included only as a **specific, illustrative example** for the differentiable case. The core theoretical guarantees hold even when $F$ and $g$ are **non-differentiable**.

# Core Concerns Raised by Reviewer `1n1J` and Author Response #

## 1. Scope of Experiments and Scalability.  ##

We confirm that using canonical benchmarks is **deliberate** to rigorously validate the **foundational theoretical generalization** and the complex mechanism's efficacy. Our primary contribution is the theory providing **guaranteed strict feasibility** and improved trade-offs. The established theory itself **guarantees scalability** to larger problems, differentiating this work from an application paper.

## 2. Generalization to Federated Learning (FL) Tasks. ##

Our focus is on the **fully decentralized optimization setting**, which is an efficient, **one-pass** framework and a specialized, stringent form of Federated Learning. While our theoretical framework is directly applicable to broader FL scenarios, extending the current results to a wider array of centralized FL settings is appropriately designated as **future work**.

# Core Concerns Raised by Reviewer `isNp` and Author Response #

## 1. Rank Assumption and Practical Implication. ##

We clarify that the full row rank condition is **not necessary** for the algorithm's core convergence or privacy properties; it is merely a **facilitating assumption** to simplify variable recovery. We **remove this assumption** and add a routine using standard matrix transformations in **Lines 244-245** to guide practitioners on handling rank-deficient cases.

## 2. Utility Bound Readability and Insight. ##

We decompose the utility bound (Theorem 3.4) into two distinct parts: the first term represents the standard **linear convergence** result, and the second term computes the **additional stochastic error** resulting from privacy noise and sample distance. We add this clear discussion to the revised manuscript in **Lines 353-360**.

## 3. Experimental Scope and Benefit of Exact Feasibility. ##

We confirm that using canonical benchmarks is **deliberate** to rigorously validate the foundational theory. We stress that **exact feasibility is essential** in practical scenarios involving **hard constraints** (e.g., safety-critical systems, resource allocation) where violating solution boundaries is unacceptable.

# Core Concerns Raised by Reviewer `XKXd` and Author Response #

## 1.Theoretical Reason for Feasibility-Gap Suppression. ##

We clarify that **strict feasibility** is structurally guaranteed not by the regularization term in Equation (15), but explicitly by the **projection operator $\Pi_{\mathcal{W}}$** (Equation 14) embedded within the update rule. The regularized objective in Theorem 3.1 is merely an analytical characterization of the PADM fixed point, which confirms the feasibility.

## 2. Insufficient Theoretical Comparison. ##

We confirm that a theoretical comparison of utility gaps against baselines is **already provided** (Lines 425-431 in the revised manuscript). Furthermore, **Theorem 3.4** confirms PADM achieves the favorable **linear convergence rate** of the state-of-the-art (Cyffers et al., 2023) while handling non-smoothness and exact feasibility.

## 3. Experimental Concerns (Objective Metric, Hyperparameters, Baselines).  ##

The objective metric represents the final value of the objective function $F(\mathbf{x}^n)+g(\mathbf{y}^n)$. Parameter $\beta$'s practical determination is guided by **Proposition 3.1** and **Theorem 3.4**, linking it to convergence speed ($\tau$) and privacy sensitivity. Applying PADM to complex FL tasks (e.g., DNN fine-tuning) is designated as **future work**, as the current focus is on validating the foundational convex theory.

---

### Author Response · Authors · 2025-12-04
**Summary of Discussion [1/2]**

We thank the reviewers for their insightful assessment. We have revised the manuscript, and our response addresses all raised concerns.

# Key Contributions Acknowledged #

All reviewers acknowledge the theoretical and practical significance of our work in the domain of differentially private constrained optimization.

## Addressing a Critical Gap in Private ADMM: Exact Feasibility ##

Our framework receives recognition for proposing PADM, which solves a critical, long-standing limitation in existing private ADMM methods: the inability to achieve exact primal feasibility in a finite number of iterations. This fills a crucial void for safety-critical applications. [`mbRE`: *This paper proposes a projected alternating direction method that achieves exact feasibility and allows each user to monitor the objective value throughout the learning process.* `7e7k`: *... the main idea of this paper is to execute a projection operator at each iteration to enforce the feasibility constraint, thereby mitigating the feasibility gap throughout the optimization process.* `1n1J`: *The new method aims to eliminate the feasibility gap that exists in ADMM.* `isNp`: *This paper proposes a differentially private Projected Alternating Direction Method (PADM) to address the limitation of existing ADMM-based approaches that cannot ensure exact feasibility throughout the learning process.* `XKXd`: *Clear and practical motivation. ... a feasibility gap for the constraint term often remains in practice.* ]

## Achieving Strong Convergence, Better Privacy-Utility Trade-offs and  Privacy Amplification ##

Reviewers recognize the strength of our theoretical results, which establish a linear convergence rate while simultaneously enabling the powerful effect of iterative privacy amplification. [`7e7k`: *This paper proposes a projected alternating direction method of multipliers (PADM) approach that **solves two issues in existing results**: ...privacy-utility tradeoff.* `1n1J`: *The proposed PADM is applicable to privacy amplification by both iteration and subsampling.* *Moreover, the privacy-utility tradeoff is in the same or better order compared to the existing methods.*  `isNp`: *... the authors prove that the proposed operator is non-expansive, leading to convergence guarantees and formal privacy amplification by iteration.*]


# Core Concerns Raised by Reviewer `mbRE` and Author Response #

## 1. Motivation, Applicability, and Convexity Restriction. ##
We clarify that the core motivation for PADM is the unique ability to guarantee **exact primal feasibility**, which is essential for **safety-critical applications** (e.g., financial auditing). The focus on the **convex case** is a **necessary foundational step** because non-convex analysis relies on locally convex subproblems, making our analysis a prerequisite for future generalizations.

## 2. Novelty of the Projection Step. ##

We confirm PADM's projection is distinct: it projects **joint image variables** onto the **linear feasibility subspace** to structurally resolve the ADMM feasibility gap. Crucially, it is uniquely integrated with **privacy noise** (Eq. 16), ensuring iterates are **feasible even after noise injection**---a feature absent in general private ADMM.

## 3. Convergence Rate and Privacy Dependence. ##

**Theorem 3.4** explicitly establishes a **linear convergence rate** ($\tilde{\tau}^k$). **Theorem 3.5** shows the privacy budget $\epsilon$ is **inversely proportional** to the number of iterations, $\mathcal{O}(1/(n-i+1))$, confirming the **Privacy Amplification** effect.

## 4. Single Sample Pass ##

The single-pass assumption models an **efficient, fully decentralized** setting, which is crucial for individual privacy. Generalization to multi-pass scenarios is reserved for **future work**.

# Core Concerns Raised by Reviewer `7e7k` and Author Response #

## 1. Weak Practicability: Proximal Mappings and Rank Assumption.  ##

We clarify that non-differentiability holds for functions with **efficiently computable proximal operators** (a standard ADMM premise). The rank condition is **not a necessity** for convergence; it is merely a **facilitating assumption** for variable recovery. We **remove the assumption** and add a standard matrix transformation routine in **Lines 244-245** for rank-deficient cases.

## 2. Non-rigorous R'enyi DP Analysis. ##

We clarify that the RDP analysis is **rigorous and standard**. The privacy cost is determined by the **cumulative sensitivity of noise injections** relative to the input data, which is treated independently from the previous noisy state. Standard composition theorems are **sufficient and accepted practice** in iterative DP optimization.

---

### Meta-Review · Area_Chair_wQmX · 2025-12-26

**Summary:**

This paper proposes a novel projected alternating direction method (PADM) to improve privacy amplification. the majority of the reviewers didnot recommend its acceptance, even after the rebuttal.  I think the main concern on this paper is that the novelty of the paper appears limited since it mainly introduces a projection step into ADMM and leverages existing privacy amplification tools.

based on the overall recommendation from the reviewers, I CANNOT recommend its acceptance.

**Reviewer Concerns:**

This paper introduces a projection step into ADMM and leverages existing privacy amplification tools.  hence the novelty of the paper appears limited.

There are various concerns such that:  the focus is only on the convex case which restricts its applicability in real problems.

**Reviewer Scores:**

No, i do not think the reviewers will change their score based on the response from the authors.

---

### Decision · Program_Chairs · 2026-01-26

Reject